# Temporal Alignment Guidance: On-manifold Sampling in Diffusion Models

## Abstract

Diffusion models have achieved remarkable success as generative models. However, even a well-trained model can accumulate errors throughout the generation process. These errors become particularly problematic when arbitrary guidance is applied to steer samples toward desired properties, which often breaks sample fidelity. In this paper, we propose a general solution to address the off-manifold phenomenon observed in diffusion models. Our approach leverages a time predictor to estimate deviations from the desired data manifold at each timestep, identifying that a larger time gap is associated with reduced generation quality. We then design a novel guidance mechanism, '*Temporal Alignment Guidance*' (TAG), attracting the samples back to the desired manifold at every timestep during generation. Through extensive experiments, we demonstrate that TAG consistently produces samples closely aligned with the desired manifold at each timestep, leading to significant improvements in generation quality across various downstream tasks.

## 1 Introduction

Diffusion models have shown remarkable performance as generative models across various domains, including image (Dhariwal & Nichol, 2021; Rombach et al., 2022), video (Liu et al., 2024; Polyak et al., 2024), audio Kong et al. (2021); Popov et al. (2021), language Austin et al. (2021), and molecular generation (Hoogeboom et al., 2022). A key factor in their success is the ability to perform guided generation, where conditions from different modalities can be effectively injected during the generative process (Dhariwal & Nichol, 2021; Ho & Salimans, 2021).

Recently, diffusion models have been applied to a variety of real-world use cases, such as black-box optimization (Krishnamoorthy et al., 2023), personalization (Zhang et al., 2023), and inverse problems (Chung et al., 2023). These downstream applications often require modifications to the standard sampling procedure, incorporating an additional guidance term during the reverse process of the diffusion model. This guidance term steers the generated samples towards desired properties relevant to the specific downstream task (Graikos et al., 2022; Wang et al., 2024; Wei et al., 2024). Notably, several works have demonstrated the ability to guide samples even towards conditions unseen during training, a technique often referred to as training-free guidance (Chung et al., 2023; Bansal et al., 2024).

However, naively modifying the originally learned reverse process of diffusion models can catastrophically break other basic properties, as it may lead samples toward low density regions where the output of diffusion model is unreliable (Song & Ermon, 2019). These score approximation errors can accumulate over each timestep (Chen et al., 2023b; Oko et al., 2023) which contributes to the final generated samples deviate significantly from the true data manifold, resulting in unrealistic outputs (Shen et al., 2024; Guo et al., 2024). In this work, we refer to this problem as the 'off-manifold' phenomenon in diffusion models and demonstrate that it can pose a significant challenge to their practical applications.

To address the off-manifold problem in diffusion models, we introduce 'Temporal Alignment Guidance' (TAG), a general solution designed to mitigate score approximation error induced by arbitrary modifications to the reverse process. Unlike traditional approaches that rely on fixed timesteps in the reverse process, TAG leverages the inherent uncertainty of the time variable by representing it as a probability distribution over a range of possible values. This novel guidance term is designed to steer samples back to the higher density region, where learned score of the model becomes reliable,

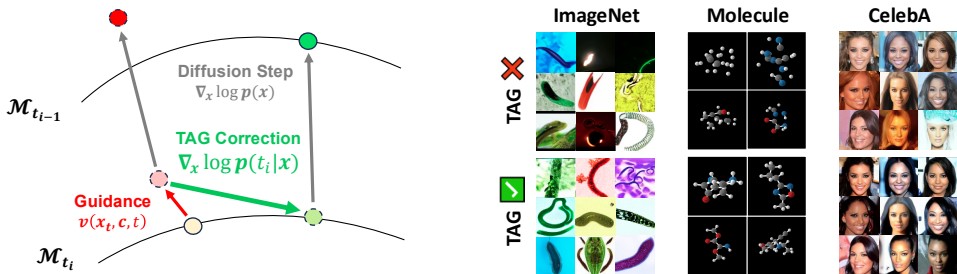

**Figure 1:** Overview of TAG algorithm. (Left) Without TAG, external guidance pushes samples off-manifold, causing the standard diffusion step $\nabla_x \log p(x)$ to miss the target manifold $\mathcal{M}_{t_{i-1}}$. TAG's correction actively steers the sample back to the correct manifold $\mathcal{M}_{t_i}$, ensuring the diffusion step accurately reaches the desired manifold $\mathcal{M}_{t_{i-1}}$. (Right) Applying TAG can greatly improve the fidelity in conditional generation tasks with target conditions: worm for ImageNet, polarizability $\alpha$ for Molecule, female and black hair for CelebA.

thereby improving sample quality while providing control in downstream tasks. This mechanism is visually summarized in Figure 1 (Left).

Through extensive experiments, we show that TAG significantly improves the quality of generated samples across multiple domains and tasks, as demonstrated in Figure 1 (Right). Promising results of TAG on these diverse scenarios implies that TAG could indeed serve as a universal solution for mitigating the off-manifold phenomenon in diffusion models, a common issue that arises in numerous downstream tasks but yet to be solved. We believe that this work represents an important stepping stone toward achieving reliable generation for real-world applications using diffusion models.

Our main contributions can be summarized as follows:

- We identify off-manifold phenomena in diffusion models across multiple scenarios and demonstrate that these phenomena can be significantly amplified when the learned reverse process of the original diffusion model is arbitrarily adjusted.
- We design a novel framework, '*Temporal Alignment Guidance*' (TAG), which pushes the samples toward the desired manifold at each timestep during generation and provide theoretical guarantees.
- We demonstrate that TAG significantly improves sample quality through extensive experiments in various domains and tasks, achieving state-of-the art results.

## 2 OFF-MANIFOLD PHENOMENON IN DIFFUSION MODELS

Off-manifold phenomenon happens in each timestep if the sample is tilted towards the low density region of true marginal distribution $p_t(\mathbf{x})$, which represents the distribution of a noisy sample $\mathbf{x}$ at timestep $t$. Below, we list typical situations where off-manifold phenomenon can occur in diffusion models.

**Controlling by external guidance** Anderson (1982) shows the forward process of diffusion model can be reversed once a score function $\nabla_{\mathbf{x}} \log q_t(\mathbf{x})$ of marginal distribution $q_t$ is given for each $t$ by the following reverse SDE:

$$d\mathbf{x} = \left[\mathbf{f}(\mathbf{x}) - g^2(t)\nabla_{\mathbf{x}} \log q_t(\mathbf{x})\right] dt + g(t)d\bar{\mathbf{w}}_t, \tag{1}$$

where $\bar{\mathbf{w}}_t$ denotes a standard wiener process with backward time flows.

In many practical scenarios, diffusion model sampling needs an extra guidance term $\mathbf{v}(\mathbf{x}, \mathbf{c}, t)$ to generate high-quality samples which modifies reverse diffusion process as follows:

$$d\mathbf{x} = \left[\mathbf{f}(\mathbf{x}) - g(t)^2 \left(\nabla_{\mathbf{x}} \log q_t(\mathbf{x}) + \mathbf{v}(\mathbf{x}, \mathbf{c}, t)\right)\right] dt + g(t)d\bar{\mathbf{w}}_t, \tag{2}$$

One notable approach is training-free guidance (Chung et al., 2023) where,

$$\mathbf{v}(\mathbf{x_t}, \mathbf{c}, t) = \nabla_{\mathbf{x}_t} \log p(\mathbf{c}|\hat{\mathbf{x}}_0), \tag{3}$$

and $\hat{\mathbf{x}}_0$ is a target estimate approximated with Tweedie's formula (Efron, 2011) as follows:

$$\hat{\mathbf{x}}_0 = \frac{\mathbf{x}_t + (1 - \bar{\alpha}_t)\nabla_{\mathbf{x}_t} \log p(\mathbf{x}_t)}{\sqrt{\bar{\alpha}_t}}. \tag{4}$$

Here, $\bar{\alpha}_t$ is a function determined by the forward process (Appendix B.3 for further details). Although training-free guidance can approximate sampling from conditional distribution only with unconditional model (Chung et al., 2023; Ye et al., 2024), this extra guidance in each timestep make samples far from the original learned data manifold (Shen et al., 2024).

**Multi-conditional guidance**  Downstream applications with diffusion models often required fine-grained control such as multi-conditional guidance (Du et al., 2023) or constrained guidance (Schramowski et al., 2023), where linear combination of more than two score functions are used to satisfy target properties. However, as stated in (Du et al., 2023), naive combination of two independent conditional score functions does not equal to multi-conditional score function:

$$\nabla_{\mathbf{x}} \log p(\mathbf{x}|\mathbf{c}_1, \mathbf{c}_2) \neq \nabla_{\mathbf{x}} \log p(\mathbf{x}|\mathbf{c}_1) + \nabla_{\mathbf{x}} \log p(\mathbf{x}|\mathbf{c}_2). \tag{5}$$

**Few-step generation**  The probability flow ODE formulation of diffusion models (Song et al., 2021b) accelerates generation by reducing the number of function evaluation (NFE) for sampling. However, discretization errors accumulate during the reverse process, resulting in off-manifold problem. We provide further details in Appendix B.5.

**Degradation of sample quality in low-density regions**  The score function $\nabla \log p_t(\mathbf{x}_t)$ of the diffusion model is trained to guide samples toward high density regions of the noisy data distribution $p_t(\mathbf{x}_t)$ at each timestep t. Ideally, in a perfectly learned diffusion process, this ensures generated outputs remain close to the original data manifold, resulting in high-fidelity samples. However, if an external force $\mathbf{v}$ drives a sample to the low density region $p_t(\mathbf{x}_t) \approx 0$, the score function $\nabla \log p_t(\mathbf{x}_t)$ estimated by the diffusion model becomes unreliable, as it is trained on noisy data that assumes the forward process is intact at the given timestep. This, often known as a score approximation error (Oko et al., 2023; Chen et al., 2023a), accumulates over time as generation process goes on, causing compounding errors that degrade sample quality in the subsequent steps of the generation process (Li & van der Schaar, 2024).

To illustrate how off-manifold phenomenon can become detrimental in diffusion sampling process, we construct a toy example of two Gaussian mixtures where external drift term is added in every timestep of the reverse process (details in Appendix E.1). Figure 2a shows that applying this external drift term in every diffusion step results in samples far from the original distribution.

## 3   METHOD

In this section, we introduce Temporal Alignment Guidance (TAG), a novel framework designed to maintain sample fidelity during diffusion model generation by mitigating off-manifold deviations at each timestep. We first formally define TAG, introducing the core concept of the Time-Linked Score (TLS) (Sec. 3.1). Subsequently, we detail how TAG integrates with practical guidance techniques to enhance conditional generation (Sec. 3.2). Finally, we provide a theoretical analysis on how TAG improves sample quality in the presence of off-manifold phenomenon (Sec. 3.3) along with illustrative example (Sec. 3.4).

### 3.1   TEMPORAL ALIGNMENT GUIDANCE (TAG)

**Projecting samples back to the On-Manifold**  We reinterpret timestep information as a conditioning variable rather than a fixed input in the reverse diffusion process. Fixed times scheduling suffices when samples remain on the original reverse path, it breaks down off-manifold because $\mathbf{x}_t$ loses its temporal identity. To project $\mathbf{x}_t$ back onto the correct manifold $\mathcal{M}_t$ (formal definition in Appendix B.4), we introduce the gradient term $\nabla_{\mathbf{x}} \log p_t(t \mid \mathbf{x})$, analogous to the conditional score in classifier guidance (Dhariwal & Nichol, 2021). Figure 2b illustrates that this vector field directs samples toward high-probability regions of the original distribution $q_t$, whereas the conventional diffusion score struggles once off-manifold. Incorporating this term into each reverse step thus keeps generated samples aligned with the data distribution (Figure 2b). In the next subsection, we formally define and analyze this new gradient correction.

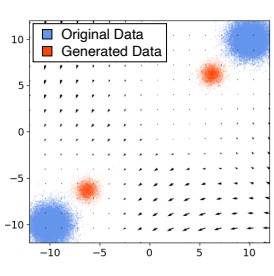 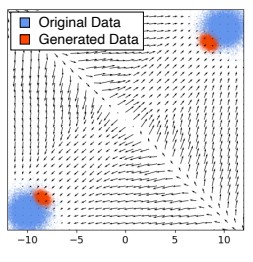

**(a)** Original model score

**(b)** Time score

**Figure 2:** Generated samples with score field. (Left) Generated outputs from reverse diffusion process with external drift, with vector field of the diffusion model output at $t = 0$. (Right) Generated outputs when applying TAG with external drift, with vector field of the TLS at $t = 0$.

---

**Algorithm 1** **T**emporal **A**lignment **G**uidance (**TAG**)

---

**Input:** Diffusion model $\boldsymbol{\theta}$, time predictor $\boldsymbol{\phi}$, guidance strength schedule $\omega_t$, number of total diffusion steps $T$
$\boldsymbol{x}_T \sim \mathcal{N}(\mathbf{0}, \boldsymbol{I})$
**for** $t = T, \cdots, 1$ **do**
 $\tilde{\mathbf{x}}_t \leftarrow \mathbf{x}_t + \omega_t \cdot \nabla \log p_\phi(t \mid \boldsymbol{x}_t)$
 Obtain $\nabla \log p(\mathbf{x})$ from a diffusion model $\boldsymbol{\theta}$
 $\mathbf{x}_{t-1} \leftarrow \tilde{\mathbf{x}}_t$ from reverse diffusion step following Eq. 1.
**end for**
**Output:** $\boldsymbol{x}_0$

---

**Time-Linked Score (TLS)** To further investigate the effect of this gradient term, we introduce the following definition:

**Definition 3.1.** Time-Linked Score for data point $\mathbf{x}$ and target time $t$ is defined as,

$$\text{TLS}(\mathbf{x}, t) := \nabla_{\mathbf{x}} \log p(t \mid \mathbf{x}). \tag{6}$$

Combining TLS with original score function of diffusion models, we now define Temporal Alignment Guidance:

**Definition 3.2.** The *Temporal Alignment Guidance (TAG)* at time $t$ is defined as

$$\text{TAG}(\mathbf{x}, t) = \nabla_{\mathbf{x}} \log p_t(\mathbf{x}) + \omega \cdot \nabla_{\boldsymbol{x}} \log p_\phi(t \mid \mathbf{x}). \tag{7}$$

where $\omega$ is a hyperparameter that controls the strength.

Applying TAG in the reverse diffusion provides a shortcut for a sample to the original manifold by sending it to the tilted probability $p(\mathbf{x}|t)p(t|\mathbf{x})^\omega$, just as in the classifier guidance Dhariwal & Nichol (2021). We provide a pseudo-code of sampling with TAG in Algorithm 1.

**Time classification by time predictor** Accurately identifying the correct manifold for each time is analytically impossible due to the complexity of the score function of real-world dataset Zhang & Chen (2023); Han et al. (2024b). Instead, we utilize a time predictor Jung et al. (2024), which is an auxiliary neural network trained with one-hot embeddings of timestep labels with following objective function:

$$\mathcal{L}_{\text{tp}}(\boldsymbol{\phi}) = -\mathbb{E}_{t, \mathbf{x}_0} \left[ \log \left( \hat{\mathbf{p}}_\phi(\mathbf{x}_t)_t \right) \right], \tag{8}$$

where $\hat{\mathbf{p}}$ denotes a logit vector of the model output. Time predictor learns to classify which timestep a random data with forward process should belong to. By calculating gradient of the time predictor, we can estimate TLS in Eq. 6. We use the simple cnn architecture that is substantially lightweight compared to the diffusion backbone. Details of the designing mechanism and performance of time predictor is in Appendix E.4.

### 3.2 IMPROVING GUIDANCE WITH TAG

We now present how TAG can be combined with a standard zero-shot conditional sampling framework like training-free guidance (TFG) (Chung et al., 2023; Ye et al., 2024) to improve conditional generation of diffusion models.

Let $\mathbf{c} \in \mathcal{Y}$ be the target property and let $\mathcal{A} : \mathcal{X} \to \mathcal{Y}$ be a off the shelf function that maps $\mathbf{x}_0 \in \mathcal{X}$ to their predicted property values. Training-free guidance is applied as,

$$\nabla_{\mathbf{x}_t} \log p_t(\mathbf{c}|\mathbf{x}_t) = \nabla_{\mathbf{x}_t} \log \mathbb{E}_{p(\mathbf{x}_0|\mathbf{x}_t)} \left[ \exp \left( -\ell_{\mathbf{c}}(\mathcal{A}(\hat{\mathbf{x}}_0), \mathbf{c}) \right] \tag{9}$$

where $\ell_c : \mathcal{Y} \times \mathcal{Y} \to \mathbb{R}$ measures the discrepancy between the estimated property and target property, and $\hat{\mathbf{x}}_0$ is the denoised estimate from Eq. 4.

One can obtain TLS with similar approach by observing

$$p(t \mid \mathbf{x}_t, \mathbf{c}) \propto \exp\big(-\ell_t(\phi(\mathbf{x}_t, \mathbf{c}), t)\big), \tag{10}$$

where $\ell_t$ is a penalty function for misalignment in time, and we set as a cross-entropy loss.

With the extended view of adding time information as another condition, we use Bayes' rule to the conditional probability as:

$$p_t(\mathbf{x}_t \mid \mathbf{c}) \propto p_t(\mathbf{x}_t) \, p(\mathbf{c} \mid \mathbf{x}_t) \, p(t \mid \mathbf{x}_t, \mathbf{c}). \tag{11}$$

Taking gradient respect to $\mathbf{x}_t$ for both sides, one can obtain conditional score function as follows:

$$\nabla_{\mathbf{x}_t} \log p(\mathbf{x}_t|\mathbf{c}) \approx \nabla_{\mathbf{x}_t} \log p(\mathbf{x}_t) + \sigma_t \nabla_{\mathbf{x}_t} \ell_{\mathbf{c}}(\mathcal{A}(\hat{\mathbf{x}}_0), \mathbf{c}) + \omega_t \nabla_{\mathbf{x}_t} \ell_t(\phi(\mathbf{x}_t, \mathbf{c}), t).$$

In essence, by treating time as an additional conditioning signal, TAG act as an on-manifold anchor at every reverse step: it pulls samples back onto the learned diffusion path, preventing off-manifold drift and markedly improving fidelity under arbitrary guidance.

### 3.3 Theoretical Analysis of TAG

Here, we provide the theoretical justification of TAG. We rigorously show that TAG can effectively reduce the error bound between the distribution of generated samples and the target distribution.

To start with, we first present the following theorem which states that TLS is a linear combination of the score functions of different timesteps in the following way:

**Theorem 3.3.** *Assuming discrete diffusion timesteps* $[t_1, t_2, \ldots, t_n]$, *Time-linked Score of a random noisy sample* $\mathbf{x}$ *to the target time* $t_i$ *can be represented as:*

$$\nabla_{\mathbf{x}} \log p\big(t_i \mid \mathbf{x}\big) = \sum_{k \neq i} \underbrace{\frac{p_{t_k}(\mathbf{x})}{p_{\text{tot}}(\mathbf{x})}}_{\substack{\text{greater when} \\ \text{off } t_i\text{-manifold}}} \Big( \underbrace{\nabla_{\mathbf{x}} \log p_{t_i}(\mathbf{x})}_{\text{pull to } t_i \text{ manifold}} - \underbrace{\nabla_{\mathbf{x}} \log p_{t_k}(\mathbf{x})}_{\text{repel other manifolds}} \Big). \tag{12}$$

Here, $p_{t_i}$'s are marginal distributions at each timestep and $p_{tot} = \sum_j p_{t_j}(\mathbf{x})$. The proof of Theorem 3.3 is in Appendix C.4.

Theorem 3.3 implies that TLS is particularly effective when $p_{t_i}(\mathbf{x}) \ll p_{\text{tot}}(\mathbf{x})$. In this regime, $\nabla_{\mathbf{x}} \log p_{t_i}(\mathbf{x})$ attracts the sample toward original data manifold, while simultaneously repelling it from competing manifolds through the negative terms $-\nabla_{\mathbf{x}} \log p_{t_k}(\mathbf{x})$ for $k \neq i$. Moreover, if $p_{t_j}(\mathbf{x})$ dominates for some $j \neq i$, the repulsive force $\nabla_{\mathbf{x}} \log p_{t_j}(\mathbf{x})$ in equation 12 grows, aiding the sample to escape an incorrect manifold. The above result can be naturally extend to continuous time (Appendix C.5).

Intuitively, at time $t$, score approximation errors tend to be larger in low-density regions of $p_t(\mathbf{x})$, since the model rarely encounters such regions during training. Consequently, corrector sampling (Song et al., 2021b) may become ineffective there, as the neural network's score estimates degrade. Moreover, even an accurate score estimate can struggle to guide samples out of inherently flat probability landscapes. Indeed, our empirical findings in Appendix D.2 show that corrector sampling becomes ineffective, sometimes degrade the sample quality under external guidance. Applying TAG can mitigate the aforementioned problems by increasing the chance of escape in this low density region. This can be formalized into the following proposition.

**Proposition 3.4.** *Applying TAG alters energy barrier map* $U_k(\mathbf{x}) = -\log p_{t_k}(\mathbf{x})$ *at timestep* $t_k$ *to* $\Phi_k(\mathbf{x})$ *for any* $k$ *by:*

$$\Phi_k(\mathbf{x}) = U_k(\mathbf{x}) - \sum_i \gamma_i \, U_i(\mathbf{x}), \tag{13}$$

*where* $\gamma_i = \frac{p_i(\mathbf{x})}{p_{tot}(\mathbf{x})}$ *for* $i \neq k$ *and* $\gamma_k = 1 - \sum_{i \neq k} \frac{p_i(\mathbf{x})}{p_{tot}(\mathbf{x})}$.

We defer the proof to Appendix C.6. Under mild assumptions, it shows that TAG sharpens the potential map via the negative repulsion of alternative timestep manifolds. Building on the Jordan–Kinderlehrer–Otto (JKO) scheme (Jordan et al., 1998), one can show that the modified Langevin

**Table 1:** Effect of TAG across strength $\omega$ of TAG when reverse process is corrupted with noise level $\sigma$.

| $\omega$ | $\sigma = 0.1$ | | | $\sigma = 0.2$ | | | $\sigma = 0.3$ | | |
|---|---|---|---|---|---|---|---|---|---|
| | TG$\downarrow$ | FID$\downarrow$ | IS$\uparrow$ | TG$\downarrow$ | FID$\downarrow$ | IS$\uparrow$ | TG$\downarrow$ | FID$\downarrow$ | IS$\uparrow$ |
| 0.0 | 104.1 | 193.6 | 2.37 | 229.6 | 351.4 | 1.50 | 274.0 | 410.1 | 1.28 |
| 0.5 | 47.9 | 127.7 | 3.65 | 200.6 | 340.1 | 1.56 | 261.6 | 408.5 | 1.28 |
| 1.0 | 41.8 | **120.9** | **3.69** | 175.5 | 323.7 | 1.61 | 250.9 | 406.7 | 1.28 |
| 2.0 | **39.0** | 132.6 | 3.33 | 140.2 | 285.1 | 1.64 | 232.6 | 390.3 | 1.27 |
| 4.0 | 44.4 | 159.8 | 3.00 | **103.4** | **246.3** | **1.70** | **197.8** | **361.2** | **1.31** |

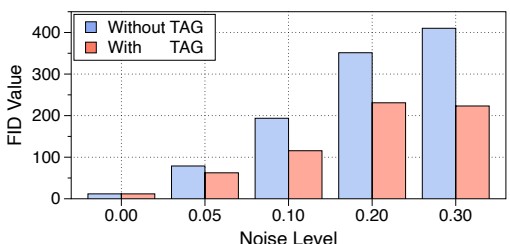

**Figure 3:** FID values over different corruption levels for original diffusion process without TAG and with TAG.

dynamics under this sharpened potential map accelerates correction with stronger gradient flows. In particular, applying a single reverse diffusion step with TAG increases the chance of a sample to move towards higher-density regions, thereby reducing expected score approximation errors. Building on prior analyses of diffusion models (Oko et al., 2023; Chen et al., 2023b), we show that TAG can improve the convergence guarantee by lowering the upper bound on the total variation distance $d_{TV}$ between the sample distribution and the target distribution:

**Theorem 3.5.** *(Informal) Let $p_t$ and $\tilde{p}_t$ be the probability distribution at time $t$ in the original reverse process in equation 1 and in the reverse process apply with TAG (Algorithm 1). Then, under mild assumptions, the upper bound of $d_{TV}(q_{data}, \tilde{p}_0)$ can be reduced compared to $d_{TV}(q_{data}, p_0)$.*

Theorem 3.5 demonstrates TAG's ability to enhance sample quality, a finding that aligns with our experimental observations. We provide a formal statement of Theorem 3.5 with corresponding proof in Appendix C.7.

### 3.4 UNDERSTANDING TAG UNDER CORRUPTED REVERSE PROCESS

To analyze TAG's corrective mechanism and evaluate its effectiveness under extreme perturbation, we conduct an experiment where artificial noise is applied at every reverse step. To quantify the temporal deviation during generation, we define the *Time-Gap* metric. Denoting the sample at timestep $t$ as $\mathbf{x}_t$ and the time predictor as $\phi$, the *Time-Gap* is defined as $\frac{1}{T}\sum_{t=1}^{T}\big|\arg\max\phi(\mathbf{x}_t) - t\big|$. A lower *Time-Gap* indicates that samples remain closer to their expected temporal manifold and correlates with improved generation quality (see Appendix F.1 for a formal definition and empirical validation).

Table 1 shows the effect of applying TAG under various noise levels ($\sigma$) and guidance strengths ($\omega$). As $\omega$ increases, both FID and IS improve, while the Time-Gap decreases, indicating that samples are drawn closer to the correct manifold. Figure 3 further illustrates that TAG significantly alleviates the degradation caused by increasing $\sigma$. These findings empirically confirm that the TLS component indeed corrects deviations and steers samples back to the appropriate temporal manifold, even under extreme perturbations. Further details of the experiments with additional results are in Appendix E.2.

## 4 EXPERIMENTS

We evaluate TAG empirically across diverse scenarios including those prone to off-manifold errors and practical applications mentioned in Sec. 3. First, we show that TAG improves standard TFG benchmarks via extensive comparisons with related methods (Sec. 4.1). Next, we extend to multi-conditional guidance, demonstrating efficient conditioning on multiple attributes without combinatorial overhead (Sec. 4.2). Then, we assess its ability to mitigate errors in few-step generation (Sec. 4.3). Finally, we demonstrate its applicability and benefits in large-scale text-to-image generation tasks (Sec. 4.4).

### 4.1 TFG BENCHMARK

**Setup** We follow the setup of TFG benchmark (Ye et al., 2024), a standard zero-shot conditional sampling framework, applying TAG to DPS (Chung et al., 2023) and TFG with their reported optimal hyperparameters. This offers a challenging comparison, since these carefully tuned baselines should exhibit less off-manifold drift than simpler methods. Experiments use 6 pretrained

Table 2: Quantitative results of TAG on TFG benchmark. Each cell presents the guidance validity / generation fidelity averaged across multiple targets in the task. The best result for each cell is reported in **bold**.

| Method | Deblur | | Super-resolution | | CIFAR10 | | ImageNet | | Audio declipping | | Audio inpainting | |
|---|---|---|---|---|---|---|---|---|---|---|---|---|
| | FID↓ | LPIPS↓ | FID↓ | LPIPS↓ | FID↓ | Acc.↑ | FID↓ | Acc.↑ | FAD↓ | DTW↓ | FAD↓ | DTW↓ |
| DPS (Chung et al., 2023) | 139.7 | 0.613 | 139.0 | 0.614 | 217.1 | 57.5 | 196.9 | 24.5 | 2.41 | 191 | 2.26 | 176 |
| DPS + TAG (ours) | **128.9** | **0.570** | **128.3** | **0.572** | **190.4** | **63.2** | **192.2** | 22.9 | **2.33** | **189** | **2.25** | **157** |
| Rel. Improvement | 7.7% | 7.0% | 7.7% | 6.8% | 12.3% | 9.9% | 2.4% | -6.5% | 3.3% | 1.0% | 0.4% | 10.8% |
| TFG (Ye et al., 2024) | 64.2 | 0.154 | 65.5 | 0.187 | 114.1 | 55.8 | 231.0 | 14.3 | 1.42 | 256 | 0.52 | 74 |
| TFG + TAG (ours) | **62.7** | **0.151** | **64.7** | **0.175** | **102.7** | **61.5** | **219.4** | **17.8** | **0.74** | **120** | **0.42** | **51** |
| Rel. Improvement | 2.3% | 1.9% | 1.2% | 6.4% | 10.0% | 10.2% | 5.0% | 24.5% | 47.9% | 53.1% | 19.3% | 31.1% |
| *Baseline Results* | | | | | | | | | | | | |
| TCS (Jung et al., 2024) | 454.7 | 0.751 | 465.1 | 0.748 | 213.4 | 29.4 | 344.9 | 12.0 | 23.89 | 567 | 21.41 | 558 |
| Timestep Guidance (Sadat et al., 2024) | 480.3 | 0.995 | 480.3 | 0.995 | 393.2 | 11.3 | 545.7 | 25.0 | 46.22 | 492 | 45.94 | 491 |
| Self-Guidance (Li et al., 2024b) | 231.8 | 0.709 | 231.0 | 0.710 | 205.4 | 51.6 | 257.4 | 10.8 | 8.90 | 521 | 6.99 | 463 |

| Method | Polarizability $\alpha$ | | Dipole $\mu$ | | Heat capacity $C_v$ | | $\epsilon_{HOMO}$ | | $\epsilon_{LUMO}$ | | Gap $\epsilon_\Delta$ | |
|---|---|---|---|---|---|---|---|---|---|---|---|---|
| | MAE↓ | Stab.↑ | MAE↓ | Stab.↑ | MAE↓ | Stab.↑ | MAE↓ | Stab.↑ | MAE↓ | Stab.↑ | MAE↓ | Stab.↑ |
| DPS (Chung et al., 2023) | 13.33 | 28.4 | 4779.92 | 34.4 | 3.47 | 36.2 | 0.68 | 30.3 | 1.57 | 17.6 | 1.65 | 10.6 |
| DPS + TAG (ours) | **7.96** | **96.4** | **1.48** | **97.2** | **3.03** | **93.0** | **0.58** | **56.2** | **1.11** | **48.4** | **1.29** | **93.5** |
| Rel. Improvement | 40.3% | 239.7% | 99.9% | 182.5% | 13.1% | 157.0% | 6.1% | 85.7% | 29.6% | 174.5% | 21.4% | 779.2% |
| TFG (Ye et al., 2024) | 8.91 | 19.2 | 2.41 | 26.3 | 2.65 | 96.2 | 0.55 | 14.6 | 1.33 | 10.8 | 1.40 | 16.1 |
| TFG + TAG (ours) | **4.46** | **43.6** | **1.28** | **94.3** | 2.67 | **96.7** | **0.43** | **93.9** | **0.89** | **92.5** | **0.78** | **82.8** |
| Rel. Improvement | 49.9% | 127.1% | 46.9% | 258.6% | 0.3% | 0.5% | 21.8% | 543.8% | 33.1% | 757.4% | 44.3% | 414.2% |
| *Baseline Results* | | | | | | | | | | | | |
| TCS (Jung et al., 2024) | 11.44 | 15.3 | 1.60 | 6.3 | 3.17 | 19.6 | 0.59 | 50.2 | 1.23 | 28.8 | 1.58 | 13.9 |
| Timestep Guidance (Sadat et al., 2024) | 25.07 | 70.2 | N/A | N/A | 4.18 | 82.9 | N/A | N/A | N/A | N/A | 1.39 | 48.7 |
| Self-Guidance (Li et al., 2024b) | 16.33 | 65.3 | 62.86 | 70.9 | 3.89 | 79.7 | N/A | N/A | 2.32 | 10.8 | 1.30 | 24.9 |

models—CIFAR10-DDPM (Nichol & Dhariwal, 2021), ImageNet-DDPM (Dhariwal & Nichol, 2021), Cat-DDPM (Elson et al., 2007), CelebA-DDPM (Karras et al., 2018), Molecule-EDM (Hoogeboom et al., 2022), and Audio-Diffusion (Kong et al., 2021; Popov et al., 2021). The tasks include image restoration (deblurring, super-resolution), conditional generation (label-guided sampling, multi-attribute generation), molecular generation (molecular property control), and audio synthesis (clipping, inpainting). For all tasks, we report generation fidelity and validity, with further details provided in Appendix E.3.

**External guidance scenario** We evaluate TAG in a single-conditional guidance setting, where the objective is to sample from the target distribution $p(\mathbf{x}_0 | \mathbf{c})$ with DPS Chung et al. (2023) and TFG (Ye et al., 2024). We set the guidance schedule of TAG as $\omega_t = \omega_0 \sqrt{(1 - \bar{\alpha}_t)}$. The final results are averaged over the best-performing guidance strength $w_0$ according to the grid search for all target values in each task.

The results in Table 2 demonstrate that TAG significantly improves the fidelity while maintaining conditioning effect across most tasks. We observe that TAG is particularly effective when the adversarial effect of external guidance becomes larger (i.e, when training free guidance guidance degrades sample fidelity). To confirm this, we compare TAG against several recent approaches applied on top of DPS, including TCS (Jung et al., 2024), Timestep Guidance (Sadat et al., 2024), Self-Guidance (Li et al., 2024b), and exposure-bias methods (Ning et al., 2024; Li et al., 2024a; Ning et al., 2023). The result confirms that while these baselines degrade under external guidance drift, TAG remains robust (see further details in Appendix D).

To further highlight this effectiveness, we conduct additional experiments by increasing the DPS strength from 1.0 to 5.0. Table 3 shows that TAG effectively mitigates the negative influence of stronger guidance strength, while applying only DPS results in generating mostly non-valid samples. In contrast, applying TAG with DPS show robust performance across all evaluation metrics. Qualitative results are in Appendix G.

### 4.2 MULTI-CONDITIONAL GUIDANCE

We next evaluate TAG in multi-conditional settings, where naively combining multiple guidance terms can induce severe off-manifold errors. Extending to multiple conditions is nontrivial, as naive approaches demand combinatorial training or multiple specialized time predictors, motivating a more efficient approach.

**Table 3:** Quantitative result of TAG for different values of DPS guidance strength. (DPS / DPS + TAG)

| | | CIFAR10 | | ImageNet | | Polar. $\alpha$ | | Heat cap. $C_v$ | |
|---|---|---|---|---|---|---|---|---|---|
| Str. | TAG | FID↓ | Acc↑ | FID↓ | Acc↑ | MAE↓ | Stab↑ | MAE↓ | Stab↑ |
| 1.0 | ✗ | 217.1 | 57.5 | 196.9 | **24.5** | 103.7 | 1.1 | 13.7 | 1.9 |
| | ✓ | **190.4** | **63.2** | **192.2** | 22.9 | **48.5** | **32.2** | **9.9** | **5.4** |
| 1.5 | ✗ | 269.5 | 51.4 | 219.3 | 27.0 | 109.8 | 0.9 | 16.2 | 2.1 |
| | ✓ | **231.9** | **62.3** | **204.1** | **32.7** | **50.3** | **31.8** | **11.1** | **14.0** |
| 2.5 | ✗ | 334.1 | 41.9 | 230.2 | 28.5 | 159.5 | 1.0 | 18.4 | 2.9 |
| | ✓ | **289.7** | **51.9** | **212.7** | **30.2** | **49.9** | **31.2** | **12.2** | **9.7** |
| 5.0 | ✗ | 384.8 | 29.4 | 246.7 | 24.3 | 112.7 | 1.1 | N/A | N/A |
| | ✓ | **347.8** | **41.0** | **233.1** | **27.2** | **51.7** | **30.4** | **14.7** | **8.0** |

**Table 4:** Quantitative evaluation of FID for few-step using DDPM sampling without external guidance.

| Dataset | TAG | Inference Steps | | | | | |
|---|---|---|---|---|---|---|---|
| | | 1 Step | 3 Step | 5 Step | 10 Step | 50 Step | 100 Step |
| CIFAR10 | ✗ | 460.0 | 234.1 | 158.6 | 106.3 | 71.8 | 67.6 |
| | ✓ | **271.1** | **160.5** | **118.8** | **93.1** | **70.9** | **66.5** |
| ImageNet | ✗ | 430.3 | 297.6 | 295.2 | 286.7 | 259.6 | 251.1 |
| | ✓ | **352.8** | **265.1** | **265.0** | **265.1** | **245.7** | **244.7** |
| Cat | ✗ | 433.7 | 313.5 | 243.9 | 209.9 | 166.4 | 154.9 |
| | ✓ | **314.8** | **178.8** | **199.5** | **188.1** | **164.2** | **152.2** |

Table 5: Quantitative results of TAG in Multi-Conditional generation on TFG benchmark. Each cell presents the guidance validity/generation fidelity averaged across multiple targets in the task. The best result for each cell is reported in **bold**.

| | | CelebA | | | | Molecule | | | | | | | | | |
|---|---|---|---|---|---|---|---|---|---|---|---|---|---|---|---|
| | | Gender + Age | | Gender + Hair | | $\alpha, \mu$ | | $C_v, \mu$ | | $\alpha, \mu, C_v, \epsilon_\Delta, \epsilon_{\text{HOMO}}, \epsilon_{\text{LUMO}}$ | | | | | |
| Method | TAG | KID↓ | Acc↑ | KID↓ | Acc↑ | MAE↓ | Stab↑ | MAE↓ | Stab↑ | MAE↓ | | | | | Stab↑ |
| Baseline | ✗ | -2.75 | 80.5 | -3.16 | 92.1 | 13.7 | 1782.8 | 68.9 | 4.97 | 1425.2 | 70.9 | 10.1 | 31.9 | 4.33 0.635 1.14 1.18 | 56.0 |
| Multi. | ✓ | -2.85 | 87.1 | -3.19 | 94.9 | **4.56** | **1.31** | 84.7 | 2.72 | **1.33** | **84.2** | 4.52 1.45 2.94 0.610 1.13 1.15 | | | **91.2** |
| Single. | ✓ | -2.86 | **91.0** | **-3.27** | **96.1** | 4.65 | 1.33 | 83.9 | **2.63** | 1.40 | 82.9 | 4.58 **1.39** 3.05 0.577 **1.05 1.11** | | | 85.9 |
| Uncon. | ✓ | **-2.87** | 89.1 | -3.08 | 96.0 | **4.56** | 1.35 | **84.9** | 2.74 | 1.36 | **84.2** | **4.48** 1.44 **2.82 0.530** 1.07 1.15 | | | 85.9 |

**Multi-condition reparametrization** For multiple conditions $\mathbf{c}_i \in \mathcal{Y}$ with corresponding predictors $\mathcal{A}_i$ and losses $\ell_i$, we write

$$p_t(\mathbf{x}_t \mid \mathbf{c}_1, \mathbf{c}_2) \propto p_t(\mathbf{x}_t)\, p(\mathbf{c}_1 \mid \mathbf{x}_t)\, p(\mathbf{c}_2 \mid \mathbf{x}_t, \mathbf{c}_1)\, p(t \mid \mathbf{x}_t, \mathbf{c}_1, \mathbf{c}_2). \quad (14)$$

Although a multi-condition time predictor $\phi(\mathbf{x}_t, \mathbf{c}_1, \mathbf{c}_2)$ is possible, it is often impractical; instead, via *single-condition reparameterization*, we approximate $p(t \mid \mathbf{x}_t, \mathbf{c}_1, \mathbf{c}_2) \approx p(t \mid \mathbf{x}'_t, \mathbf{c}_2)$ by

$$\mathbf{x}'_t \approx \mathbf{x}_t - \eta_t^2 \, \nabla_{\mathbf{x}_t} \ell_1(\mathcal{A}_1(\hat{\mathbf{x}}_0), \mathbf{c}_1), \quad (15)$$

) where $\hat{\mathbf{x}}_0 = \mathbb{E}[\mathbf{x}_0 \mid \mathbf{x}_t]$. A detailed derivation is provided in Proposition B.1. For an *unconditional* time predictor, we iteratively incorporate each condition:

$$\mathbb{E}[\mathbf{x}'_t \mid \mathbf{x}_t, \mathbf{c}_1, \mathbf{c}_2] \approx \mathbf{x}_t - \eta_t^2 \nabla_{\mathbf{x}_t} \ell_1(\mathcal{A}_1(\mathbf{x}_t), \mathbf{c}_1) - \eta_t^2 \nabla_{\mathbf{x}_t} \ell_2(\mathcal{A}_2(\mathbf{x}''_t), \mathbf{c}_2), \quad (16)$$

where $\mathbf{x}''_t$ reflects $\mathbf{c}_1$, leading to $p(t \mid \mathbf{x}_t, \mathbf{c}_1, \mathbf{c}_2) \approx p(t \mid \mathbf{x}'_t)$ and naturally extending to more conditions while remaining efficient. The formal details are provided in Proposition B.2.

**Setup** We consider molecule-generation tasks with (i) $\alpha, \mu$, (ii) $C_v, \mu$, and (iii) all six molecular properties $(\alpha, \mu, C_v, \epsilon_{\text{HOMO}}, \epsilon_{\text{LUMO}}, \epsilon_\Delta)$, along with CelebA (Gender+Age, Gender+Hair). We follow the TFG framework (Ye et al., 2024) to combine these conditions and compare three time-predictor variants—multi, single, and unconditional as introduced in Sec. 3.2. (Refer to Appendix E.3 for setting details).

**Result** As shown in Table 5, TAG significantly outperforms the baseline combination of independent guidance for all tasks. Notably, single and unconditional time predictors match or exceed multi-conditional performance, indicating that explicit training of a multi-conditional time predictor is not strictly necessary, and TAG can achieve effective multi-conditional guidance.

### 4.3 FEW-STEP GENERATION

We evaluate TAG in widely-used accelerated sampling, where diffusion models skip timesteps to reduce computation but risk larger discretization errors. We compare a standard DDIM sampler (Song et al., 2021a) with TAG for various step counts. As shown in Table 4, TAG consistently boosts sample quality, particularly under fewer steps. Notably, in an extreme single-step scenario on CIFAR10

(Table 17), TAG lowers FID by 41.1%. This aligns with our theoretical analysis indicating stronger negative guidance helps the sample escape incorrect manifolds. While one can analytically reduce discretization error (Karras et al., 2022), our focus is on treating it as external noise and demonstrating how TAG mitigates off-manifold drift in practice (see Appendix B.5 for further discussion).

## 4.4 LARGE-SCALE TEXT-TO-IMAGE GENERATION

We further evaluate TAG on large-scale text-to-image generations by integrating it into models based on Stable Diffusion v1.5 (Rombach et al., 2022), demonstrating its effectiveness on more practical generative tasks. Further details of the experimental setup are provided in Appendix E.5.

**Enhanced Reward Alignment** We integrate TAG into DAS (Kim et al., 2025)—a state-of-the-art test-time sampler that optimizes text-to-image generation under explicit reward functions (e.g., Aesthetic score (Schuhmann et al., 2022) or CLIP score (Radford et al., 2021)). First, we follow Kim et al. (2025) to evaluate reward alignment using simple animal prompts and an Aesthetic target score. Next, we switch to a CLIP-based reward and the HPSv2 prompt set (Wu et al., 2023). Finally, we evaluate a multi-objective scenario where the target reward is a linear combination of the Aesthetic and CLIP scores with HPSv2 prompt dataset. In each setting, we compare the original DAS sampler against DAS enhanced with TAG (DAS+TAG) on 256 randomly selected prompts.

As shown in Table 6, adding TAG substantially increases the final reward while reducing the average Time-Gap (Def. F.1) which measures off-manifold deviation, confirming TAG's stabilization capability in practical, large-scale alignment scenario.

Table 6: TAG enhances reward alignment with signle objective DAS, multi-objective DAS and Style Transfer on SD v1.5. Higher reward scores and lower Time-Gap (TG) are better.

| Method | Single-objective DAS | | | | Multi-objective DAS | | | Method | Style Transfer | |
| --- | --- | --- | --- | --- | --- | --- | --- | --- | --- | --- |
| | Aesthetic ↑ | TG ↓ | CLIP ↑ | TG ↓ | Aesthetic↑ | CLIP ↑ | TG ↓ | | Style Score ↓ | TG ↓ |
| DAS (Kim et al., 2025) | 7.948 | 90.04 | 0.389 | 20.73 | 8.107 | 0.439 | 20.73 | TFG (Ye et al., 2024) | 4.82 | 80.6 |
| DAS + TAG | **9.087** | **28.84** | **0.439** | **11.62** | **8.572** | **0.463** | **9.765** | TFG + TAG | **3.03** | **23.6** |

**Improved Style Transfer** We also apply TAG to style transfer task building on TFG (Ye et al., 2024). Specifically, we combine text prompts (Partiprompts (Yu et al., 2022)) and reference style images (WikiArt (Yu et al., 2022)) via a CLIP-based (Radford et al., 2021) Gram matrix alignment. Table 6 compares TFG alone with TFG+TAG, reporting Style Score and Time-Gap. Integrating TAG yields a sizable drop in Style Score and substantially reduces the Time-Gap, indicating more faithful style adherence and fewer off-manifold deviations.

## 4.5 ABLATION STUDY

We also probe how the time predictor's training steps influence off-manifold correction, exploring the effect of different guidance strengths under added noise, verifying that TAG's gains persist when scaling to 50k samples, and analyzing how the Time-Gap metric correlates with standard image quality scores. Detailed analyses of predictor robustness, hyperparameter sensitivity, and additional baseline comparisons are in Appendix E–F.

## 5 CONCLUSION AND FUTURE WORKS

In this work, we identify when off-manifold phenomenon happen in diffusion models by measuring Time-Gap using a time prediction mechanism. To reduce a time gap, we introduce Temporal Alignment Guidance (TAG) as a novel guidance mechanism to force the samples to desired manifold in each timestep. Our experimental results demonstrates TAG can significantly reduce this off-manifold phenomenon in multiple scenarios which shows the robustness of our method. We believe our method could be especially effective when applied to real-world downstream tasks where desired condition can vary in real-time. For future work, it would be promising to investigate the effect of TAG in another domains such as in reinforcement learning tasks Janner et al. (2022), discrete diffusion models Austin et al. (2021); Chen et al. (2023c).

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

TABLE OF CONTENTS

## A   BROADER IMPACT AND LIMITATIONS

**Broader impact**   Our algorithm improves the sample quality of diffusion models. While beneficial for applications like image generation or drug discovery, this also carries the risk of misuse common to generative models, potentially enabling harmful generation of images (e.g., disinformation), molecules (e.g., unsafe compounds), or audio. Developing stronger safeguard mechanisms within generative systems, including diffusion models, is essential to counteract such potential negative societal impacts.

**Limitations**   In our experiments, we noticed that once sample fidelity reaches a high level, further narrowing the time-gap yields only marginal or no improvements in quality. Although our existing time-predictor training procedure is sufficient to demonstrate TAG's practical benefits (see Section 4), we anticipate that more sophisticated predictor architectures could unlock additional gains. We leave this exploration to future work.

**Usage of Large Language Models**   We utilized a large language model to aid in polishing the writing and improving the clarity of this manuscript. The model's role was strictly limited to assistance with grammar, phrasing, and style. All scientific ideas, methodologies, experimental results, and conclusions presented in this paper are the original work of the authors.

## B   FURTHER BACKGROUND

In this section, we introduce more background of the key concepts used in this work.

### B.1   DIFFUSION MODELS

**Diffusion Models**   Diffusion models are generative models that sample from the data distribution, denoted as $\mathbf{x}_0 \sim q_{data}$. Following the stochastic differential equation (SDE) framework (Song et al., 2021a), the forward diffusion process can be defined by the following SDE:

$$\mathrm{d}\mathbf{x} = \mathbf{f}(\mathbf{x}, t)dt + g(t)d\mathbf{w}_t, \tag{17}$$

where $\mathbf{w_t}$ is a standard wiener process (Øksendal, 2003). Ideally, if we denote $q_t(\mathbf{x})$ as the marginal distribution of the forward process in equation 17, it becomes close to $\mathcal{N} \sim (0, \mathbf{I})$ when $t$ goes to large enough $T$.

Then, diffusion model $\boldsymbol{\theta}$ is trained to learn how to denoise a noisy data by learning a score function which is done by minimizing the following objective function (Song & Ermon, 2019; Song et al., 2021b):

$$\mathcal{L}(\boldsymbol{\theta}) = \mathbb{E}_{t,\mathbf{x}_0}\lambda(t)\|\boldsymbol{s_\theta}(\mathbf{x}_t, t) - \nabla_{\mathbf{x}_t} \log q_t(\mathbf{x}_t|\mathbf{x}_0)\|_2^2, \tag{18}$$

where $t$ is uniformly sampled from $[0, T]$, $\mathbf{x}_t$ denotes $\mathbf{x}$ at timestep $t$ in equation 17, and $\lambda(t)$ is a weight parameter usually set to be a constant Ho et al. (2020).

**Conditional Diffusion Model** The aim of conditional diffusion models is to sample from the conditional posterior $p_0(\mathbf{x}|\mathbf{c})$ with given condition $\mathbf{c}$. This is achieved by learning a conditional score function $\nabla_\mathbf{x} \log q_t(\mathbf{x}|\mathbf{c})$. Using Bayes' rule the conditional score can be re-expressed as:

$$\nabla_\mathbf{x} \log q_t(\mathbf{x}|\mathbf{c}) = \nabla_\mathbf{x} \log q_t(\mathbf{x}) + \nabla_\mathbf{x} \log q_t(\mathbf{c}|\mathbf{x}). \tag{19}$$

One could obtain $\nabla_\mathbf{x} \log q_t(\mathbf{c}|\mathbf{x})$ with auxiliary classifier (Dhariwal & Nichol, 2021) (classifier guidance), or train with condition-labeled data (Ho & Salimans, 2021) (classifier-free guidance)

### B.2 SCORE BASED DIFFUSION MODEL

Here, we systematically present different forms of forward and reverse diffusion model processes and their types in the existing literature.

**Denoising score matching** Learning score function $\nabla_\mathbf{x} \log p(\mathbf{x})$ perfectly for all $\mathbf{x}$ can ideally guide the sample towards high density region Hyvärinen & Dayan (2005). However, Song & Ermon (2019) suggests that neural network struggles to accurately model low density region. One alternative is use denoising score matchng (Vincent, 2011; Song & Ermon, 2019) where a neural network instead models a score function of perturbed dataset $\nabla_\mathbf{x} \log p_t(\mathbf{x}_t)$ where $\mathbf{x}_t \sim \mathcal{N}(\mathbf{x}, \sigma(t)^2 \mathbf{I})$.

**SDE framework** Song et al. (2021b) define the forward and reverse process of diffusion model by the following form of stochastic differential equation (SDE).

$$\mathrm{d}\mathbf{x} = \mathbf{f}(\mathbf{x}, t)dt + g(t)d\mathbf{w}_t, \tag{20}$$

where $\mathbf{w_t}$ is a standard wiener process. Two types of SDE is widely used in current diffusion models, one is variance preserving SDE (VP-SDE) which has a following form:

$$\mathrm{d}\mathbf{x} = \sqrt{\sigma(t)\sigma'(t)}\, d\mathbf{w}_t, \tag{21}$$

where $\sigma(t)$ is noise schedule as in Song & Ermon (2019). The other is variance exploding SDE (VE-SDE) which has a following form:

$$\mathrm{d}\mathbf{x}_t = -\frac{1}{2}\beta(t)\mathbf{x}\, dt + \sqrt{\beta(t)}\, d\mathbf{w}_t, \tag{22}$$

where $\beta(t)$ is another noise schedule.

**ODE framework** Reverse process of SDE in Eq. 1 has its corresponding ODE with same marginal probability density which is called probability flow ODE Song et al. (2021b):

$$\mathrm{d}\mathbf{x} = \left[\mathbf{f}(\mathbf{x}) - \frac{1}{2}g^2(t)\nabla_\mathbf{x} \log q_t(\mathbf{x})\right] dt. \tag{23}$$

A discretized version of the PF-ODE sampler can be interpreted as DDIM sampling Song et al. (2021a). This ODE formulation can be leveraged to skip network evaluation, enabling faster inference time of diffusion models (Lu et al., 2022; Song et al., 2023).

**Connection to DDPM** Here we offer the relationship between different frameworks for convenience. Song et al. (2021b) unified denoising score matching with DDPM Ho et al. (2020) by viewing forward process of DDPM as a discretized version of VP-SDE in Eq. 21. In DDPM Ho et al. (2020), forward noise schedule is defined by $\mathbf{x}_t = \sqrt{\bar{\alpha}_t}\mathbf{x}_0 + \sqrt{1 - \bar{\alpha}_t}\boldsymbol{\epsilon}$ where $\boldsymbol{\epsilon}$ is a random noise from $\mathcal{N}(0, \mathbf{I})$. This is a discretized version of VP-SDE in Eq. 21 Song et al. (2021b), where notations have following relations:

$$\bar{\alpha}_t = \exp(-\frac{1}{2} \int_0^t \beta(s)\,ds). \tag{24}$$

In DDPM, model output is denoted as $\boldsymbol{\epsilon}_\theta(\mathbf{x}, t)$ which has following relationship with a score function $\nabla_{\mathbf{x}_t} \log p_t(\mathbf{x}_t)$:

$$\nabla_{\mathbf{x}_t} \log p_t(\mathbf{x}_t) = -\frac{1}{\sqrt{1 - \bar{\alpha}_t}}\boldsymbol{\epsilon}_\theta(\mathbf{x_t}). \tag{25}$$

Unless otherwise stated, this work utilizes a VP-SDE diffusion process with DDIM sampling.

### B.3 TRAINING-FREE GUIDANCE

Training free guidance leverages clean estimates $\mathbf{x}_0$ during the reverse process. Specifically, Tweedie's formula Efron (2011) is used to estimate original data during the reverse diffusion process. For VE-SDE, this can be represented as:

$$\hat{\mathbf{x}}_0 := \mathbb{E}[\mathbf{x}_0 | \mathbf{x}_t] = \frac{\mathbf{x}_t + (1 - \bar{\alpha}_t)\nabla_{\mathbf{x}_t} \log p(\mathbf{x}_t)}{\sqrt{\bar{\alpha}_t}}. \tag{26}$$

where $\bar{\alpha}_t = e^{-\frac{1}{2} \int_0^t \beta(s)ds}$ by Eq. 24. And for VE-SDE in Eq. 22, estimation of $\hat{\mathbf{x}}_0$ can be represented as

$$\hat{\mathbf{x}}_0 := \mathbb{E}[\mathbf{x}_0 | \mathbf{x}_t] = \mathbf{x}_t + \sigma^2(t)\nabla_{\mathbf{x}_t} \log p_t(\mathbf{x}_t). \tag{27}$$

$\hat{\mathbf{x}}_0$, conditional probability for the target condition $\mathbf{c}$ can be obtained as

$$p(\mathbf{c} \,|\, \hat{\mathbf{x}}_0) \propto \exp\big(-\ell_{\mathbf{c}}(\mathcal{A}(\hat{\mathbf{x}}_0), \mathbf{c})\big), \tag{28}$$

where $\mathcal{A}$ denotes a classifier or an analytic function that outputs a condition given the clean estimate $\hat{\mathbf{x}}_0$ and $\ell_c : \mathcal{Y} \times \mathcal{Y} \to \mathbb{R}$ measures the discrepancy between the estimated property and the target property which is usually heuristically chosen function. Now conditional score function in Eq. 19 can be approximated by

$$\begin{aligned} \nabla_{\mathbf{x}_t} \log p_t(\mathbf{c}|\mathbf{x}_t) &= \nabla_{\mathbf{x}_t} \log \mathbb{E}_{p(\mathbf{x}_0|\mathbf{x}_t)} \big[\exp\big(-\ell_{\mathbf{c}}(\mathcal{A}(\hat{\mathbf{x}}_0))\big] \\ &\approx \nabla_{\mathbf{x}_t}\hat{\mathbf{x}}_0 \cdot \nabla_{\hat{\mathbf{x}}_0}(-\ell_{\mathbf{c}}(\mathcal{A}(\hat{\mathbf{x}}_0)), \end{aligned} \tag{29}$$

where we use chain-rule and the Tweedie's formula.

**Extended view by TAG** One can view applying TAG with Training Free Guidance as an extended framework.

Denote $\phi : \mathcal{X} \times \mathcal{Y} \to [0, T]$ as a time predictor mapping noisy samples $x_t \in \mathcal{X}$ and conditions $\mathbf{c}$ to plausible time indices $t \in [0, T]$. The corresponding likelihood of having a correct time $t$ becomes,

$$p(t \,|\, \mathbf{x}_t, \mathbf{c}) \propto \exp\big(-\ell_t(\phi(\mathbf{x}_t, \mathbf{c}), t)\big), \tag{30}$$

where $\ell_t : \mathbb{R} \times [0, T] \to \mathbb{R}$ is a loss function that quantifies the difference between estimated time and the desired time.

With the extended view of adding time information as another condition, we can approximate the conditional distribution $p_t(\mathbf{x}_t \,|\, \mathbf{c})$ as:

$$p_t(\mathbf{x}_t \,|\, \mathbf{c}) \propto p_t(\mathbf{x}_t)\, p(\mathbf{c} \,|\, \mathbf{x}_t)\, p(t \,|\, \mathbf{x}_t, \mathbf{c}), \tag{31}$$

where $p_t(\mathbf{x}_t)$ is from the pre-trained unconditional diffusion model. However, we only have access to $p(\mathbf{c} \mid \mathbf{x}_0)$ and $p(t \mid \mathbf{x}_t, \mathbf{c})$. To bridge $\mathbf{x}_0$ and $\mathbf{x}_t$, we replace $\mathbf{x}_0$ with its denoised estimate $\hat{\mathbf{x}}_0 = \mathbb{E}[\mathbf{x}_0 \mid \mathbf{x}_t]$. This gives:

$$p(\mathbf{c} \mid \mathbf{x}_t) \propto \exp\big(-\ell_{\mathbf{c}}(\mathcal{A}(\hat{\mathbf{x}}_0), \mathbf{c})\big). \tag{32}$$

To further align $\mathbf{x}_t$ to the temporal manifold, we reparameterize $\mathbf{x}_t$ as $\mathbf{x}_t' \approx \mathbf{x}_t - \eta_t \nabla_{\mathbf{x}_t} \ell_c(\mathcal{A}(\hat{\mathbf{x}}_0), \mathbf{c})$ and write,

$$p(t \mid \mathbf{x}_t, \mathbf{c}) \propto \exp\big(-\ell_t(\phi(\mathbf{x}_t', \mathbf{c}), t)\big). \tag{33}$$

Consequently, the approximated conditional distribution becomes,

$$p_t(\mathbf{x}_t \mid \mathbf{c}) \propto p_t(\mathbf{x}_t) \exp\big(-\ell_{\mathbf{c}}(\mathcal{A}(\hat{\mathbf{x}}_0), \mathbf{c})\big) \exp\big(-\ell_t(\phi(\mathbf{x}_t', \mathbf{c}), t)\big). \tag{34}$$

If $\boldsymbol{\epsilon}_\theta(\mathbf{x}_t, t) \approx -\sigma_t \nabla_{\mathbf{x}_t} \log p_t(\mathbf{x}_t)$ represents the unconditioned diffusion score, the new guided score for single-condition TAG is given by,

$$\tilde{\boldsymbol{\epsilon}}_\theta(\mathbf{x}_t, \mathbf{c}, t) = \boldsymbol{\epsilon}_\theta(\mathbf{x}_t, t) - \sigma_t \nabla_{\mathbf{x}_t} \ell_{\mathbf{c}}(\mathcal{A}(\hat{\mathbf{x}}_0), \mathbf{c})$$
$$- \sigma_t \nabla_{\mathbf{x}_t} \ell_t(\phi(\mathbf{x}_t', \mathbf{c}), t). \tag{35}$$

In practice, one updates $\mathbf{x}_t \to \mathbf{x}_t'$ before applying $\ell_t$, ensuring that each sampling step remains aligned with both the property $\mathbf{c}$ and the correct time $t$, mitigating off-manifold drifting.

**Muti-conditional TAG** Let $\mathbf{c}_1 \in \mathcal{Y}_1, \mathbf{c}_2 \in \mathcal{Y}_2$ be the target property value, and let $\mathcal{A}_1, \mathcal{A}_2 : \mathcal{X} \to \mathcal{Y}$ be property classifiers that map samples $\mathbf{x}_0 \in \mathcal{X}$ to their respective predicted property values. To sample from the conditional distribution $p_t(\mathbf{x}_t \mid \mathbf{c}_1, \mathbf{c}_2)$, we factorize,

$$p_t(\mathbf{x}_t \mid \mathbf{c}_1, \mathbf{c}_2) \propto p_t(\mathbf{x}_t) p(\mathbf{c}_1 \mid \mathbf{x}_t) p(\mathbf{c}_2 \mid \mathbf{x}_t, \mathbf{c}_1) p(t \mid \mathbf{x}_t, \mathbf{c}_1, \mathbf{c}_2), \tag{36}$$

where $p(t \mid \mathbf{x}_t, \mathbf{c}_1, \mathbf{c}_2)$ ensures ensures alignment of $\mathbf{x}_t$ to the temporal manifold under $\mathbf{c}_1$ and $\mathbf{c}_2$.

A straightforward method is to directly model $p(t \mid \mathbf{x}_t, \mathbf{c}_1, \mathbf{c}_2)$ via a multi-condition time predictor $\phi(\mathbf{x}_t, \mathbf{c}_1, \mathbf{c}_2)$:

$$p(t \mid \mathbf{x}_t, \mathbf{c}_1, \mathbf{c}_2) \propto \exp\big(-\ell_t(\phi(\mathbf{x}_t, \mathbf{c}_1, \mathbf{c}_2), t)\big). \tag{37}$$

While this method fully accounts for multi-condition effects, it requires training a separate model for every condition combination, which becomes infeasible for complex or high-dimensional conditions.

To address this challenge, we employ a single-condition time predictor $\phi(\mathbf{x}_t, \mathbf{c})$ that models $p(t \mid \mathbf{x}_t, \mathbf{c})$ for a single condition $\mathbf{c}$. In this case, we approximate $p(t \mid \mathbf{x}_t, \mathbf{c}_1, \mathbf{c}_2)$ by re-parameterizing $\mathbf{x}_t$ to reflect $\mathbf{c}_1$.

**Proposition B.1.** *Let $\mathbf{x}_t'$ be a latent variable conditioned on $\mathbf{x}_t$ and target property $\mathbf{c}_1$, with prior distribution $p(\mathbf{x}_t \mid \mathbf{x}_t', \mathbf{c}_1) \sim \mathcal{N}(\mathbf{x}_t', \eta_t^2 \mathbf{I})$. Given a first-order approximation of the property likelihood:*

$$p(\mathbf{c}_1 \mid \mathbf{x}_t') \propto \exp\left(-\ell_1(\mathcal{A}_1(\mathbf{x}_t), \mathbf{c}_1) - (\mathbf{x}_t' - \mathbf{x}_t)^\top \nabla_{\mathbf{x}_t} \ell_1(\mathcal{A}_1(\mathbf{x}_t), \mathbf{c}_1)\right), \tag{38}$$

*the posterior expectation of $\mathbf{x}_t'$ under $p(\mathbf{x}_t' \mid \mathbf{x}_t, \mathbf{c}_1)$ satisfies:*

$$\mathbb{E}_{\mathbf{x}_t' \sim p(\mathbf{x}_t' \mid \mathbf{x}_t, \mathbf{c}_1)}[\mathbf{x}_t'] = \mathbf{x}_t - \eta_t^2 \nabla_{\mathbf{x}_t} \ell_1(\mathcal{A}_1(\mathbf{x}_t), \mathbf{c}_1). \tag{39}$$

*Proof.* See Appendix C.2 $\qquad\qquad\qquad\qquad\qquad\qquad\qquad\qquad\qquad\qquad\qquad\qquad\qquad$ $\square$

Practically, Using Tweedie's formula Efron (2011); Chung et al. (2023), we replace $\mathcal{A}_1(\mathbf{x}_t)$ with $\mathcal{A}_1(\hat{\mathbf{x}}_0)$, where $\hat{\mathbf{x}}_0 = \mathbb{E}[\mathbf{x}_0 \mid \mathbf{x}_t]$ is the denoised estimate. Thus we have an approximation:

$$\mathbf{x}_t' \approx \mathbf{x}_t - \eta_t^2 \nabla_{\mathbf{x}_t} \ell_1(\mathcal{A}_1(\hat{\mathbf{x}}_0), \mathbf{c}_1). \tag{40}$$

As a result of Proposition B.1, the single-condition time predictor allows us to approximate $p(t \mid \mathbf{x}_t, \mathbf{c}_1, \mathbf{c}_2)$ by reparameterizing $\mathbf{x}_t$ to reflect the influence of $\mathbf{c}_1$, yielding,

$$p(t \mid \mathbf{x}_t, \mathbf{c}_1, \mathbf{c}_2) \approx p(t \mid \mathbf{x}_t', \mathbf{c}_2),$$

where $\mathbf{x}'_t = \mathbf{x}_t - \eta_t^2 \nabla_{\mathbf{x}_t} \ell_1(\mathcal{A}_1(\mathbf{x}_t), \mathbf{c}_1)$. This reparameterization ensures that $\mathbf{x}'_t$ partially aligns with $\mathbf{c}_1$, reducing the approximation error when conditioning on $\mathbf{c}_2$ (see Algorithms 1 for implementation).

We could further extend this framework to the case of an unconditional time predictor $\phi(\mathbf{x}_t)$, which models $p(t \mid \mathbf{x}_t)$ without explicit dependence on any condition. This extension significantly reduces the computational cost of training by requiring only a single predictor for all possible conditions, relying on additional approximations of $p(t \mid \mathbf{x}_t, \mathbf{c}_1, \mathbf{c}_2)$ to capture the influence of $\mathbf{c}_1$ and $\mathbf{c}_2$ within the unconditional framework.

**Proposition B.2.** *Let $\mathbf{x}'_t$ be a latent variable conditioned on $\mathbf{x}_t$ and target properties $\mathbf{c}_1, \mathbf{c}_2$, with priors:*

$$p(\mathbf{x}_t \mid \mathbf{x}'_t, \mathbf{c}_1, \mathbf{c}_2) \sim \mathcal{N}(\mathbf{x}'_t, \eta_t^2 \mathbf{I}),$$
$$p(\mathbf{x}'_t \mid \mathbf{x}''_t, \mathbf{c}_1) \sim \mathcal{N}(\mathbf{x}''_t, \tilde{\eta}_t^2 \mathbf{I}), \tag{41}$$

*where $\mathbf{x}''_t$ are intermediate samples reflecting $\mathbf{c}_1$ before updating $\mathbf{c}_2$. The posterior expectation satisfies:*

$$\mathbb{E}_{\mathbf{x}'_t \sim p(\mathbf{x}'_t \mid \mathbf{x}_t, \mathbf{c}_1, \mathbf{c}_2)}[\mathbf{x}'_t] = \mathbf{x}_t - \eta_t^2 \nabla \ell_1(\mathcal{A}_1(\mathbf{x}_t), \mathbf{c}_1) - \eta_t^2 \nabla \ell_2(\mathcal{A}_2(\mathbf{x}''_t), \mathbf{c}_2). \tag{42}$$

*Proof.* See Appendix C.3 $\qquad\square$

Again, in practical scenarios using Tweedie's formula Efron (2011); Chung et al. (2023), we replace $\mathcal{A}_1(\mathbf{x}_t)$ and $\mathcal{A}_2(\mathbf{x}'_t)$ with denoised estimates:

$$\nabla \ell_1(\mathcal{A}_1(\mathbf{x}_t), \mathbf{c}_1) \approx \nabla \ell_1(\mathcal{A}_1(\hat{\mathbf{x}}_0), \mathbf{c}_1),$$
$$\nabla \ell_2\left(\mathcal{A}_2\left(\mathbf{x}'_t - \tilde{\eta}_t^2 \nabla \ell_1(\mathcal{A}_1(\mathbf{x}_t), \mathbf{c}_1)\right), \mathbf{c}_2\right) \approx \nabla \ell_2(\mathcal{A}_2(\hat{\mathbf{x}}'_0), \mathbf{c}_2), \tag{43}$$

where $\hat{\mathbf{x}}'_0 = \mathbb{E}[\mathbf{x}_0 \mid \mathbf{x}_t - \tilde{\eta}_t^2 \nabla \ell_1(\mathcal{A}_1(\hat{\mathbf{x}}_0), \mathbf{c}_1)]$. Substituting these approximations gives:

$$\mathbf{x}'_t \approx \mathbf{x}_t - \eta_t^2 \nabla \ell_1(\mathcal{A}_1(\hat{\mathbf{x}}_0), \mathbf{c}_1) - \eta_t^2 \nabla \ell_2(\mathcal{A}_2(\hat{\mathbf{x}}'_0), \mathbf{c}_2). \tag{44}$$

The unconditional time predictor incorporates the influences of $\mathbf{c}_1$ and $\mathbf{c}_2$ by sequentially reparameterizing $\mathbf{x}_t$ through iterative updates. This approach leverages reparameterization steps that align $\mathbf{x}_t$ to the conditions $\mathbf{c}_1$ and $\mathbf{c}_2$, reducing the approximation gap to the true conditional distribution. The framework naturally extends to handle $k > 2$ conditions, iteratively integrating each condition while maintaining computational efficiency (see Algorithms 2 for implementation).

**Pseudo-Code**   We provide the pseudo-code for implementing multi-conditional guidance using a single-conditional (B.1) time predictor and an unconditional time predictor (B.2) in Alg. 1 and Alg. 2, respectively.

### B.4   MANIFOLD ASSUMPTION

Ideally, even if original data manifold $\mathcal{M}_0$ can be a low-dimensional object as pointed out in several works (Bortoli, 2022; He et al., 2024), with noise added from forward process in Eq. 17, $p_t(\mathbf{x}_t) > 0$ for all $\mathbf{x}_t \in \mathcal{X}$ where $\mathcal{X}$ denotes the data domain. Since our motivation of off-manifold phenomenon happens in low-density region, we redefine the target data manifold for each timestep by the following definition.

**Definition B.3.** Let $\epsilon_t > 0$ be some threshold. The correct manifold at timestep $t$ is defined as

$$\mathcal{M}_t = \{\mathbf{x} \in \mathcal{X} : p_t(\mathbf{x}) \geq \epsilon_t\}, \tag{45}$$

where $\mathcal{X}$ is domain of the data. In other words, $\mathcal{M}_t$ consists of all points in $\mathcal{X}$ whose probability density is at least $\epsilon_t$.

With above definition, we can formally define the off-manifold in diffusion models.

**Definition B.4.** For given timestep $t$ in reverse diffusion process in Eq. 1, we define off-manifold phenomenon by $\mathbf{x}_t$ becomes out of the correct manifold $\mathcal{M}_t$ defined in Definition B.3. In other words:

$$\mathbf{x}_t \notin \mathcal{M}_t. \tag{46}$$

We leave further theoretical understanding of off-manifold phenomenon from the above definition as a future work.

---

**Algorithm 1: DDIM Sampling with Single-Conditional Time Predictor**

**Input** : Unconditional score model $\nabla_{\mathbf{x}_t} \log p_t(\mathbf{x}_t)$, property classifier $\mathcal{A}_1 : \mathcal{X} \to \mathbb{R}$, loss function $\ell_1 : \mathbb{R} \times \mathbb{R} \to \mathbb{R}$, single-condition time predictor $\tau(\mathbf{x}_t, \mathbf{c})$, operator $\mathcal{G}$, target properties $\mathbf{c}_1, \mathbf{c}_2$, guidance strength $\rho_t$, temporal alignment strength $\omega_t$, time steps $T$.

**Output** : Conditional sample $\mathbf{x}_0$.

1 Initialize $\mathbf{x}_T \sim \mathcal{N}(0, I)$;

2 **for** $t = T, \ldots, 1$ **do**

3     Compute $\hat{\mathbf{x}}_0 \leftarrow \frac{\mathbf{x}_t + (1-\alpha_t)\nabla_{\mathbf{x}_t} \log p_t(\mathbf{x}_t)}{\sqrt{\alpha_t}}$;

4     Reparameterize $\mathbf{x}_t'$ to reflect $\mathbf{c}_1$: $\mathbf{x}_t' \leftarrow \mathbf{x}_t - \eta_t^2 \nabla \ell_1(\mathcal{A}_1(\hat{\mathbf{x}}_0), \mathbf{c}_1)$;

5     Compute temporal alignment term using $\tau(\mathbf{x}_t', \mathbf{c}_2)$: $\mathcal{T} \leftarrow -\nabla_{\mathbf{x}_t} \ell_t(\tau(\mathbf{x}_t', \mathbf{c}_2), t)$;

6     Define the generalized guidance operator $\mathcal{G}(\mathbf{x}_t, \mathbf{c}_1, \mathbf{c}_2)$ to compute joint or independent guidance contributions;

7     $\mathbf{x}_{t-1} \leftarrow \sqrt{\alpha_{t-1}} \left( \frac{\mathbf{x}_t - \sqrt{1-\alpha_t}\nabla_{\mathbf{x}_t} \log p_t(\mathbf{x}_t)}{\sqrt{\alpha_t}} \right) + \sqrt{1 - \alpha_{t-1} - \sigma_t^2} \cdot \nabla_{\mathbf{x}_t} \log p_t(\mathbf{x}_t) +$
    $\rho_t \mathcal{G}(\mathbf{x}_t, \mathbf{c}_1, \mathbf{c}_2) + \omega_t \mathcal{T} + \sigma_t \boldsymbol{\epsilon}_t.$

8 **return** $\mathbf{x}_0$;

---

---

**Algorithm 2: DDIM Sampling with Unconditional Time Predictor**

**Input** : Unconditional score model $\nabla_{\mathbf{x}_t} \log p_t(\mathbf{x}_t)$, property classifiers $A_1 : \mathcal{X} \to \mathbb{R}$, $A_2 : \mathcal{X} \to \mathbb{R}$, loss functions $\ell_1, \ell_2 : \mathbb{R} \times \mathbb{R} \to \mathbb{R}$, unconditional time predictor $\tau(\mathbf{x}_t)$, operator $\mathcal{G}$, target properties $\mathbf{c}_1, \mathbf{c}_2$, guidance strength $\rho_t$, temporal alignment strength $\omega_t$, time steps $T$.

**Output** : Conditional sample $\mathbf{x}_0$.

1 Initialize $\mathbf{x}_T \sim \mathcal{N}(0, I)$;

2 **for** $t = T, \ldots, 1$ **do**

3     Compute $\hat{\mathbf{x}}_0 \leftarrow \frac{\mathbf{x}_t + (1-\alpha_t)\nabla_{\mathbf{x}_t} \log p_t(\mathbf{x}_t)}{\sqrt{\alpha_t}}$;

4     Reparameterize $\mathbf{x}_t'$ to reflect $\mathbf{c}_1$: $\mathbf{x}_t' \leftarrow \mathbf{x}_t - \eta_t^2 \nabla \ell_1(A_1(\hat{\mathbf{x}}_0), \mathbf{c}_1)$;

5     Compute $\hat{\mathbf{x}}_0' \leftarrow \frac{\mathbf{x}_t' + (1-\alpha_t)\nabla_{\mathbf{x}_t'} \log p_t(\mathbf{x}_t')}{\sqrt{\alpha_t}}$;

6     Reparameterize $\mathbf{x}_t''$ to reflect $\mathbf{c}_2$: $\mathbf{x}_t'' \leftarrow \mathbf{x}_t' - \tilde{\eta}_t^2 \nabla \ell_2(A_2(\hat{\mathbf{x}}_0'), \mathbf{c}_2)$;

7     Compute temporal alignment term using $\tau(\mathbf{x}_t'')$: $\mathcal{T} \leftarrow -\nabla_{\mathbf{x}_t} \ell_t(\tau(\mathbf{x}_t''), t)$;

8     Define the generalized guidance operator $\mathcal{G}(\mathbf{x}_t, \mathbf{c}_1, \mathbf{c}_2)$ to compute joint or independent guidance contributions;

9     $\mathbf{x}_{t-1} \leftarrow \sqrt{\alpha_{t-1}} \left( \frac{\mathbf{x}_t - \sqrt{1-\alpha_t}\nabla_{\mathbf{x}_t} \log p_t(\mathbf{x}_t)}{\sqrt{\alpha_t}} \right) + \sqrt{1 - \alpha_{t-1} - \sigma_t^2} \cdot \nabla_{\mathbf{x}_t} \log p_t(\mathbf{x}_t) +$
    $\rho_t \mathcal{G}(\mathbf{x}_t, \mathbf{c}_1, \mathbf{c}_2) + \omega_t \mathcal{T} + \sigma_t \boldsymbol{\epsilon}_t.$

10 **return** $\mathbf{x}_0$;

---

### B.5 FEW STEP GENERATION

As shown in Lu et al. (2022), PF-ODE in Eq. 23 sends $\mathbf{x}_s$ at timestep $s$ to $\mathbf{x}_t$ at timestep $t$ by solving,

$$\mathbf{x}_t = e^{\int_s^t f(\tau) d\tau} \mathbf{x}_s + \int_s^t (e^{\int_\tau^t f(\tau) d\tau} \cdot \frac{g^2(\tau)}{2\sigma_\tau} \epsilon_\theta(\mathbf{x}_\tau, \tau)) d\tau. \tag{47}$$

Here, forward SDE is defined as follows.

$$d\mathbf{x}_t = f(t)\mathbf{x}_t \cdot dt + \frac{g^2(t)}{2\sigma_t} \epsilon_\theta(\mathbf{x}_t, t) \cdot dt, \quad \mathbf{x}_t \sim \mathcal{N}(0, \sigma_t^2 \boldsymbol{I}), \tag{48}$$

which incorporates both VP-SDE and VE-SDE scenarios (Appendix B.2) and $f(t), g(t)$ are defined as:

$$f(t) = \frac{d \log \alpha_t}{dt}, \quad g^2(t) = \frac{d\sigma_t^2}{dt} - 2\frac{d \log \alpha_t}{dt}\sigma_t^2. \tag{49}$$

After using change of variable $\lambda(t) := \log(\frac{\alpha_t}{\sigma_t})$, Lu et al. (2022) show following equation holds:

$$\mathbf{x}_t = \frac{\alpha_t}{\alpha_s}\mathbf{x}_s - \alpha_t \int_{\lambda_s}^{\lambda_t} e^{-\lambda}\hat{\epsilon}_\theta(\hat{\mathbf{x}}_\lambda, \lambda)d\lambda. \tag{50}$$

Now, from Eq. 50, one can observe how discretization error occurs if we skip the evaluation of the diffusion models for some of timesteps. Note that the discretization errors can be reduced by considering higher-order term in Eq. 50 (Karras et al., 2022; Lu et al., 2022; 2023) where we leave combining TAG with higher order diffusion solver as a future work.

Table 7: Mathematical terms and notations used in this work.

| Notation | Descriptions |
|---|---|
| $x_t$ | Sample of the diffusion process at timestep $t \in [0, T]$ |
| $q_t(x)$ | Marginal probability density of the forward diffusion process at time $t$ |
| $p_t(x)$ | Marginal probability density of the learned reverse process at time $t$ |
| $s_\theta(x, t)$ | Score network approximating the score function $\nabla_x \log q_t(x)$ |
| $\mathcal{M}_t$ | Target data manifold at time $t$, defined as the $\epsilon$-support $\{x \in \mathcal{X} : p_t(x) \geq \epsilon_t\}$ |
| $\phi(x)$ | Time predictor network estimating the posterior $p(t|x)$ |
| $\nabla_x \log p(t|x)$ | Time-Linked Score (TLS); the gradient field for temporal alignment |
| $\mathrm{TAG}(x, t)$ | Total score function modified by Temporal Alignment Guidance |
| $\omega_t$ | Time-dependent scalar controlling the strength of TAG |
| $v(x, c, t)$ | Arbitrary external guidance vector field (e.g., classifier gradient) |
| $\mathcal{A}(\cdot)$ | Pre-trained property predictor or classifier for condition $c$ |
| $\hat{x}_0$ | Tweedie's estimate of $x_0$ given noisy observation $x_t$ |
| $\mathbb{P}, \tilde{\mathbb{P}}$ | Path measures of the standard and guided reverse processes |
| $d_{TV}(\cdot, \cdot)$ | Total variation distance between two probability measures |
| $\tau$ | Stopping time indicating escape from the manifold $\mathcal{M}_t$ |

## C  MATHEMATICAL DERIVATIONS

### C.1  UPPER BOUND BY EXTERNAL DRIFT

To analyze the error induced by the random shift, we compare how the samples follow original reverse SDE in equation 1, and the modified SDE in equation 2 differs by the following proposition:

**Proposition C.1** (Error bound by the drift). *Let $p_t$ and $\tilde{p}_t$ be the probability distribution at time $t$ in the original reverse process in equation 1 and in the reverse process with external guidance $\mathbf{v}(\mathbf{x}, \mathbf{c}, t)$ in equation 2, respectively. The total variation distance $p_0$ and $\tilde{p}_0$ can be bounded as follows:*

$$d_{TV}^2(p_0, \tilde{p}_0) \leq KL(p_0, \tilde{p}_0) \leq \frac{1}{2} \int_0^T \int_{\mathbf{x}} g(t)^{-2} p_t(\mathbf{x}) \|\mathbf{v}(\mathbf{x}, \mathbf{c}, t)\|_2^2 \, d\mathbf{x} \, dt. \tag{51}$$

Proposition C.1 provides an upper bound indicates that external guidance $\mathbf{v}$ can induce distributional divergence in the worst case, even if the underlying score function for $p_t(\mathbf{x})$ is perfectly known.

**Proof of Proposition C.1**    For the ease of analysis, we first redefine the notations. Suppose $\mathbf{Y}_t$ and $\tilde{\mathbf{Y}}_t$ be the random variable of backward process of original reverse diffusion process by satisfying $\mathbf{Y}_{T-t} = \mathbf{x}_t$ in Eq. 1 and reverse process with external guidance by satisfying $\tilde{\mathbf{Y}}_{T-t} = \mathbf{x}_t$ in Eq. 2, respectively. This can be restated with following formulations:

$$d\mathbf{Y}_t = \left[ -\mathbf{f}(\mathbf{Y}_t, t) + g(t)^2 \nabla \log q_t(\mathbf{Y}_t) \right] dt + g(t) d\mathbf{w}_t, \ \mathbf{Y}_0 \sim \mathcal{N}(0, \mathbf{I})$$

$$d\tilde{\mathbf{Y}}_t = \left[ -\mathbf{f}(\tilde{\mathbf{Y}}_t, t) + g(t)^2 \left( \nabla \log q_t(\tilde{\mathbf{Y}}_t) + \mathbf{v}(\tilde{\mathbf{Y}}_t, \mathbf{c}, t) \right) \right] dt + g(t) d\bar{\mathbf{w}}_t, \ \tilde{\mathbf{Y}}_0 \sim \mathcal{N}(0, \mathbf{I}). \tag{52}$$

Also, denote $p_t$ and $\tilde{p}_t$ be probability distributions of $\mathbf{Y}_t$ and $\tilde{\mathbf{Y}}_t$, respectively and denote path measure of two process by $\mathbb{P}, \tilde{\mathbb{P}}$, respectively. Now, the goal is to bound the distance between $p_T$ and $\tilde{p}_T$ which are final output of two SDE processes. This can be proved by automatic consequence of Girsanov's Theorem (Karatzas & Shreve, 1991). To start, we first define the stochastic process

$$M_t = \exp\left( -\int_0^T \sigma(t)^{-1} \mathbf{v} \cdot d\mathbf{w}_t - \frac{1}{2} \int_0^T \int_{\mathbf{y}} \sigma(t)^{-2} \|\mathbf{v}\|^2 d\mathbf{y} \, dt \right) \tag{53}$$

and assume $M_t$ is a Martingale. Then, Girsanov's Theorem states that the Radon-Nikodym derivative of $\mathbb{P}$ with respect to $\tilde{\mathbb{P}}$ becomes

$$d\mathbb{P} = M_T d\tilde{\mathbb{P}}, \tag{54}$$

and this consequently bounds the KL divergence between two path measures as follows:

$$KL(\mathbb{P}, \tilde{\mathbb{P}}) = \frac{1}{2} \int_0^T \int_{\mathbf{y}} p_t(\mathbf{y}) \sigma(t)^{-2} \|\mathbf{v}\|^2 d\mathbf{y} dt. \tag{55}$$

Finally, using data processing inequality and Pinsker's inequality together (Cover, 1999), one can obtain:

$$d_{TV}^2(p_0, \tilde{p}_0) \le KL(p_0, \tilde{p}_0) \le KL(\mathbb{P}, \tilde{\mathbb{P}}) = \mathbb{E}_{\mathbb{P}} \left[ \frac{1}{2} \int_0^T \int_{\mathbf{y}} \sigma(t)^{-2} \|\mathbf{v}\|^2 d\mathbf{y}\, dt \right]. \tag{56}$$

It is known that following is a sufficient condition for for $M_t$ to be a Martingale (Novikov's condition):

$$\mathbb{E}_{\mathbb{P}} \left[ \exp \left( \frac{1}{2} \int_0^T \int_{\mathbf{y}} \sigma(t)^{-2} \|\mathbf{v}\|^2 d\mathbf{y}\, dt \right) \right] < \infty, \tag{57}$$

and this can be further relaxed by the following condition:

$$\int_{\mathbf{y}} p_t(\mathbf{y}) \sigma(t)^{-2} \|\mathbf{v}\|^2 d\mathbf{y} \le C \tag{58}$$

for all $t$ and some constant $C$ (Chen et al., 2023b). $\qquad\square$

Note that similar analysis has been conducted to prove the convergence rate of diffusion models in (Chen et al., 2023b; Oko et al., 2023) while their analysis does not contain any additional guidance.

### C.2 PROOF OF PROPOSITION B.1

**Proposition B.1** Let $\mathbf{x}_t'$ be a latent variable conditioned on $\mathbf{x}_t$ and target property $\mathbf{c}_1$, with prior distribution $p(\mathbf{x}_t \mid \mathbf{x}_t', \mathbf{c}_1) \sim \mathcal{N}(\mathbf{x}_t', \eta_t^2 \mathbf{I})$. Given a first-order approximation of the property likelihood:

$$p(\mathbf{c}_1 \mid \mathbf{x}_t') \propto \exp \left( -\ell_1(A_1(\mathbf{x}_t), \mathbf{c}_1) - (\mathbf{x}_t' - \mathbf{x}_t)^\top \nabla_{\mathbf{x}_t} \ell_1(A_1(\mathbf{x}_t), \mathbf{c}_1) \right), \tag{59}$$

the posterior expectation of $\mathbf{x}_t'$ under $p(\mathbf{x}_t' \mid \mathbf{x}_t, \mathbf{c}_1)$ satisfies:

$$\mathbb{E}_{\mathbf{x}_t' \sim p(\mathbf{x}_t' \mid \mathbf{x}_t, \mathbf{c}_1)}[\mathbf{x}_t'] = \mathbf{x}_t - \eta_t^2 \nabla_{\mathbf{x}_t} \ell_1(A_1(\mathbf{x}_t), \mathbf{c}_1). \tag{60}$$

*Proof.* Similar to Han et al. (2024a), which assumes a prior on the clean sample estimate given a latent variable and applies a first-order expansion of the loss function, we assume a prior on $\mathbf{x}_t$ at each $t$. We model the temporal distribution $p(t \mid \mathbf{x}_t, \mathbf{c}_1, \mathbf{c}_2)$ via a property loss function, whereas Han et al. (2024a) models $p(\mathbf{c}_2 \mid \hat{\mathbf{x}}_0, \mathbf{c}_1)$, with $\hat{\mathbf{x}}_0$ as the clean estimate.

The posterior distribution is derived via Bayes' rule:

$$p(\mathbf{x}_t' \mid \mathbf{x}_t, \mathbf{c}_1) \propto p(\mathbf{x}_t \mid \mathbf{x}_t', \mathbf{c}_1) p(\mathbf{c}_1 \mid \mathbf{x}_t') p(\mathbf{x}_t'). \tag{61}$$

Assuming a flat prior $p(\mathbf{x}_t') \propto 1$, the posterior simplifies to:

$$p(\mathbf{x}_t' \mid \mathbf{x}_t, \mathbf{c}_1) \propto p(\mathbf{x}_t \mid \mathbf{x}_t', \mathbf{c}_1) p(\mathbf{c}_1 \mid \mathbf{x}_t'). \tag{62}$$

The Gaussian prior is given by:

$$p(\mathbf{x}_t \mid \mathbf{x}_t', \mathbf{c}_1) \propto \exp \left( -\frac{\|\mathbf{x}_t - \mathbf{x}_t'\|^2}{2\eta_t^2} \right). \tag{63}$$

The likelihood $p(\mathbf{c}_1 \mid \mathbf{x}_t')$ is approximated using a first-order Taylor expansion of $\ell_1(A_1(\mathbf{x}_t'), \mathbf{c}_1)$ around $\mathbf{x}_t$:

$$\ell_1(A_1(\mathbf{x}_t'), \mathbf{c}_1) \approx \ell_1(A_1(\mathbf{x}_t), \mathbf{c}_1) + (\mathbf{x}_t' - \mathbf{x}_t)^\top \nabla_{\mathbf{x}_t} \ell_1(A_1(\mathbf{x}_t), \mathbf{c}_1). \tag{64}$$

Thus, the likelihood becomes:

$$p(\mathbf{c}_1 \mid \mathbf{x}'_t) \propto \exp\left(-\ell_1(A_1(\mathbf{x}_t), \mathbf{c}_1) - (\mathbf{x}'_t - \mathbf{x}_t)^\top \nabla_{\mathbf{x}_t} \ell_1(A_1(\mathbf{x}_t), \mathbf{c}_1)\right). \tag{65}$$

Combining the prior and likelihood, the log-posterior is:

$$\log p(\mathbf{x}'_t \mid \mathbf{x}_t, \mathbf{c}_1) \propto -\frac{\|\mathbf{x}_t - \mathbf{x}'_t\|^2}{2\eta_t^2} - \ell_1(A_1(\mathbf{x}_t), \mathbf{c}_1) - (\mathbf{x}'_t - \mathbf{x}_t)^\top \nabla_{\mathbf{x}_t} \ell_1(A_1(\mathbf{x}_t), \mathbf{c}_1). \tag{66}$$

Differentiating the log-posterior with respect to $\mathbf{x}'_t$ yields:

$$\frac{\partial}{\partial \mathbf{x}'_t} \log p(\mathbf{x}'_t \mid \mathbf{x}_t, \mathbf{c}_1) = -\frac{\mathbf{x}'_t - \mathbf{x}_t}{\eta_t^2} - \nabla_{\mathbf{x}_t} \ell_1(A_1(\mathbf{x}_t), \mathbf{c}_1). \tag{67}$$

Setting the gradient to zero for the MAP estimate gives:

$$\mathbf{x}'_t = \mathbf{x}_t - \eta_t^2 \nabla_{\mathbf{x}_t} \ell_1(A_1(\mathbf{x}_t), \mathbf{c}_1). \tag{68}$$

For Gaussian posteriors, the MAP estimate coincides with the expectation. $\qquad\square$

### C.3 PROOF OF PROPOSITION B.2

**Proposition B.2** Let $\mathbf{x}'_t$ be a latent variable conditioned on $\mathbf{x}_t$ and target properties $\mathbf{c}_1, \mathbf{c}_2$, with priors:

$$\begin{aligned} p(\mathbf{x}_t \mid \mathbf{x}'_t, \mathbf{c}_1, \mathbf{c}_2) &\sim \mathcal{N}(\mathbf{x}'_t, \eta_t^2 \mathbf{I}), \\ p(\mathbf{x}'_t \mid \mathbf{x}''_t, \mathbf{c}_1) &\sim \mathcal{N}(\mathbf{x}''_t, \tilde{\eta}_t^2 \mathbf{I}), \end{aligned} \tag{69}$$

where $\mathbf{x}''_t$ are intermediate samples reflecting $\mathbf{c}_1$ before updating $\mathbf{c}_2$. The posterior expectation satisfies:

$$\mathbb{E}_{\mathbf{x}'_t \sim p(\mathbf{x}'_t \mid \mathbf{x}_t, \mathbf{c}_1, \mathbf{c}_2)}[\mathbf{x}'_t] = \mathbf{x}_t - \eta_t^2 \nabla \ell_1(A_1(\mathbf{x}_t), \mathbf{c}_1) - \eta_t^2 \nabla \ell_2(A_2(\mathbf{x}''_t), \mathbf{c}_2). \tag{70}$$

*Proof.* The posterior distribution is derived via hierarchical Bayesian inference:

$$p(\mathbf{x}'_t \mid \mathbf{x}_t, \mathbf{c}_1, \mathbf{c}_2) \propto p(\mathbf{x}_t \mid \mathbf{x}'_t, \mathbf{c}_1, \mathbf{c}_2) p(\mathbf{c}_1, \mathbf{c}_2 \mid \mathbf{x}'_t) p(\mathbf{x}'_t). \tag{71}$$

Assuming flat priors $p(\mathbf{x}'_t) \propto 1$ and $p(\mathbf{x}''_t) \propto 1$, the model simplifies to:

$$p(\mathbf{x}'_t \mid \mathbf{x}_t, \mathbf{c}_1, \mathbf{c}_2) \propto p(\mathbf{x}_t \mid \mathbf{x}'_t, \mathbf{c}_1, \mathbf{c}_2) p(\mathbf{c}_1 \mid \mathbf{x}'_t) p(\mathbf{c}_2 \mid \mathbf{x}'_t, \mathbf{c}_1). \tag{72}$$

The Gaussian prior for $p(\mathbf{x}_t \mid \mathbf{x}'_t, \mathbf{c}_1, \mathbf{c}_2)$ is:

$$p(\mathbf{x}_t \mid \mathbf{x}'_t, \mathbf{c}_1, \mathbf{c}_2) \propto \exp\left(-\frac{\|\mathbf{x}_t - \mathbf{x}'_t\|^2}{2\eta_t^2}\right). \tag{73}$$

The likelihood for $\mathbf{c}_1$ is approximated using a first-order Taylor expansion of $\ell_1(A_1(\mathbf{x}'_t), \mathbf{c}_1)$ around $\mathbf{x}_t$:

$$\ell_1(A_1(\mathbf{x}'_t), \mathbf{c}_1) \approx \ell_1(A_1(\mathbf{x}_t), \mathbf{c}_1) + (\mathbf{x}'_t - \mathbf{x}_t)^\top \nabla \ell_1(A_1(\mathbf{x}_t), \mathbf{c}_1). \tag{74}$$

Thus, the likelihood becomes:

$$p(\mathbf{c}_1 \mid \mathbf{x}'_t) \propto \exp\left(-\ell_1(A_1(\mathbf{x}_t), \mathbf{c}_1) - (\mathbf{x}'_t - \mathbf{x}_t)^\top \nabla \ell_1(A_1(\mathbf{x}_t), \mathbf{c}_1)\right). \tag{75}$$

For $p(\mathbf{c}_2 \mid \mathbf{x}'_t, \mathbf{c}_1)$, we introduce an intermediate latent variable $\mathbf{x}''_t$ conditioned on $\mathbf{x}'_t$ and $\mathbf{c}_1$:

$$p(\mathbf{x}'_t \mid \mathbf{x}''_t, \mathbf{c}_1) \propto \exp\left(-\frac{\|\mathbf{x}'_t - \mathbf{x}''_t\|^2}{2\tilde{\eta}_t^2}\right). \tag{76}$$

The likelihood for $\mathbf{c}_2$ is approximated using a first-order Taylor expansion of $\ell_2(A_2(\mathbf{x}''_t), \mathbf{c}_2)$ around $\mathbf{x}'_t$:

$$\ell_2(A_2(\mathbf{x}''_t), \mathbf{c}_2) \approx \ell_2(A_2(\mathbf{x}'_t), \mathbf{c}_2) + (\mathbf{x}''_t - \mathbf{x}'_t)^\top \nabla \ell_2(A_2(\mathbf{x}'_t), \mathbf{c}_2). \tag{77}$$

Substituting $\mathbf{x}''_t = \mathbf{x}'_t - \tilde{\eta}_t^2 \nabla \ell_1(A_1(\mathbf{x}_t), \mathbf{c}_1)$ (from Proposition C.2), the likelihood becomes:

$$p(\mathbf{c}_2 \mid \mathbf{x}'_t, \mathbf{c}_1) \propto \exp\left(-\ell_2\left(A_2\left(\mathbf{x}'_t - \tilde{\eta}_t^2 \nabla \ell_1(A_1(\mathbf{x}_t), \mathbf{c}_1)\right), \mathbf{c}_2\right)\right). \tag{78}$$

Combining the Gaussian prior and the likelihood, the log-posterior is:

$$\log p(\mathbf{x}'_t \mid \mathbf{x}_t, \mathbf{c}_1, \mathbf{c}_2) \propto -\frac{\|\mathbf{x}_t - \mathbf{x}'_t\|^2}{2\eta_t^2} - \ell_1(A_1(\mathbf{x}_t), \mathbf{c}_1) - (\mathbf{x}'_t - \mathbf{x}_t)^\top \nabla \ell_1(A_1(\mathbf{x}_t), \mathbf{c}_1) - \ell_2(A_2(\mathbf{x}''_t), \mathbf{c}_2).$$
(79)

Differentiating with respect to $\mathbf{x}'_t$ and setting the gradient to zero for the MAP estimate gives:

$$\mathbf{x}'_t = \mathbf{x}_t - \eta_t^2 \nabla \ell_1(A_1(\mathbf{x}_t), \mathbf{c}_1) - \eta_t^2 \nabla \ell_2(A_2(\mathbf{x}''_t), \mathbf{c}_2).$$
(80)

For Gaussian posteriors, the MAP estimate coincides with the expectation, completing the proof. □

### C.4 PROOF OF THEOREM 3.3

For discretized diffusion timesteps $[t_1, t_2, \ldots, t_n]$, and with denoting $p_{tot} := \sum_j p_j(\mathbf{x})$, TAG for $i$-th timestep $t_i$ can be represented by rearranging the terms as follows:

$$
\begin{aligned}
\nabla_\mathbf{x} \log p(t_i|\mathbf{x}) &= \nabla_\mathbf{x} \log \left( \frac{p(\mathbf{x}|t_i)p(t_i)}{\sum_k p(\mathbf{x}|t_k)p(t_k)} \right) \\
&= \nabla_\mathbf{x} \log \left( \frac{p_i(\mathbf{x})}{p_{tot}(\mathbf{x})} \right) \\
&= \frac{\nabla_\mathbf{x} p_i(\mathbf{x})}{p_i(\mathbf{x})} - \frac{\nabla_\mathbf{x} p_{tot}(\mathbf{x})}{p_{tot}(\mathbf{x})} \\
&= \frac{\nabla_\mathbf{x} p_i(\mathbf{x})}{p_i(\mathbf{x})} - \frac{\sum_k \nabla_\mathbf{x} p_k(\mathbf{x})}{p_{tot}(\mathbf{x})} \\
&= (1 - \frac{p_i(\mathbf{x})}{p_{tot}(\mathbf{x})}) \nabla_\mathbf{x} \log p_i(\mathbf{x}) - \sum_{k \neq i} \frac{p_k(\mathbf{x})}{p_{tot}(\mathbf{x})} \nabla_\mathbf{x} \log p_k(\mathbf{x}) \\
&= \sum_{k \neq i} \frac{p_k(\mathbf{x})}{p_{tot}(\mathbf{x})} \left( \nabla_\mathbf{x} \log p_i(\mathbf{x}) - \nabla_\mathbf{x} \log p_k(\mathbf{x}) \right).
\end{aligned}
$$
(81)

□

### C.5 CONTINUOUS TIME LIMIT OF TAG

**Theorem C.2.** *(Continuous time TAG decomposition) For continuous time diffusion models, TLS score can be decomposed in the following way.*

$$\nabla_\mathbf{x} \log p(t|\mathbf{x}) = \nabla_\mathbf{x} \log p_t(\mathbf{x}) - \int \gamma_s \nabla_\mathbf{x} \log p_s(\mathbf{x}) ds,$$
(82)

*where $\gamma_s = \frac{p_s(\mathbf{x})}{\int p_k(\mathbf{x})dk}$.*

*Proof.*

$$
\begin{aligned}
\nabla_\mathbf{x} \log p(t|\mathbf{x}) &= \nabla_\mathbf{x} \log \left( \frac{p(\mathbf{x}|t)p(t)}{\int_s p(\mathbf{x}|s)p(s)} \right) \\
&= \nabla_\mathbf{x} \log \left( \frac{p_t(\mathbf{x})}{\int_s p(\mathbf{x}|s)ds} \right) \\
&= \frac{\nabla_\mathbf{x} p_t(\mathbf{x})}{p_t(\mathbf{x})} - \frac{\int \nabla_\mathbf{x} p_s(\mathbf{x})ds}{\int p_s(\mathbf{x})ds} \\
&= \nabla_\mathbf{x} \log p_t(\mathbf{x}) - \int \frac{p_s(\mathbf{x})}{\int p_k(\mathbf{x})dk} \nabla_\mathbf{x} \log p_s(\mathbf{x})ds,
\end{aligned}
$$
(83)

gives the result. □

## C.6 Proof of Proposition 3.4

We restate Proposition 3.4 below for convenience.

**Proposition C.3.** *Applying TAG alters energy barrier map $U_k(\mathbf{x}) = -\log p_k(\mathbf{x})$ at timestep $t_k$ to $\Phi_k(\mathbf{x})$ for any $k$ by:*

$$\Phi_k(\mathbf{x}) = U_k(\mathbf{x}) - \sum_i \gamma_i U_i(\mathbf{x}), \tag{84}$$

*where $\gamma_i = \frac{p_i(\mathbf{x})}{p_{tot}(\mathbf{x})}$ for all $i$.*

*Proof.* Denote $s_k$ as a new score term obtained by applying TAG at timestep $t_k$. Then, from Theorem 3.3, one can see that:

$$\begin{aligned}
\tilde{s}_k &:= \sum_{i \neq k} \gamma_i \left( s_k - s_i \right) \\
&= s_k - (1 - \sum_{i \neq k} \gamma_i) s_k - \sum_{i \neq k} \gamma_i s_i,
\end{aligned} \tag{85}$$

where $\gamma_i = \frac{p_i(\mathbf{x})}{p_{tot}(\mathbf{x})}$ as before. From the definition of the potential $U_i(\mathbf{x}) = -\log p_k(\mathbf{x})$ gradient of the $U_i$ equals to the score function $s_i$ for all $i$. Integrating both sides of the above equation and noting that the potential $U_k$ is defined up to additive constants, we get the result. $\qquad\square$

## C.7 Formal version of Theorem 3.5 with its proof

JKO scheme (Jordan et al., 1998) establishes the foundational argument that the Fokker-Planck equation of the Langevin dynamic is the gradient flow of the KL divergence with respect to the Wasserstein-2 metric. We can leverage this to analyze the convergence guarantee of the modified correction sampling by TAG. We start by defining original and modified Langevin dynamics below.

**Modified Langevin dynamics**  Original Langevin dynamics at timestep $t_k$ can be stated as,

$$d\mathbf{y}_t = \mathbf{s}_k(\mathbf{y}_t)dt + \sqrt{2}dW_t. \tag{86}$$

When applying TAG, from Theorem 3.3, Langevin dynamics in each step can be modified as,

$$d\mathbf{x}_t = \left[ \mathbf{s}_k(\mathbf{x}_t) - \sum_{i \neq k} \gamma_i \mathbf{s}_i(\mathbf{x}_t) \right] dt + \sqrt{2}dW_t. \tag{87}$$

**Fokker-Plank equation and gradient flow**

**Proposition C.4.** *(Fokker-Plank equation) For any smoothly evolving density $q_t$ driven by the Langevin dynamics of*

$$d\mathbf{x}_t = \mathbf{v}(\mathbf{x}_t, t)dt + \sqrt{2}dW_t, \tag{88}$$

*following equation holds:*

$$\partial_t q_t = -\nabla \cdot (q_t \mathbf{v}) + \Delta q_t, \tag{89}$$

*where $\Delta$ denotes Laplacian operator.*

**Theorem C.5.** *For Langevin dynamics in eq. 88, gradient flow of the KL functional has the following form:*

$$\frac{d}{dt} KL(q_t \| p_k) = -\mathbb{E}_{q_t} \left[ \| \nabla \log \frac{q_t}{p_k} \|^2 - \nabla \log \frac{q_t}{p_k} \cdot (\mathbf{v} - \nabla \log p_k) \right]. \tag{90}$$

*Proof.* Define the mismatch score $\mathbf{r}_k(\mathbf{x}, t) = \nabla_{\mathbf{x}} \log \frac{q_t(\mathbf{x})}{p_k(\mathbf{x})}$. One can observe that,

$$
\begin{aligned}
\frac{d}{dt} KL(q_t \| p_k) &= \int (\partial_t q_t) \log \frac{q_t}{p_k} d\mathbf{x} \\
&= \int [-\nabla \cdot (q_t \mathbf{v}) + \Delta q_t] \log \frac{q_t}{p_k} d\mathbf{x} \\
&= \int q_t \mathbf{v} \cdot \nabla \log \frac{q_t}{p_k} d\mathbf{x} \ - \int \nabla q_t \cdot \nabla \log \frac{q_t}{p_k} d\mathbf{x} \\
&= \int q_t \mathbf{v} \cdot \nabla \log \frac{q_t}{p_k} d\mathbf{x} \ - \int q_t \nabla \log q_t \cdot \nabla \log \frac{q_t}{p_k} d\mathbf{x} \\
&= \mathbb{E}_{q_t} \left[ \mathbf{v} \cdot \mathbf{r}_k - \nabla \log q_t \cdot \mathbf{r}_k \right],
\end{aligned} \tag{91}
$$

where second equality comes from the Proposition C.4, third equality comes from the integration-by-parts, and the last equality comes from the definition of $\mathbf{r}_k$. Now, from the definition of $\mathbf{r}_k$, we can rewrite,

$$
\nabla \log q_t = \mathbf{r}_k + \nabla \log p_k. \tag{92}
$$

Putting this into the above result in Eq. 91, we get the result. $\qquad\square$

Above theorem gives exact decreasing rate of KL divergence as shown by the following corollary:

**Corollary C.6.** *(Gradient flow of KL divergence) Applying Theorem C.5 to the original Langevin (Eq. 86), we can observe $\mathbf{v} - \nabla \log p_k = 0$, and from this, the last term in Eq. 90 is canceled out which gives,*

$$
\frac{d}{dt} KL(q_t \| p_k) = -\mathbb{E}_{q_t} \|\mathbf{r}_k\|^2, \tag{93}
$$

*Similarly, by applying Proposition C.3, we can obtain decreasing rate of modified Langevin (Eq. 87) as follows:*

$$
\frac{d}{dt} KL(\tilde{q}_t \| p_k) = -\mathbb{E}_{\tilde{q}_t} \left[ \|\tilde{\mathbf{r}}_k\|^2 + A(t) \right], \tag{94}
$$

*where $\tilde{\mathbf{r}}_k = \nabla \log \frac{\tilde{q}_t}{p_k}$ as before and $A(t) = \sum_i \gamma_i \mathbb{E}_{\tilde{q}_t} \left[ \tilde{\mathbf{r}}_k(\mathbf{x}, t) \cdot \mathbf{s}_i(\mathbf{x}) \right]$ is the extra term from the TAG.*

Intuitively, if the expectation of $A(t)$ in Eq. 94 is strictly positive, this helps escaping the low-density region faster compared to the original Langevin dynamics. To formalize this, we first define a low-density region in the following way.

**Definition C.7.** (Low density region) We say $\mathbf{x}$ falls into low-density region whenever,

$$
\mathbf{x} \in D_{k,\epsilon} \quad D_{k,\epsilon} = \{p_k(\mathbf{x}) \leq \epsilon\}, \tag{95}
$$

for some constant $\epsilon > 0$.

**Definition C.8.** (Escape time) Define stopping times $\tau, \tilde{\tau}$ as follows:

$$
\tau = \inf\{t \geq 0 : \mathbf{y}_t \neq D_{k,\epsilon}\}, \ \mathbf{y}_0 \sim q_0, \tag{96}
$$

where $q_t$ follows from original Langevin (Eq. 86) and

$$
\tilde{\tau} = \inf\{t \geq 0 : \mathbf{x}_t \neq D_{k,\epsilon}\}, \ \mathbf{x}_0 \sim \tilde{q}_0, \tag{97}
$$

where $\tilde{q}$ is from the modified Langevin by TAG (Eq. 87).

One can see that $\tau, \tilde{\tau}$ is the escaping time of the low-desnity region. Thus, lower $\tau, \tilde{\tau}$ implies a faster convergence toward high-density region, meaning accelerated initial convergence speed. This is captured by the following theorem.

**Theorem C.9.** *Assume the support of initial distribution $q_0(\mathbf{x})$ is inside $D_{k,\epsilon}$ and for all $t < \tilde{\tau}$, following equation holds:*

$$
\mathbb{E}_{q_t} \left[ \sum_j \gamma_j \, \tilde{\mathbf{r}}_k(\mathbf{x}_t) \cdot \mathbf{s}_j(\mathbf{x}_t) \ \big| \ \mathbf{x} \in D_{k,\epsilon} \right] \geq \beta \, , \ \beta > 0. \tag{98}
$$

*Moreover, assume the mixture score satisfies $\sum_i \gamma_i \mathbb{E}_{q_t}[\mathbf{s}_i] \leq \eta$ for $t \leq \tilde{\tau}$.*

*Then, the expectation of the stopping time $\tilde{\tau}$ is bounded as,*

$$\mathbb{E}[\tilde{\tau}] \leq \frac{KL(q_0||p_k)}{(\beta + \frac{\beta}{\eta^2})}, \tag{99}$$

*and consequently, tail bound of the escaping probability becomes:*

$$\Pr(\tilde{\tau} \geq t) \leq \frac{KL(q_0||p_k)}{t(\beta + \frac{\beta}{\eta^2})}. \tag{100}$$

*Proof.* First, from Cauchy-Schwarz inequality, one can observe:

$$\mathbb{E}_{q_t} \|\tilde{\mathbf{r}}_k(\mathbf{x})\|^2 \geq \frac{\mathbb{E}_{q_t} \sum_i \gamma_i \tilde{\mathbf{r}}_k(\mathbf{x}) \cdot \mathbf{s}_i(\mathbf{x})}{\mathbb{E}_{q_t} \|\sum_i \gamma_i \mathbf{s}_i(\mathbf{x})\|^2} \geq \frac{\beta}{\eta^2}. \tag{101}$$

As a result, gradient flow of KL divergence in Eq. 94 can be upper bounded by,

$$\frac{d}{dt} KL(\tilde{q}_t||p_k) = -\mathbb{E}_{\tilde{q}_t}\left[\|\tilde{\mathbf{r}}_k\|^2 + A(t)\right]$$

$$= -\mathbb{E}_{\tilde{q}_t} \|\tilde{\mathbf{r}}_k\|^2 - \sum_i \gamma_i \mathbb{E}_{\tilde{q}_t}[\tilde{\mathbf{r}}_k \cdot \mathbf{s_i}] \tag{102}$$

$$\leq -(\beta + \frac{\beta}{\eta^2}).$$

Now, it is straightforward to see that for $t^\star = t \wedge \tilde{\tau}$ and $\delta := \beta + \frac{\beta}{\eta^2}$,

$$KL(q_{t^\star}||p_k) \leq KL(q_0||p_K) - \delta t^\star. \tag{103}$$

From the positiveness of the KL,

$$\delta t^\star \leq KL(q_0||p_k). \tag{104}$$

Now, taking expectation and sends $t \to \infty$ gives

$$\mathbb{E}[\tilde{\tau}] = \frac{KL(q_0||p_k)}{\delta}, \tag{105}$$

which recovers Eq. 99. Now, form Markov's inequality, Eq. 100 holds. $\square$

**Corollary C.10.** *Applying TAG can accelerate convergence speed in a sense that upper bound of $\mathbb{E}[\tilde{\tau}]$ is reduced compared to the upper bound of $\mathbb{E}[\tau]$ by the factor of $1 + \eta^2$. Moreover, continuous flow of the modified Langevin dynamics until time $t$ reduces KL divergence to the target measure by,*

$$KL(q_0||p_k) - KL(q_t||p_k) \geq t\beta(1 + \frac{1}{\eta^2}) \tag{106}$$

The improvement factor $1 + \eta^2$ grows over increasing $\eta$ which implies that if the expectation of the mixture score increases, faster convergence can be guaranteed. This agrees with our intuition that even the $\mathbf{x} \sim q_t$ mostly resides in the low density region of the single timestep distribution $p_k$, $p_j(\mathbf{x})$ can be high for some $j \neq k$ and thereby contribute to the term $\eta$.

**Assumption C.11.** Score approximation error is *monotonically decreasing* function of the density function $p_t(\mathbf{x})$. Specifically, assume for all $t$ in the diffusion process, there exist a monotonic increasing function $h_t : \mathbb{R}_{\geq 0} \to \mathbb{R}_{\geq 0}$ with $\|\nabla h_t\| \geq m > 0$ such that following relation holds:

$$\mathbb{E}_{\mathbf{x} \sim q_t} \|\nabla_{\mathbf{x}} \log p_k(\mathbf{x}) - s_\theta(\mathbf{x}, t_k)\|_2^2 = h_t\left(KL\left(q_t||p_k\right)\right) \tag{107}$$

The above assumption implies that if a particle deviates far from the true distribution $p_k$, score approximation error increases. This is reasonable to assume in a sense that a neural network is trained only with the sample from $p_k$ and rarely sees the sample from $p_k(\mathbf{x}) \approx 0$.

With above assumptions, we provide the formal version of the Theorem 3.5.

**Theorem C.12.** *(Formal version of Theorem 3.5) Denote $\tilde{p}_t$ is reverse process of diffusion in Eq. 2. Given, Assumption C.11 and assumptions in Theorem C.5, the convergence guarantee for small $t_0 > 0$ can be improved by simulating modified Langevin correction in Eq. 87 until time $s$ in the following way.*

$$d_{TV}(\tilde{p}_{t_0}, q_0) \leq d_{TV}(p_{t_0}, q_0) - \frac{G}{4\sqrt{F}}, \tag{108}$$

*where*

$$F = (T - t_0)\sqrt{\mathbb{E}_{\mathbf{x} \sim p_t}\left[\frac{1}{2}\int_{t_0}^{T} g(t)^{-2}\|\mathbf{s}_\theta(\mathbf{x}) - \nabla_{\mathbf{x}} \log p_t(\mathbf{x})\|_2^2\, dt\right]}, \tag{109}$$

*is the original score approximation error and*

$$G = m\beta(1 + \frac{1}{\eta^2})s \cdot \int_{t_0}^{T} g(t)^{-2}dt. \tag{110}$$

*Proof.* For path measure of forward process $\mathbb{Q}$ defined from $t_0$ to $T$ and the path measure of the corresponding reverse process $\mathbb{P}$, estimation error is decomposed as

$$\mathbb{E}[TV(\mathbf{x}_0, \mathbf{x}_{t_0})] + \mathbb{E}[TV(\mathbf{x}_T, \mathcal{N}(0, \mathbf{I})] + TV(\mathbb{P}, \mathbb{Q}) \tag{111}$$

where first term is the truncation error, second term is initial noise mismatch between forward and reverse process, and the third term is KL divergence between path measures score approximation errors(for discrete sampling, additional discretization error is added as in (Chen et al., 2023b)). Chen et al. (2023b); Oko et al. (2023) show that the third term can be bounded by score approximation errors (please refer to Appendix C.1 and Section 5.2 of (Chen et al., 2023b) for details). Specifically, it can be shown from the Proposition C.1 and triangle inequality,

$$\begin{aligned}
TV(\mathbb{P}, \mathbb{Q}) \leq &\sqrt{\mathbb{E}_{\mathbf{x} \sim p_t}\left[\frac{1}{2}\int_{t_0}^{T} g(t)^{-2}\|\mathbf{s}_\theta(\mathbf{x}) - \nabla_{\mathbf{x}} \log p_t(\mathbf{x})\|_2^2\, dt\right]} \\
&+ \sqrt{\mathbb{E}_{\mathbf{x} \sim p_t}\left[\frac{1}{2}\int_{t_0}^{T} g(t)^{-2}\|\mathbf{v}(\mathbf{x}, \mathbf{c}, t)\|_2^2\, dt\right]}.
\end{aligned} \tag{112}$$

Now, one can observe from Corollary C.10, reduced score approximation error by simulating modified Langevin (Eq. 87) until time $s$ gives,

$$\mathbb{E}_{\mathbf{x} \sim q_s}\|\nabla_{\mathbf{x}} \log p_k(\mathbf{x}) - s_\theta(\mathbf{x}, t_k)\|_2^2 \leq \mathbb{E}_{\mathbf{x} \sim q_0}\|\nabla_{\mathbf{x}} \log p_k(\mathbf{x}) - s_\theta(\mathbf{x}, t_k)\|_2^2 - m\beta(1 + \frac{1}{\eta^2})s. \tag{113}$$

Now, observing that for two constants $f, g > 0$ and $f > g$,

$$\sqrt{f} - \sqrt{f - g} = \frac{g}{\sqrt{f} + \sqrt{f - g}} \geq \frac{g}{2\sqrt{f}}. \tag{114}$$

Putting $f = \mathbb{E}_{\mathbf{x} \sim q_0}\|\nabla_{\mathbf{x}} \log p_k(\mathbf{x}) - s_\theta(\mathbf{x}, t_k)\|_2^2$, $g = m\beta(1 + \frac{1}{\eta^2})s$ into above and combining with Eq. 112 by applying Cauchy-Schwarz inequality gives the result.

$\square$

Note that for the single discretized Langevin step can be also analyzed similarly for small step-size $h$ from the Girasonov theorem.

# D  RELATION TO PRIOR WORKS

## D.1  RELATED WORKS

**External Guidance in Diffusion Models**  Diffusion models can be leveraged in downstream applications by combining an unconditional diffusion process with external guidance—without any additional training. Graikos et al. (2022) use off-the-shelf diffusion models to generate samples

constrained to specific conditions, demonstrating applications in combinatorial optimization, while Chung et al. (2023) apply similar guidance to solve inverse problems. Bansal et al. (2024) extend this approach to user-specific conditioning in the image domain. TFG (Ye et al., 2024) provide a unified training-free guidance framework by consolidating the design space of prior methods, searching for optimal hyperparameter combinations, and establishing benchmarks for training-free guidance. For scenarios involving multiple constraints, MultiDiffusion (Bar-Tal et al., 2023) achieves spatial control by fusing diffusion paths from different prompts.

**Off-Manifold Phenomenon**    Diffusion models exhibit exposure bias (Ning et al., 2024), as the reverse process does not match the learned forward process. Ning et al. (2023) reduce exposure bias by randomly perturbing the training data in diffusion models. Ning et al. (2024) show that scaling the vector norm of the diffusion model outputs can alleviate errors, while Li et al. (2024a) identify variance across sample batches to correct the time information in diffusion models. Song & Ermon (2019) explore Langevin dynamics–based steps that utilize the learned score function for iterative refinement. Li & van der Schaar (2024) theoretically analyze how errors accumulate during the reverse process of diffusion models.

**Timestep in Diffusion Models**    Several studies have investigated the impact of timestep information in diffusion models. Xia et al. (2024) optimize timestep embeddings to correct the sampling direction, and San-Roman et al. (2021) demonstrate the effectiveness of adjusting the noise schedule. Kim & Ye (2023) and Kahouli et al. (2024) train neural networks to estimate accurate timestep information, while Jung et al. (2024) leverage a time predictor to modify the reverse diffusion schedule and correct the reverse process. Sadat et al. (2024) and Li et al. (2024b) perturb time inputs in the primary score model to derive contrastive signals. Yadin et al. (2024) utilize time classifiers for score function approximation. In contrast, our TAG framework learns a dedicated time predictor for $p(t \mid \mathbf{x}_t)$ and directly leverages its score $\nabla_{\mathbf{x}} \log p(t \mid \mathbf{x})$ to provably pull samples back onto their true temporal manifold—yielding both convergence guarantees and empirical gains for the first time. A formal proof and discussion are provided in Appendix D.

**Comparison with other regularization techniques**    Recent works (Fan et al., 2023; Wallace et al., 2024) show fine-tune diffusion model using the reinforcement learning algorithm can boost the performance of diffusion models. To not diverge from the original diffusion process, they utilize KL regularization technique. However, fine-tuning diffusion models for practical downstream tasks is highly costly where target condition vary in real-time. Another option is to utilize control theory (Berner et al., 2024; Uehara et al., 2025) where the objective is to refine the sample trajectory to the desired reward weighted distribution with the KL regularization term added. However, this usually rely on generating multiple sample trajectories. In contrast to above techniques, our method does not require offline history nor multiple iterations, readily applicable even when target condition changes for every generation.

**Off-Manifold in Classifier-Free Guidance and TAG**    Although, well-optimized CFG methods may show fewer deviations, they fundamentally rely on a linear extrapolation between conditional and unconditional scores. Chung et al. (2025); Sadat et al. (2024) analyze that at high guidance scales, this extrapolation inevitably pushes the sampling trajectory away from the natural data manifold, resulting in over-saturated or "burned" artifacts. This aligns precisely with the off-manifold deviation defined in our work (Section 2), where the generated samples drift into low-density regions of the data distribution.

TAG is uniquely positioned to address this issue because it introduces an orthogonal corrective force. While CFG modifies the score to satisfy the condition $c$ (often at the cost of realism), TAG's Time-Linked Score (TLS) $\nabla_x \log p(t|x)$ provides a distinct gradient that pulls the sample back to the high-density manifold of the current noise level. As discussed visualized in Figure 2, this correction helps restore the valid noise level without conflicting with the semantic steering of the guidance. Therefore, integrating TAG into the CFG framework can serve as an effective manifold constraint, mitigating the "burned" artifacts caused by excessive extrapolation.

This perspective is consistent with recent manifold-preserving approaches like Chung et al. (2025) and He et al. (2024), which also identify the need for correction in CFG. However, unlike methods that

require specific architectural changes or heavy optimization, TAG offers a lightweight, plug-and-play solution by leveraging the temporal consistency inherent in the diffusion process itself.

**Relation to Prior Manifold-Preserving Approaches**   Prior approaches address manifold deviation through geometric constraints or vector corrections within the data space, TAG introduces a completely novel perspective: identifying and correcting off-manifold errors via the temporal axis. We are the first to frame this issue as a timestep deviation and correct it using the Time-Linked Score (TLS), providing a unique orthogonal contribution that can be applied on top of the prior methods.

DSG (Yang et al., 2024) constrains the guidance step within a spherical shell centered at the unconditional mean $\mu_\theta(x_t)$, based on the concentration of high-dimensional Gaussian distributions. While DSG relies on a projection onto a sphere to mitigate deviation, TAG employs a learned Time-Linked Score (TLS), $\nabla_x \log p(t|x)$. Unlike DSG's hard constraint which assumes isotropy, TAG provides a soft correction that respects the complex, non-isotropic structure of the density field, actively pulling samples back to the high-density region of the specific timestep manifold $\mathcal{M}_t$.

MPGD (He et al., 2024) projects the guidance gradient onto the tangent space of the data manifold using a pre-trained autoencoder's Jacobian. MPGD relies on the locally linear approximation and requires an auxiliary autoencoder, which may not be available for all domains (e.g., molecular graphs). In contrast, TAG is model-agnostic and does not require manifold linearity or external autoencoders. It leverages the diffusion process's own temporal information, making it applicable to diverse domains (images, molecules, audio) where such geometric assumptions may fail.

CFG++ (Chung et al., 2025) modifies the vector field of classifier-free guidance to remain on the data manifold by correcting the unconditional score component. CFG++ is specific to the CFG framework and focuses on correcting the extrapolation error of the guidance vector itself. TAG is a general-purpose correction mechanism applicable to various guidance settings (classifier guidance, cfg, and training-free gudiance). TAG can be integrated on top of CFG to further correct off-manifold drifts caused by aggressive scaling, offering an orthogonal layer of robustness.

### D.2   COMPARISON WITH BASELINE METHODS

Here, we elaborate on the detailed discussions of TAG with other relevant literatures that were briefly introduced in Sec. D.1.

**Early timestep and schedule optimizations**   Early works exploring the role of time include Xia et al. (2024) optimizing timestep embeddings, and San-Roman et al. (2021) focusing adjusting noise schedules. These approaches typically aim to find globally or locally optimal fixed schedules or input representations for time and generally do not offer sample-adaptive corrections at each step based on the evolving state of $\mathbf{x}$. TAG, in contrast, provides such a dynamic, sample-specific correction via its TLS (Eq. 12). This helps the sample adhere to the manifold implied by the schedule at each current timestep $t$ by considering the full posterior $p_\phi(\cdot|\mathbf{x})$, thus acting as a more flexible and adaptive generalized constraint than pre-defined temporal strategies.

**Time Correction Sampler**   TCS (Jung et al., 2024) also employs a time predictor. However, TCS uses the predictor's output, $\tilde{t} = \arg\max \phi(\mathbf{x}_t)$, to directly modify the perceived timestep of the sample $\mathbf{x}_t$, subsequently altering the solver step to use $\mathbf{s}_\theta(\mathbf{x}_t, \tilde{t})$ and adjusting the noise schedule. This constitutes a "hard" temporal reassignment. TAG differs significantly by maintaining the solver's current timestep $t$ for $\mathbf{s}_\theta(\mathbf{x}_t, t)$ and instead adding the TLS as an additive correction to the sample itself. The TLS decomposition (Eq. 12) suggests TAG's correction is a generalized constraint, as it considers attractive and repulsive forces from all potential timesteps based on $p_\phi(\cdot|\mathbf{x})$, rather than a singular reassignment, offering a robust means to maintain manifold fidelity; our comparative experiments (Table 2) demonstrate TAG's superior performance over TCS.

**Time perturbation methods**   TSG (Sadat et al., 2024) and SG (Li et al., 2024b) leverage the score model's ($\mathbf{s}_\theta$) local sensitivity to time by perturbing its time input $\tau$ (e.g., $\tau \pm \delta\tau$) to derive contrastive guidance. In contrast, TAG employs its distinct TLS (Eq. 12) as an additive corrective term, without altering $\mathbf{s}_\theta$'s time input. Theorem 3.3 shows that applying TAG has effect of getting negative guidance from all timesteps except the target timestep (i.e, current timestep) which potentially renders TAG

more robust than the typically symmetric or single-perturbation strategies of TSG and SG, especially for significantly off-manifold scenarios.

**Exposure bias** While methods like Epsilon Scaling (Ning et al., 2024) and the Time-Shift Sampler (Li et al., 2024a) act during inference, similar to TAG, they typically apply heuristic adjustments (e.g., scaling model outputs or shifting time inputs) to mitigate train-inference mismatch. Other approaches, such as that by Ning et al. (2023), modify the training data itself. TAG differs fundamentally by introducing a learned score term – the adaptive TLS, $\nabla_{\mathbf{x}} \log p_\phi(t|\mathbf{x})$ – which provides active, sample-specific correction based on learned temporal consistency, rather than relying on pre-defined heuristics or training data alterations. We compare TAG against (Ning et al., 2023) (Table 8), (Ning et al., 2024) and (Li et al., 2024a) (in Table 9). TAG demonstrate consistent improvements against all baselines.

**Classification Diffusion Models** While both TAG and CDM (Yadin et al., 2024) uses a timestep classifier, the primary purpose of CDM is to estimate the log density of the generative output and approximate the score function in each timestep. Contrary to this, TAG leverages the gradient of a time classifier whose purpose is to attract the sample to the desired timestep for reducing off-manifold phenomenon. We notice that Theorem 3.1 in CDM (Yadin et al., 2024) can be deduced from rearranging term in Theorem 3.3 of ours as follows:

We begin by noting that in Theorem 3.3, $t_i$ is an arbitrary label in $\{t_1, \ldots, t_n\}$. In CDM (Yadin et al., 2024) setting, we simply identify $t_i = t$ and $t_{T+1} = T + 1$.

We now show that

$$\nabla_{\mathbf{x}} \log\big(p_{\tau|\tilde{\mathbf{x}}}(t \mid \mathbf{x})\big) \;-\; \nabla_{\mathbf{x}} \log\big(p_{\tau|\tilde{\mathbf{x}}}(T + 1 \mid \mathbf{x})\big) = \nabla_{\mathbf{x}} \log p\big(t_i \mid \mathbf{x}\big) \;-\; \nabla_{\mathbf{x}} \log p\big(t_{T+1} \mid \mathbf{x}\big)$$
$$= \nabla \log p_i(\mathbf{x}) \;-\; \nabla \log p_{T+1}(\mathbf{x}).$$

Let $\gamma_k = \frac{p_k(\mathbf{x})}{p_{\text{tot}}(\mathbf{x})}$. Then

$$\nabla_{\mathbf{x}} \log\big(p_{\tau|\tilde{\mathbf{x}}}(t \mid \mathbf{x})\big) = \sum_{k \neq i} \gamma_k \Big(\nabla \log p_i(\mathbf{x}) \;-\; \nabla \log p_k(\mathbf{x})\Big)$$

$$= \underbrace{(1 - \gamma_i)\nabla \log p_i(\mathbf{x})}_{\sum_{k \neq i} \gamma_k \nabla \log p_i(\mathbf{x})} - \underbrace{\sum_k \gamma_k \nabla \log p_k(\mathbf{x}) \;+\; \gamma_i \nabla \log p_i(\mathbf{x})}_{(\star)},$$

because $\sum_{k \neq i} \gamma_k = 1 - \gamma_i$. Similarly,

$$-\nabla_{\mathbf{x}} \log\big(p_{\tau|\tilde{\mathbf{x}}}(T + 1 \mid \mathbf{x})\big) = - \sum_{k \neq T+1} \gamma_k \Big(\nabla \log p_{T+1}(\mathbf{x}) \;-\; \nabla \log p_k(\mathbf{x})\Big)$$

$$= -\big(1 - \gamma_{T+1}\big) \nabla \log p_{T+1}(\mathbf{x}) + \underbrace{\sum_k \gamma_k \nabla \log p_k(\mathbf{x})}_{(\star)} \;-\; \gamma_{T+1} \nabla \log p_{T+1}(\mathbf{x}).$$

The terms $\sum_k \gamma_k \nabla \log p_k(\mathbf{x})$, labeled $(\star)$, cancel out. What remains is $\nabla \log p_i(\mathbf{x}) - \nabla \log p_{T+1}(\mathbf{x})$.

Thus,

$$\nabla_{\mathbf{x}} \log\big(p_{\tau|\tilde{\mathbf{x}}}(t \mid \mathbf{x})\big) \;-\; \nabla_{\mathbf{x}} \log\big(p_{\tau|\tilde{\mathbf{x}}}(T + 1 \mid \mathbf{x})\big) = \nabla \log p_i(\mathbf{x}) \;-\; \nabla \log p_{T+1}(\mathbf{x})$$
$$= \nabla_{\mathbf{x}} \log\big(p_{\mathbf{x}_t}(\mathbf{x})\big) \;-\; \nabla_{\mathbf{x}} \log\big(p_{\mathbf{x}_{T+1}}(\mathbf{x})\big).$$

As shown in Appendix B.1, Eq. (15) of (Yadin et al., 2024), combining this with the Gaussian prior argument and Tweedie's formula Efron (2011) yields

$$\mathbb{E}\big[\varepsilon_t \mid \mathbf{x}_t = \mathbf{x}\big] = \sqrt{1 - \overline{\alpha}_t} \; \Big[ \nabla_{\mathbf{x}} \log\big(p_{\tau|\tilde{\mathbf{x}}}(T + 1 \mid \mathbf{x})\big) \;-\; \nabla_{\mathbf{x}} \log\big(p_{\tau|\tilde{\mathbf{x}}}(t \mid \mathbf{x})\big) \;+\; \mathbf{x}\Big],$$

which completes the derivation.

**Predictor-Corrector (PC) sampling** Energy Diffusion (Du et al., 2024) and NCSN (Song & Ermon, 2019) rely on the score $\nabla_{\mathbf{x}} \log p_t(\mathbf{x})$ for generation and refinement. TAG, however, adds a separate correction using the distinct TLS gradient, $\nabla_{\mathbf{x}} \log p_\phi(t|\mathbf{x})$. Theorem 3.3 implies TAG's correction is uniquely adaptive—strengthening when samples are far from the manifold based on relative time probabilities. This adaptiveness may offer greater robustness than relying solely on $\nabla_{\mathbf{x}} \log p_t(\mathbf{x})$, which can be error-prone when off-manifold phenomena are present. Our direct experimental comparisons support this distinction; for instance, while some score-based correction sampling approaches (e.g., (Song et al., 2021b)) can degrade sampling quality under external guidance (such as with DPS), Table 9 demonstrates TAG outperforms over such baselines.

**Multi-condition generation** MultiDiffusion (Bar-Tal et al., 2023) achieves spatial multi-condition control by fusing diffusion paths from multiple prompts, primarily targeting different image regions. Our approach to handling multiple conditions with TAG differs: we focus on combining multiple standard guidance terms and mitigating any resulting off-manifold drift by applying TAG. For efficiency in such scenarios, TAG's corrective temporal gradient, the TLS ($\nabla_{\mathbf{x}} \log p_\phi(t|\mathbf{x})$), can be approximated using time predictors conditioned on single conditions or even an unconditional time predictor, as detailed in Sec. 3.2 and Appendix B.3. Thus, while MultiDiffusion employs path fusion mainly for spatial control objectives, TAG utilizes approximated temporal alignment to maintain manifold adherence when faced with combined guidance from multiple standard conditional inputs.

**Fine-tuning vs. TAG** Standard approaches to adapt diffusion models for downstream tasks—such as adding spatial conditioning modules in ControlNet (Zhang et al., 2023), text-compatible prompt adapters in IP-Adapter (Ye et al., 2023), or rl–based reward tuning (Fan et al., 2023; Clark et al., 2024)—require collecting task-specific labeled data, modifying model architectures, and performing hours of gradient-based optimization. In contrast, TAG trains only a lightweight time predictor on noisy vs. clean timestep labels, completing in minutes on a mimal computational resources (Jung et al., 2024). At inference, TAG injects a temporally driven corrective gradient that steers samples back onto the appropriate diffusion manifold without altering the base model's weights. This inference-time, training-free correction avoids fine-tuning's cost and overfitting risks while supporting new guidance objectives such as reward alignment with DAS (Kim et al., 2025), multi-condition steering (Uehara et al., 2025), and style control via RB-Modulation (Rout et al., 2025). Moreover, by leveraging fundamental temporal consistency, TAG improves out-of-distribution robustness in tasks ranging from image and audio restoration to molecular generation (Ye et al., 2024; Bar-Tal et al., 2023). Thus, TAG offers a general, low-overhead alternative to fine-tuning for mitigating off-manifold drift in guided diffusion.

**Baseline experiments** Here, we present experimental results comparing TAG against the baselines and prior works discussed throughout the section.

Table 8: Effect of Input perturbation on DPS, CIFAR-10. For fair comparison, we train diffusion models with different $\eta$ from scratch following the official implementation code in [9]. No improvement over original diffusion model ($\eta = 0$) is observed in the presence of off-manifold phenomenon. We report the average value for 512 samples per each conditioning labels.

| Method | FID ↓ | Acc. ↑ |
|---|---|---|
| $\eta = 0$ | 332.0 | 28.5 |
| $\eta = 0.05$ | 409.9 | 23.3 |
| $\eta = 0.10$ | 376.6 | 25.4 |
| $\eta = 0.15$ | 326.7 | 29.2 |

# E  IMPLEMENTATION DETAILS

## E.1  TOY EXPERIMENT

**Setup** We construct the dataset from randomly generate 40,000 samples from the mixtures of two Gaussians as $q_0 \sim \frac{1}{2}\mathcal{N}((10,10), \boldsymbol{I}) + \frac{1}{2}\mathcal{N}((-10,-10), \boldsymbol{I})$. DDPM (VP-SDE) is utilized for

Table 9: Additional baselines when applying DPS on CIFAR-10. TAG improves the performance of DPS while other method struggles.

| Method | FID ↓ | Acc. ↑ |
|---|---|---|
| DPS | 217.1 | 57.5 |
| TAG (ours) | 190.4 | **63.2** |
| TCS [15] | 213.4 | 29.4 |
| Timestep Guidance [16] | 393.2 | 9.4 |
| Self-Guidance [17] | 205.4 | 51.6 |
| Epsilon Scaling [10] | **186.0** | 53.0 |
| Time Shift Sampler [11] | 237.0 | 60.8 |
| Langevin Dynamics [13] | 226.8 | 58.2 |

diffusion process with total 100 diffusion timesteps. $\mathbf{v}(\mathbf{x}, t) = -0.01\,\mathbf{x}$ is applied as an external drift for every timestep.

**Training details**  For diffusion models, we use 3-layer MLP with 5000 training epochs with full-batch size. For time predictor, 5-layer MLP is utilized with 5000 training epochs with full-batch size. We utilize a single RTX 3090 GPU for the experiment.

The predictor size in the toy experiment was not critical; TAG performed well even with a predictor smaller than the score network, as shown in our ablation study Table 10. Importantly, in our main experiments (Table 2), the effective SimpleCNN predictor is significantly smaller than the UNet diffusion backbone, demonstrating TAG's efficiency and lack of dependence on a large predictor relative to the main model. See Appendix E.4 for further discussion on classifier robustness.

| Layers | W1 distance ↓ |
|---|---|
| 0 (No TAG) | 6.458 |
| 1 | 1.716 |
| 2 | 1.681 |
| 3 | 1.975 |
| 4 | 1.714 |
| 5 | 1.713 |
| 6 | 1.788 |

Table 10: Robustness of time classifier network on toy experiment. We measure Wasserstein distance ($W_1$) for 10,000 samples. Consistent improvement compared to original reverse process when applying TAG independent of layer numbers.

**Trajectory analysis**  We further analyze the effect of the TAG along the diffusion sampling trajectory. To do this, we quantitatively compare the W1 distance between the two objective at each sampling timestep: (1) original sampling without external drift term ($\mathbf{v}(\mathbf{x}, t) = -0.01\mathbf{x}$) and original sampling with external drift, (2) original sampling without external drift and TAG with external drift.
Fig. 4 demonstrates that our Temporal Alignment Guidance (TAG, red curve) actively steers the generation process significantly more than standard DDPM sampler (Blue), resulting in high gap of W1 at t=0. We also put the visualization result in Fig. 5. Here, while original DDPM sample struggle to move the sample to the data manifold due to the external drift, TAG effectively push the sample toward the target manifold, staying near the target distribution even with strong external drift term $\mathbf{v}$.

**Score approximation error**  We further conduct an experiment to directly observe the score approximation error accumulation. We do this by measuring the Fisher Divergence at each timestep during the sampling, which is defined by the $\mathbb{E}_{\mathbf{x}_t}\left[\|s_\theta(\mathbf{x}_t, t) - \nabla_{\mathbf{x}_t} \log p_t(\mathbf{x}_t)\|_2^2\right]$. We compare DDPM sampler and TAG in the presence of the external drift $\mathbf{v}(\mathbf{x}, t) = -0.01\mathbf{x}$ and measure (1)

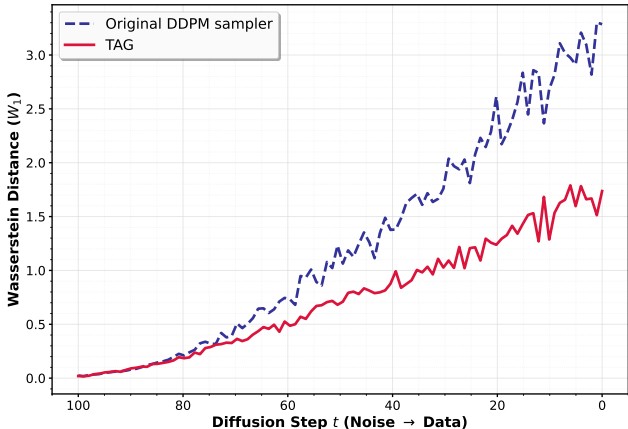

**Figure 4:** We measure the Sliced Wasserstein ($W_1$) distance between samples with the external drift and without the external drift for each timestep.

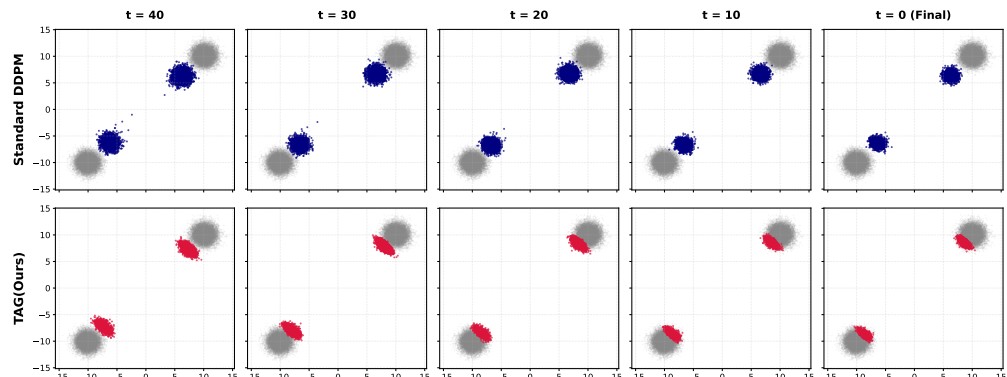

**Figure 5:** Visualization of the sampling trajectory for the toy experiment. Samples in timestep 40,30,20,10, and 0 for standard DDPM vs. TAG with external drift term.

instantaneous error: score approximation error for each timestep, (2) cumulative error: cumulative score approximation error during the sampling history. The result in Fig. 6 shows how the ratio between score approximation error (TAG's approximation error divided by DDPM's approximation error). The result shows that TAG significantly reduce the score approximation error both at each timestep (instantaneous) and over the trajectory (cumulative), reaching more than 40 times of reduction in error, demonstrating TAG's effectiveness for reducing score approximation errors.

The above analyses together show that TAG consistently performs as an effective corrector throughout the diffusion sampling process, proving robustness of our approach as a new sampler.

## E.2 CORRUPTED REVERSE PROCESS

**Setup** We use the CIFAR-10 dataset and intentionally add random noise $\mathbf{z}_t \sim \mathcal{N}(0, \sigma^2 \mathbf{I})$ to the sample $\mathbf{x}_t$ at each reverse timestep $t$. This simulates a strong, non-physical perturbation pushing samples off-manifold. We generate 50,000 samples and evaluate using FID (Karras et al., 2018), IS (Salimans et al., 2016), and the Time-gap (Def. F.1). We utilize the pre-trained model CIFAR10-DDPM Nichol & Dhariwal (2021) and use DDIM sampling with 50 diffusion timesteps. We run our our experiment on a single A6000 GPU for the inference.

**Algorithm** In Algorithm 2, we provide a pseudo-code of the corrupted reverse process setting conducted in Section 3.4.

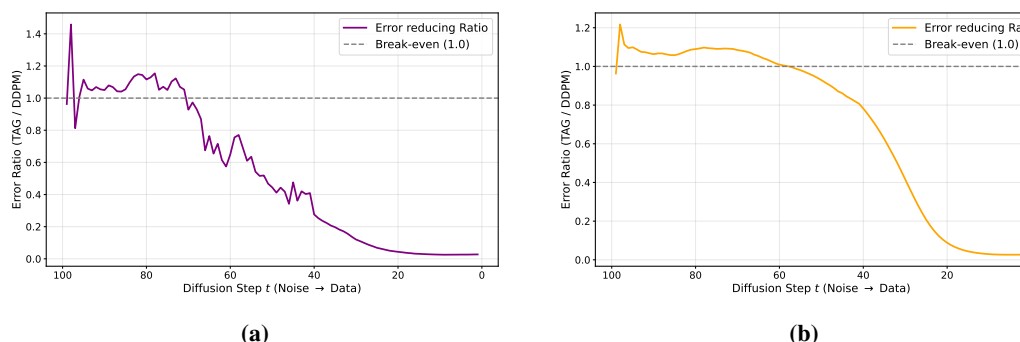

**(a)**                                                                 **(b)**

**Figure 6:** Visualization of score approximation error between standard TAG and DDPM (TAG/DDPM): (a) Instantaneous ratio of each timestep. (b) Cumulative ratio along diffusion sampling trajectory.

---

**Algorithm 2** Corrupted reverse process with TAG

**Input:** Diffusion model $\boldsymbol{\theta}$, time predictor $\phi$, guidance strength schedule $\omega(t)$, number of total diffusion steps $T$, Noise level $\sigma$.
$\boldsymbol{x}_T \sim \mathcal{N}(\mathbf{0}, \boldsymbol{I})$
**for** $t = T, \cdots, 1$ **do**
    $\mathbf{x}_t \leftarrow \mathbf{x}_t + \sigma \cdot \epsilon_\mathbf{t}$ where $\epsilon_t \sim \mathcal{N}(0, \boldsymbol{I})$ ▷ Random noise with strength $\sigma$ is added at each reverse diffusion timestep
    Obtain $\nabla \log p(\mathbf{x})$ from a diffusion model $\boldsymbol{\theta}$
    $\tilde{\mathbf{x}}_{t-1} \leftarrow \boldsymbol{x}_t$ from reverse diffusion step                       ▷ following Eq. 1
    Calculate $\nabla \log p_\phi(\tilde{\mathbf{x}}_{t-1})$          ▷ Calculating TLS score from the time predictor $\phi$
    $\mathbf{x}_{t-1} \leftarrow \tilde{\mathbf{x}}_{t-1} + \omega(t) \cdot \nabla \log p_\phi(\tilde{\mathbf{x}}_{t-1})$.              ▷ Applying TAG
**end for**
**Output:** $\boldsymbol{x}_0$

---

**Guidance schedule** For the experiment, we use guidance schedule of $w_t = w \cdot (1 - \bar{\alpha}_t)$ and where we refer $\omega$ as the guidance strength unless stated otherwise.

**Additional experimental results** Here, to illustrate the correlation between TAG strength $\omega$ and performance metrics, we provide a grid search results on the effect of guidance strength weight $\omega$.

For the experiment, we fix the noise schedule $\sigma = 0.3$ and follow the same setting in Section 3.4. The result is in Table 11 and one can observe that applying strong time guidance consistently increase the generation quality until performance is saturated.

Table 11: Grid search result on the effects of TAG strength $\omega$ with $\sigma = 0.3$.

| $\omega$ | 0 | 0.5 | 1.0 | 2.0 | 3.0 | 4.0 | 5.0 | 6.0 | 7.0 | 8.0 | 9.0 |
|---|---|---|---|---|---|---|---|---|---|---|---|
| TG ↓ | 273.9 | 261.6 | 250.9 | 232.6 | 213.3 | 197.8 | 185.1 | 175.1 | 167.9 | 162.7 | 158.9 |
| FID ↓ | 410.1 | 408.5 | 406.7 | 390.3 | 376.2 | 361.2 | 350.6 | 344.3 | 339.1 | 335.6 | 334.6 |
| IS ↑ | 1.27 | 1.28 | 1.28 | 1.27 | 1.29 | 1.31 | 1.35 | 1.38 | 1.41 | 1.43 | 1.45 |

| $\omega$ | 10.0 | 15.0 | 20.0 | 25.0 | 30.0 | 40.0 | 50.0 | 75.0 | 100.0 | 150.0 | 200.0 |
|---|---|---|---|---|---|---|---|---|---|---|---|
| TG ↓ | 156.0 | 143.2 | 129.1 | 116.2 | 107.5 | 98.8 | 108.6 | 140.0 | 153.7 | 160.0 | 158.9 |
| FID ↓ | 345.6 | 318.8 | 291.4 | 277.1 | 270.7 | 257.6 | 247.1 | 240.8 | 236.7 | 229.5 | 223.2 |
| IS ↑ | 1.43 | 1.52 | 1.57 | 1.60 | 1.63 | 1.72 | 1.78 | 1.90 | 2.01 | 2.14 | 2.17 |

We also provide quantitative results of Figure 3 in Table 12. We measure FID (Karras et al., 2018), IS (Salimans et al., 2016), with time gap F.1 for the experiment and the best values for each noise strength $\sigma$ where the guidance strength $w$ with the lowest FID value is reported. We find $w = 0.2, 1.25, 4.5, 200.0$ shows the lowest FID value for the noise strength level of $\sigma = 0.05, 0.1, 0.2, 0.3$ respectively. The results shows that applying TAG consistently improves the FID and IS scores

across all noise levels, particularly with higher external noise strength (higher $\sigma$), demonstrating the effectiveness of the TAG under the presence of external guidance.

Table 12: Comparison between original diffusion process and diffusion process with TAG across different noise strengths.

| | $\sigma = 0.05$ | | | $\sigma = 0.1$ | | | $\sigma = 0.2$ | | | $\sigma = 0.3$ | | |
|---|---|---|---|---|---|---|---|---|---|---|---|---|
| TAG | TG↓ | FID↓ | IS↑ | TG↓ | FID↓ | IS↑ | TG↓ | FID↓ | IS↑ | TG↓ | FID↓ | IS↑ |
| ✗ | 43.0 | 78.9 | 5.29 | 104.1 | 193.6 | 2.37 | 229.6 | 351.4 | 1.50 | 274.0 | 410.1 | 1.28 |
| ✓ | 42.1 | 62.5 | 5.73 | 41.6 | 115.6 | 3.80 | 97.3 | 230.9 | 1.67 | 158.9 | 223.2 | 2.17 |

### E.3 TRAINING-FREE GUIDANCE BENCHMARK

Here, we provide experimental details that we follow in Section 4.1. We mainly follow TFG benchmark (Ye et al., 2024) for the fair comparison. All of the experiments in this subsection utilize training-free guidance as a conditional guidance as stated in B.3.

#### E.3.1 LABEL GUIDANCE

**Task description**   Label guidance target to generate designated label condition with only unconditional diffusion models.

**Dataset**   Two experiments are conducted using two image dataset: CIFAR10 (Krizhevsky et al., 2009) and ImageNet (Russakovsky et al., 2015).

**Evaluation**   Following the image generation literature, we measure FID (Heusel et al., 2017) to assess fidelity and use accuracy to evaluate generation validity, defined as the proportion of generated samples classified as the target label. In other words, we measure:

$$p(\arg\max \rho(\mathbf{x}) = \mathbf{c}_{\text{target}}), \tag{115}$$

where $\rho$ denotes a classifier and $\mathbf{c}_{\text{target}}$ refers to the target label.

For CIFAR 10, we average the result across all 10 targets. For ImageNet, following (Ye et al., 2024), we randomly take different target values and report the average value across the selected target. We set the sample sizes to 512 for CIFAR10 and 256 for ImageNet in Table 2, while using 128 samples for ImageNet in the rest of the experiments. We note that the use of fewer evaluation samples is the primary reason for initially higher FIDs, which might consequently lag CFG SOTA. In Table 13, we present standard comparisons on CIFAR-10 using 50,000 samples that yielded improved and benchmark-consistent scores.

Table 13: Originally, 512 samples were used for rapid, extensive experiments across various tasks. For a more rigorous evaluation, we used 50,000 samples on CIFAR-10 with 100 inference steps. As expected, increasing the number of samples to match standard benchmark protocols led to improved FID scores.

| Method | FID↓ | Acc. ↑ |
|---|---|---|
| *512 samples* | | |
| TFG | 114.1 | 55.8 |
| TFG + TAG (ours) | 102.7 | 61.5 |
| *50000 samples* | | |
| TFG | 77.5 | 54.3 |
| TFG + TAG (ours) | **47.1** | **84.4** |

**Models**   For backbone diffusion models, DDPM in Nichol & Dhariwal (2021) is utilized for both CIFAR-10 and ImageNet.

### E.3.2 GUASSIAN DEBLURRING

**Task description**  Gaussian deblurring task aims to restore the noisy images which are blurred by a Guassian process. This inverse problem has been extensively studied with diffusion models and notably, DPS (Chung et al., 2023) utilize training free guidance when given the blurring operator:

$$\mathbf{y} = \mathcal{A}_{\text{blur}}(\mathbf{x}). \tag{116}$$

We set the loss objective function $\ell_{\mathbf{c}}$ in Eq. 29 as $l_2$ norm between the blurred estimates and the target,

$$\ell_{\mathbf{c}} = \|\mathcal{A}_{\text{blur}}(\mathbf{x}) - \mathbf{y}\|_2. \tag{117}$$

**Dataset**  Cat images (Elson et al., 2007) is utilized for the diffusion model training with resolution $256 \times 256$.

**Evaluation**  We measure FID score for the sample fidelity and LPIPIS (Zhang et al., 2018) for evaluating conditioning effects.

### E.3.3 SUPER-RESOLUTION

**Task description**  Super-resolution targets to upscale the originally lower-resolution images to the higher resolution images. Previous works (Saharia et al., 2022; Ho et al., 2022) show one can leverage diffusion models for this task. In super-resolution case, we assume having an downgrade operator $\mathcal{A}_{\text{down}}$. With the operator we suppose low-resolution images $\mathbf{y}$ is obtained from a higher resolution image $\mathbf{x}$ by

$$\mathcal{A}_{\text{down}} : \mathbb{R}^{256 \times 256 \times 3} \to \mathbb{R}^{64 \times 64 \times 3}, \ \mathbf{y} = \mathcal{A}_{\text{down}}(\mathbf{x}). \tag{118}$$

Now, by setting following loss objective function in Eq. 29:

$$\ell_{\mathbf{c}} = \|\mathcal{A}_{\text{down}}(\mathbf{x} - \mathbf{y})\|_2, \tag{119}$$

we leverage training free guidance to restore the target high-resolution image.

**Dataset**  Cat images Elson et al. (2007) is utilized for the diffusion model training with resolution $256 \times 256$.

**Evaluation**  FID is used for the sample fidelity and LIPIPS is used for evaluating conditioning effects. We set the sample size as 256 for the result in the Table 2 and 128 for all the other experiments.

### E.3.4 MULTI-CONDITIONAL GUIDANCE

**Task Description**  Multi-conditional guidance leverages multiple target functions to guide a single sample towards multiple attribute-based targets. We explore two scenarios: (gender, hair color) and (gender, age). Each attribute has two labels: Gender: {male, female}, Age: {young, old}, Hair color: {black, blonde}.

Following Ye et al. (2024), we sampled images that maximize the marginal probability:

$$\max_{\mathbf{x}_0} p_{\text{combined}}(\mathbf{x}_0) = \max_{\mathbf{x}_0} p_{\text{target1}}(\mathbf{x}_0) p_{\text{target2}}(\mathbf{x}_0), \tag{120}$$

where $p_{\text{target}}(\mathbf{x}_0)$ is estimated using label guidance. However such a naive approach of summing score functions for each condition, as in Eq. 5, can lead to off-manifold artifacts.

**Dataset**  Experiments are conducted on CelebA-HQ (Karras et al., 2018) at a resolution of $256 \times 256$.

**Evaluation**  We assess sample fidelity using Kernel Inception Distance (KID) (Bińkowski et al., 2018), with 1,000 randomly sampled CelebA-HQ images as references. The KID(log) scores are reported in Section 4.

For validity, we compute classification accuracy using three independent attribute classifiers, evaluating the conjunction of target attributes:

$$\text{Accuracy} = \frac{\# \bigwedge_{\text{target label}}(\text{classified as target label})}{\#\text{generated samples}}. \tag{121}$$

We set the sample size as 256 across all experiments.

**Models** We use the CelebA-DDPM model, trained on CelebA-HQ, as the base diffusion model. Binary classifiers are employed for attribute validation.

### E.3.5 MOLECULAR GENERATION

**Task description** The goal of molecular generation in this work is to guide 3D molecules generated from unconditional diffusion models to the deisred quantum chemical properties (Hoogeboom et al., 2022). Utilizing the property predictor $\mathcal{A}_{\text{property}}$ is trained for each quantum chemical property,

$$\mathcal{A}_{\text{property}} : \mathbb{R}^d \to \mathbb{R}, \ \mathcal{A}_{\text{property}}(\mathbf{x}) = c. \tag{122}$$

Then, we set the training-free guidance objective function $\ell_{\mathbf{c}}$ as a square of $l_2$ norm of the property gap as follows:

$$\ell_{\mathbf{c}} = \|\mathcal{A}_{\text{property}}(\hat{\mathbf{x}}_0) - c\|_2^2, \tag{123}$$

where $\hat{\mathbf{x}}_0$ is obtained from the Tweedie's formula (Appendix B.3).

**Dataset** We use QM-9 dataset (Ramakrishnan et al., 2014), which consists of 134k molecules with molecules having maximum 9 heavy atoms (C, N, O, F) labeled with 12 quantum chemical properties. The dataset is split into 130k / 18k / 13k molecules of training, valid, test data following (Hoogeboom et al., 2022). Following previous works (Hoogeboom et al., 2022; Bao et al., 2023; Xu et al., 2023), we take 6 quantum chemical properties as a target property where we describe detils in the following.

- **Polarizability** ($\alpha$): The extent to which a molecule's electron cloud can be distorted by an external electric field.
- **HOMO-LUMO gap** ($\Delta\epsilon$): The energy difference between the highest occupied and lowest unoccupied molecular orbitals, signifying possible electronic transitions.
- **HOMO energy** ($\epsilon_{\text{HOMO}}$): The energy of the highest occupied orbital, often linked to how easily a molecule donates electrons.
- **LUMO energy** ($\epsilon_{\text{LUMO}}$): The energy of the lowest unoccupied orbital, often linked to how readily a molecule can accept electrons.
- **Dipole moment** ($\mu$): A numerical measure of charge separation within a molecule, reflecting its polarity.
- **Heat capacity** ($C_v$): The amount of heat required to change the temperature of a molecule by a given amount.

**Models** We utilize unconditional EDM from Hoogeboom et al. (2022) for the backbone diffusion model which consists of EGNN (Satorras et al., 2021). For the property prediction, we utilize EGNN backbone architecture as in (Bao et al., 2023) where each specialized prediction model which outputs scalar value is used.

**Evaluation** We evaluate Atom Stability (AS) which measures percentage of atoms within generated molecules that have right valencies. An atom is stable if its total bond count matches the expected valency for its atomic number (Hoogeboom et al., 2022). To measure conditioning effect, we calculate Mean Absolute Error (MAE) values between the target condition and predicted condition. We set the sample size as 4096 for the result in the Table 2 and 1024 for all the other experiments.

### E.3.6 AUDIO GENERATION

**Task description** We conduct experiments for two types of tasks with audio diffusion models: Audio declipping and Audio inpainitng (Moliner & Välimäki, 2024; Moliner et al., 2023). Audio declipping is a process that repairs distorted audio signals, specifically addressing the issue of clipping. Clipping occurs when the audio signal's intensity surpasses the limits of the recording system, resulting in a distorted sound with missing portions of the waveform. Audio inpainting is a technique used to reconstruct missing or damaged parts of an audio signal.

For the declipping test, we assume having clipping operator $\mathcal{A}_{\text{blur}}$ which corrupts mel spectrograms (Shen et al., 2018) as follows.

$$\mathcal{A}_{\text{clip}} : \mathbb{R}^{256 \times 256} \to \mathbb{R}^{256 \times 256}, \ \mathcal{A}_{\text{clip}}(\mathbf{x}) = \mathbf{y}, \tag{124}$$

where clipping operator is operated by zeroing out the 40 highest dimensions and zeroing out the lowest 40 dimensions in terms of the frequency values.

For inpainting task, we assume we have blurring operator $\mathcal{A}_{\text{blur}}$ as

$$\mathcal{A}_{\text{blur}} : \mathbb{R}^{256 \times 256} \to \mathbb{R}^{256 \times 256}, \ \mathcal{A}_{\text{blur}}(\mathbf{x}) = \mathbf{y}, \tag{125}$$

where deblurring is conducted by zeroing out the values of middle 80 dimensions in the mel sepctrograms.

For both tasks, we set the training-free guidance objective function $\ell_{\mathbf{c}}$ with $l_2$ norm.

$$\ell_{\mathbf{c}} = \|\mathcal{A}_{\text{clip}}(\hat{\mathbf{x}}_0) - \mathbf{y}\|_2, \ \ell_{\mathbf{c}} = \|\mathcal{A}_{\text{blur}}(\hat{\mathbf{x}}_0) - \mathbf{y}\|_2. \tag{126}$$

**Dataset** We borrow open-source training data of Audio-DDPM [1] following Ye et al. (2024).

**Evaluation** For both tasks, we use Frechet Audio Distance (FAD) (Kilgour et al., 2018) for measure how close the generated data is from the original distribution and Dynamic Time Warping (DTW) (Müller, 2007) for evaluating how generated samples are derived into the desired conditions. We set the sample size as 256 across all experiments.

---

[1]https://huggingface.co/teticio/audio-diffusion-256

### E.3.7 HYPER-PARAMETERS

In Table 14, we provide hyper-parameter settings for the DPS and TFG where we follow the optimal reported values in Ye et al. (2024).

Table 14: Parameter table $(\bar{\rho}, \bar{\mu}, \bar{\gamma})$ DPS, TFG for all methods, tasks, and targets.

| | DPS | | | TFG | | |
|---|---|---|---|---|---|---|
| Target | $\bar{\rho}$ | $\bar{\mu}$ | $\bar{\gamma}$ | $\bar{\rho}$ | $\bar{\mu}$ | $\bar{\gamma}$ |
| **CIFAR-10 label guidance** | | | | | | |
| 0 | 1 | 0 | 0 | 1 | 2 | 0.001 |
| 1 | 8 | 0 | 0 | 0.25 | 2 | 0.001 |
| 2 | 1 | 0 | 0 | 2 | 0.25 | 1 |
| 3 | 4 | 0 | 0 | 4 | 0.25 | 0.01 |
| 4 | 0.5 | 0 | 0 | 1 | 0.5 | 0.001 |
| 5 | 4 | 0 | 0 | 2 | 0.25 | 0.001 |
| 6 | 1 | 0 | 0 | 0.25 | 0.5 | 1 |
| 7 | 2 | 0 | 0 | 1 | 0.5 | 0.001 |
| 8 | 2 | 0 | 0 | 1 | 0.25 | 0.001 |
| 9 | 4 | 0 | 0 | 0.5 | 2 | 0.001 |
| **ImageNet label guidance** | | | | | | |
| 111 | 2 | 0 | 0 | 2 | 0.5 | 0.1 |
| 222 | 2 | 0 | 0 | 0.5 | 1 | 0.1 |
| 333 | 2 | 0 | 0 | 1 | 4 | 1 |
| 444 | 4 | 0 | 0 | 0.5 | 2 | 0.1 |
| **Fine-grained guidance** | | | | | | |
| 111 | 0.25 | 0 | 0 | 0.5 | 0.5 | 0.01 |
| 222 | 0.25 | 0 | 0 | 0.5 | 0.5 | 0.01 |
| 333 | 0.25 | 0 | 0 | 0.5 | 0.5 | 0.01 |
| 444 | 0.25 | 0 | 0 | 0.5 | 0.5 | 0.01 |
| **Combined Guidance (gender & hair)** | | | | | | |
| (0,0) | 4 | 0 | 0 | 1 | 2 | 0.01 |
| (0,1) | 4 | 0 | 0 | 2 | 8 | 0.01 |
| (1,0) | 4 | 0 | 0 | 1 | 1 | 0.01 |
| (1,1) | 2 | 0 | 0 | 0.5 | 1 | 0.1 |

| | DPS | | | TFG | | |
|---|---|---|---|---|---|---|
| Target | $\bar{\rho}$ | $\bar{\mu}$ | $\bar{\gamma}$ | $\bar{\rho}$ | $\bar{\mu}$ | $\bar{\gamma}$ |
| **Combined Guidance (gender & age)** | | | | | | |
| (0,0) | 8 | 0 | 0 | 1 | 2 | 0.01 |
| (0,1) | 1 | 0 | 0 | 0.5 | 8 | 1 |
| (1,0) | 4 | 0 | 0 | 0.5 | 2 | 0.01 |
| (1,1) | 2 | 0 | 0 | 1 | 0.5 | 0.1 |
| **Super-resolution** | | | | | | |
| | 16 | 0 | 0 | 4 | 2 | 0.01 |
| **Gaussian Deblur** | | | | | | |
| | 16 | 0 | 0 | 1 | 8 | 0.01 |
| **Molecule Property** | | | | | | |
| $\alpha$ | 0.005 | 0 | 0 | 0.016 | 0.001 | 0.0001 |
| $\mu$ | 0.02 | 0 | 0 | 0.001 | 0.002 | 0.1 |
| $C_v$ | 0.005 | 0 | 0 | 0.004 | 0.001 | 0.001 |
| $\epsilon_{\text{HOMO}}$ | 0.005 | 0 | 0 | 0.002 | 0.004 | 0.001 |
| $\epsilon_{\text{LUMO}}$ | 0.005 | 0 | 0 | 0.016 | 0.002 | 0.0001 |
| $\Delta$ | 0.005 | 0 | 0 | 0.032 | 0.001 | 0.001 |
| **Audio Declipping** | | | | | | |
| | 1 | 0 | 0 | 1 | 1 | 0.1 |
| **Audio Inpainting** | | | | | | |
| | 16 | 0 | 0 | 0.25 | 2 | 0.1 |

### E.4 TIME PREDICTOR

**Architecture** Time Predictor is a foundational component of our TAG framework, designed to estimate $p(t|\mathbf{x}_t)$ or $p(t|\mathbf{x}_t, \mathbf{c})$ for guiding noisy samples back to the desired data manifold. Its architecture is tailored to the input modality.

For image and audio data, a *SimpleCNN* is employed, comprising four convolutional layers with channel sizes $(32, 64, 128, 256)$, each followed by ReLU activation and average pooling. This design is significantly lighter than the diffusion backbone. A final linear layer produces logits for all timesteps. In conditional settings, learned embedding vectors for conditions are concatenated before the linear layer to model $p(t|\mathbf{x}_t, \mathbf{c})$.

For molecular data, a modified *Equivariant Graph Neural Network (EGNN)* Satorras et al. (2021) processes node and edge features along with spatial coordinates. The concatenated node and spatial features are passed through a feed-forward network to output logits representing the time distribution.

These architectures are lightweight compared to the diffusion model backbone yet expressive enough to capture temporal and conditional relationships, involving minimal computational cost during sample generation. We present the performance analysis in next subsection.

Following Jung et al. (2024), training involves minimizing a cross-entropy loss between the true time step $t$ and the predicted distribution over timesteps. For each sample $\mathbf{x}_0$ from the data distribution, a noisy version $\mathbf{x}_t$ is generated at a random $t$ using the forward process. The objective is:

$$\mathcal{L}_{\text{time-predictor}}(\phi) = -\mathbb{E}_{t,\mathbf{x}_0}\left[\log\left(\hat{\mathbf{p}}_\phi(\mathbf{x}_t)_t\right)\right], \tag{127}$$

where $\hat{\mathbf{p}}_\phi(\mathbf{x}_t)_t$ is the predicted probability for $t$. Cross-entropy is chosen over regression due to overlapping supports of $p_t(\mathbf{x})$ and $p_s(\mathbf{x})$, ensuring ambiguity is handled probabilistically. The model is trained using the Adam optimizer (learning rate $1 \times 10^{-4}$) for 300K iterations on most datasets, except for ImageNet, which uses 600K iterations. The batch sizes and GPU configurations for each dataset are summarized in Table 15.

Table 15: Training Details for the Time Predictor

| Dataset | Batch Size | Training Iterations | A100 GPUs |
|---------|-----------|---------------------|-----------|
| ImageNet | 1024 | 600K | 4 |
| CIFAR10 | 256 | 300K | 1 |
| CelebAHQ | 256 | 300K | 1 |
| Cat | 128 | 300K | 1 |
| Molecule | 128 | 300K | 2 |
| Audio | 128 | 300K | 1 |

**Performance**   We compare the performance of time predictors across diverse datasets and tasks. The *time gap* (Def. F.1) is presented in the Appendix F.2, where we evaluate its behavior across different datasets and tasks. Given true forward noise samples $\boldsymbol{x}_t \sim q(\boldsymbol{x}_t|\boldsymbol{x}_0)$, we measure the *time gap*, where a lower value indicates higher prediction accuracy.

The results presented in Figure F.2 demonstrate that the time predictor achieves strong performance across most datasets and tasks, despite employing a relatively simple CNN architecture. Notably, for timesteps $t < 600$, nearly all models accurately predict the true timestep. However, for lower-dimensional datasets such as CIFAR-10 and molecular data, the prediction error increases as $t$ approaches the final timestep $T$ of the diffusion process, indicating degraded performance. This observation aligns with the findings of Kahouli et al. (2024), reported that higher data dimensionality enhances predictability, whereas overlapping distributions near $T$ impede accurate predictions. Consistent with these observations, our results indicate that the some model struggles in this regime, which we leave as an avenue for future work.

The performance of TAG improves with a better classifier, as it provides a more accurate estimate of the true TLS. We conducted experiments using different training checkpoints (10K and 30K).Table 16 shows that performance on all metrics improved at the 30K checkpoint, correlating with the better performance of the more trained classifier.

Table 16: Quantitative evaluation of TFG+TAG across varying training steps on CIFAR-10 confirms the relationship between classifier robustness and TAG performance.

| | Training Steps | |
|---|---|---|
| | 10 K | 30 K |
| FID $\downarrow$ | 116.0 | **102.7** |
| Acc. $\uparrow$ | 55.3 | **61.5** |

### E.5   LARGE-SCALE TEXT-TO-IMAGE GENERATION

**Enhanced Reward Alignment**   All reward alignment experiments build on the DAS test-time sampler (Kim et al., 2025) with Stable Diffusion v1.5 (Rombach et al., 2022) as the base model. Unless stated otherwise, we follow the hyperparmeter setting in DAS (Kim et al., 2025). We evaluate with two settings:

Single-objective alignment: We optimize two separate reward functions: the LAION Aesthetic predictor (Schuhmann et al., 2022) using 256 simple animal prompts from ImageNet (Russakovsky et al., 2015), and CLIPScore (Radford et al., 2021) using the HPSv2 prompt set (Wu et al., 2023).

Multi-objective alignment: We combine aesthetic and CLIP rewards via

$$r_{\text{total}}(x) = w\, r_{\text{Aesthetic}}(x) + (1 - w)\, r_{\text{CLIP}}(x), \quad w = 0.5.$$

In all experiments we use $T = 100$ diffusion steps, single particle setting, and the tempering schedule $\lambda_t = (1 + \gamma)^{t-1}$ with $\gamma = 0.008$ following original setting for the fair comparison. Resampling is triggered when the effective sample size ESS $< 0.5$. We set the KL coefficient $\alpha = 0.01$ for aesthetic alignment and $\alpha = 0.001$ for CLIP alignment (single-objective), and $\alpha = 0.005$ for multi-objective trials. For each prompt set, we sample 256 images and report the mean reward and mean Time-Gap (Def. F.1) over three independent runs.

**Improved Style Transfer**    We adopt the setup of (Ye et al., 2024). Our goal is to steer the latent diffusion model Stable-Diffusion-v-1-5 (Rombach et al., 2022) so that the generated images both match the input text prompts and reflect the style of given reference images. We achieve this by matching the Gram matrices (Johnson et al., 2016) of intermediate features from a CLIP image encoder for the generated and reference images.

Specifically, let $x_{\text{ref}}$ be a reference style image and $D(z_{0|t})$ be the decoded image obtained from the estimated latent $z_{0|t}$. We extract features from the third layer of the CLIP image encoder and compute their Gram matrices

$$G(x_{\text{ref}}), \quad G\big(D(z_{0|t})\big)$$

following the methodology of MPGD (He et al., 2024) and FreeDoM (Yu et al., 2023). The style-guidance objective maximizes

$$\exp\big(-\| G(x_{\text{ref}}) - G(D(z_{0|t})) \|_F^2\big),$$

where $\| \cdot \|_F$ denotes the Frobenius norm.

We measure style transfer quality using the Style Score and CLIP Score. As reference styles, we use the same four WikiArt images employed by MPGD (He et al., 2024), and for text prompts we select 64 samples from Partiprompts (Yu et al., 2022). For each style, we generate 64 images. To prevent inflated CLIP scores, we compute guidance and evaluation with two different CLIP models from the Hugging Face Hub, Guidance[2] and Evaluation[3]. Throughout our experiments, we fix the guidance strength at $\omega_t = 1$. We leave exploring hyperparameter tuning to improve results as a future work.

# F    ABLATION STUDIES

## F.1    FEW STEP UNCONDITIONAL GENERATION

In few step generation experiments in Section 4.3, we study on the effect of TAG in unconditional generation scenario where no extra guidance is applied in reverse diffusion process. Specifically, we focus on few step generation scenario where discretization error happens as introduced in Appendix B.5.

We further report the evaluation results with 50,000 samples in Table 17 where number of function evaluation (NFE) refers to how many times we evaluate with diffusion models during the reverse process. The result shows that TAG significantly improves FID and IS scores when less evaluation steps are used which alings with our intuition that fewer NFE induces more severe off-manifold phenomenon in reverse diffusion process. For the experiment, we utilize CIFAR-10 DDPM Song & Dhariwal (2024) as in Section 4.3 and use DDPM sampling.

Table 17: Comparison of FID values before and after applying TAG in unconditional generation scenario.

|  | NFE 1 | | NFE 3 | | NFE 5 | |
|---|---|---|---|---|---|---|
| TAG | FID$\downarrow$ | IS$\uparrow$ | FID$\downarrow$ | IS$\uparrow$ | FID$\downarrow$ | IS$\uparrow$ |
| ✗ | 449.8 | 1.26 | 194.5 | 2.04 | 116.5 | 3.08 |
| ✓ | 232.9 | 2.26 | 124.2 | 3.55 | 97.4 | 3.66 |

---

[2]Guidance: openai/clip-vit-base-patch16
[3]Evaluation: openai/clip-vit-base-patch32

### F.2 TIME GAP

**Time-Gap (TG)** To quantify the temporal deviation during generation, we define the Time-Gap metric. Denoting the sample at timestep $t$ as $\mathbf{x}_t$ and the time predictor as $\phi$, the Time-Gap is the average absolute difference between the predicted timestep index and the true index:

**Definition F.1** (Time-Gap).

$$\text{Time-gap} := \frac{1}{T} \sum_{t=1}^{T} |\arg\max \phi(\mathbf{x}_t) - t|. \tag{128}$$

A lower Time-gap indicates samples are closer to their expected temporal manifold.

**Time Gap across different timesteps** To further identify how time gap varies across different diffusion timesteps, we conduct an ablation study where we measure time gap for every timestep in diffusion models. For each step, average time gap value over 512 samples are reported.

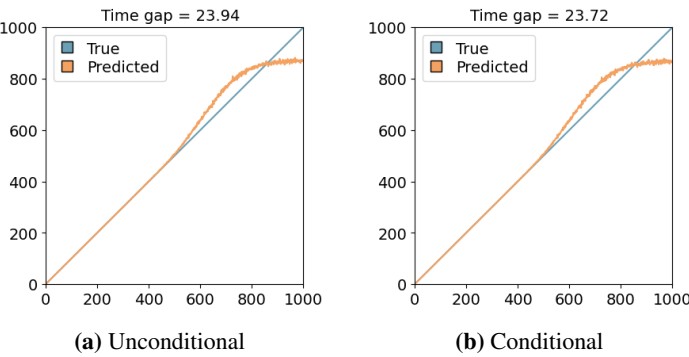

**(a)** Unconditional      **(b)** Conditional

**Figure 7:** Time gap in CIFAR10.

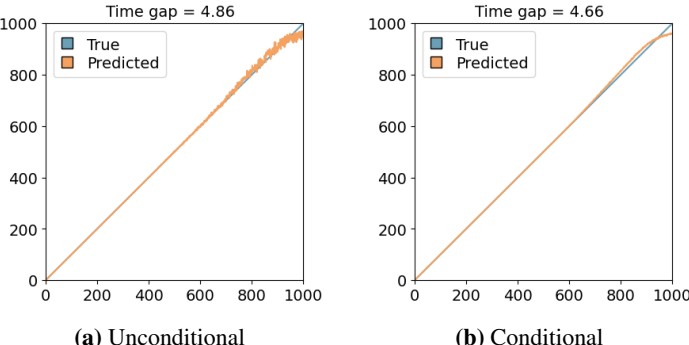

**(a)** Unconditional      **(b)** Conditional

**Figure 8:** Time gap in ImageNet.

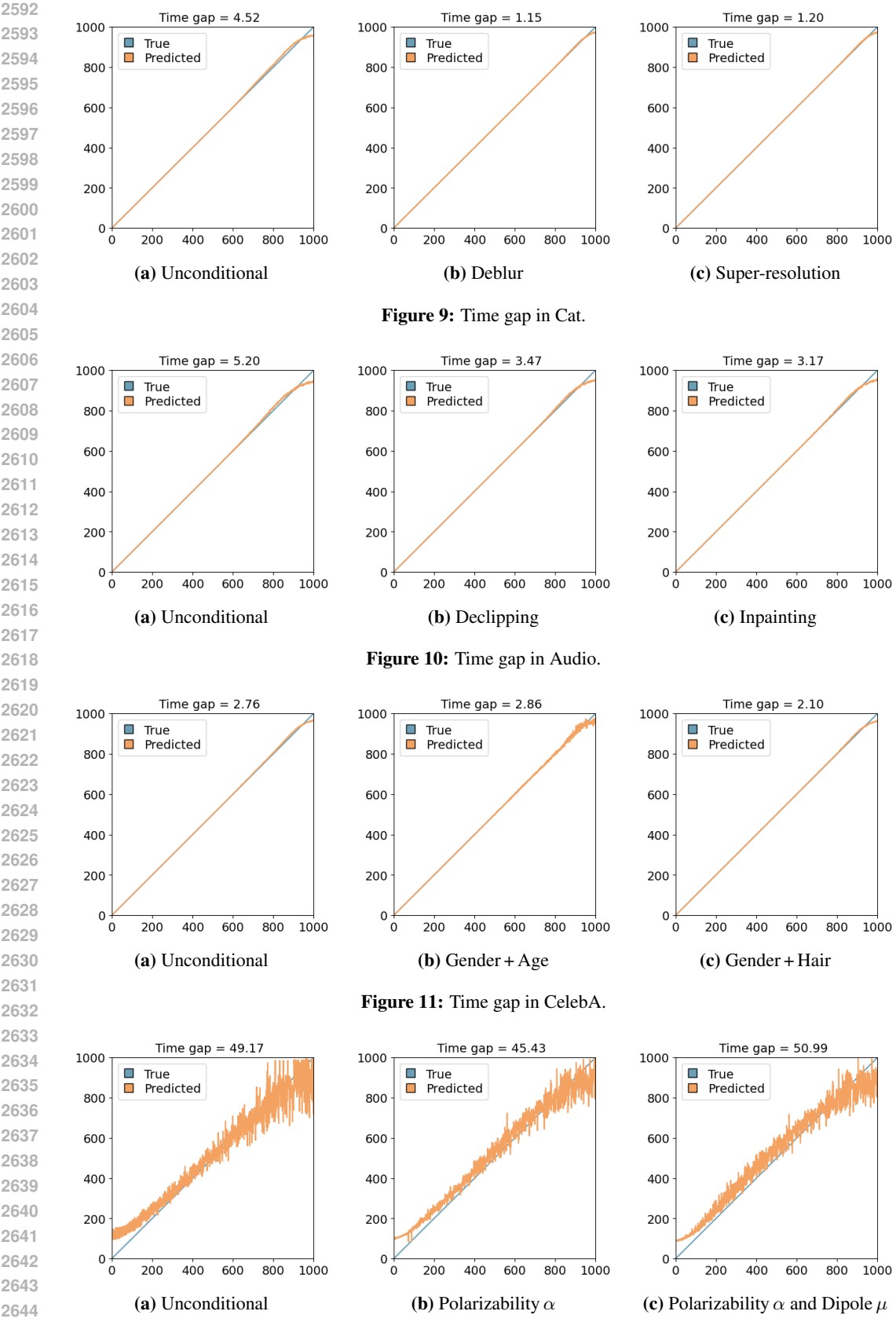

**Figure 9:** Time gap in Cat.

**Figure 10:** Time gap in Audio.

**Figure 11:** Time gap in CelebA.

**Figure 12:** Time gap in Molecule.

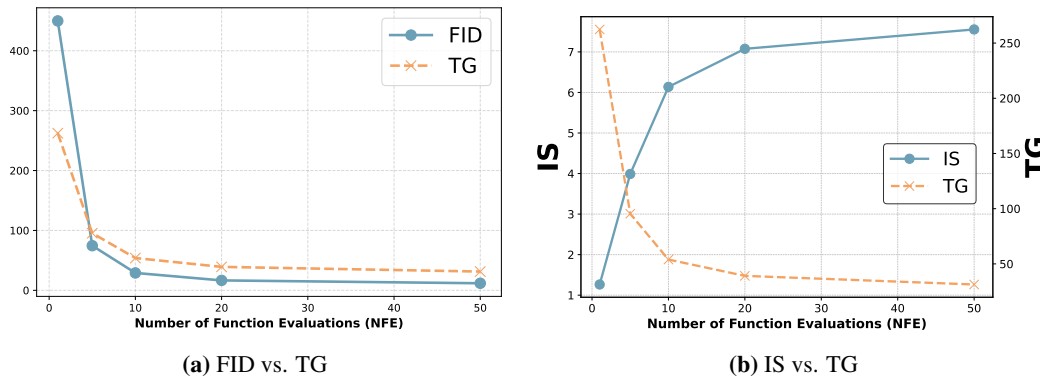

(a) FID vs. TG  (b) IS vs. TG

**Figure 13:** Correlation between time gap and standard metrics for image generation quality.

**Network Architecture**   To assess the trade-off between model size and time-gap accuracy, we compare two backbones. Our SimpleCNN consists of four convolutional blocks with channel widths $(32, 64, 128, 256)$. Each block uses a $3 \times 3$ convolution, ReLU activation, and $2 \times 2$ average pooling. At just 1.48 M parameters—8.5 % of the 17.38 M-parameter UNet encoder (Dhariwal & Nichol, 2021)—SimpleCNN matches its time-gap performance (Table 18). This demonstrates that a lightweight, single-path network can rival much larger UNet based classifiers.

Table 18: FID on CIFAR-10 for time predictors using SimpleCNN (1.48 M parameters) and UNet encoder (17.38 M parameters), comparing unconditional and conditional models across training checkpoints.

| Checkpoint | SimpleCNN | | UNet | |
|---|---|---|---|---|
| | Unconditional | Conditional | Unconditional | Conditional |
| 50K | 24.19 | 23.64 | 25.59 | 21.85 |
| 100K | 23.24 | 27.33 | 22.08 | 19.68 |
| 200K | 23.11 | 22.64 | 22.58 | 20.53 |
| 300K | 22.93 | 21.11 | 24.40 | 22.49 |

**Correlation with other standard metrics**   We conduct ablation study on the correlation between Time Gap and standard metrics (FID and IS). Figure 13 illustrates how time gap and standard measures for image generation quality. We vary different number of function evaluation (NFE) in unconditional diffusion models. For the experiment, we generate 50,000 samples with DDIM sampling and utilize CIFAR10-DDPM model (Nichol & Dhariwal, 2021) with the NFE of 1, 5, 10, 20, 50. The result shows that as NFE increases, time gap reduces while FID decreases and IS increases. This demonstrates that Time Gap serves as a good measure to evaluate the off-manifold phenomenon.

**Limitation**   Once the reverse diffusion process is good enough (i.e, time gap is already small), it often loses correlation with FID measure. We believe improving the performance of the time predictor network will reduce this problem and thereby further boost the effect of the TAG.

We further note that the motivation of introducing a Time Gap in this work is not to suggest a new metric, but to quantify the amount of off-manifold phenomenon where applying TAG is intended to reduce the Time Gaps in each every timestep of the reverse diffusion process.

## G   VISUALIZATIONS OF GENERATED SAMPLES

Here, we present qualitative examples corresponding to the experiments presented in Section 4.1. We provide visualizations for all four experimental settings: DPS, DPS+TAG, TFG, and TFG+TAG. Below, we detail the dataset configurations used for generating these qualitative examples.

**CIFAR-10**  We generate images conditioned on the target class 8 (corresponding to the "Ship" category). The images are produced using 250 inference steps with an TAG strength of $\omega = 0.15$.

**ImageNet**  We present generated samples for target classes 111 ("Worm") and 222 ("Kuvasz"), using 100 inference steps with an TAG strength of $\omega = 0.15$.

**QM9**  We show qualitative results for the target molecular properties polarizability $\alpha$ and dipole moment $\mu$. In this setting, we employ a 0.1 guidance strength for DPS, following the default configuration in Ye et al. (2024), with 100 inference steps.

**CelebA-HQ**  We provide qualitative examples for two specific conditions: Gender+Hair and Gender+Age. The target attributes in these cases are black hair, young age, and female gender, all represented as binary variables to be satisfied in our conditional generation.

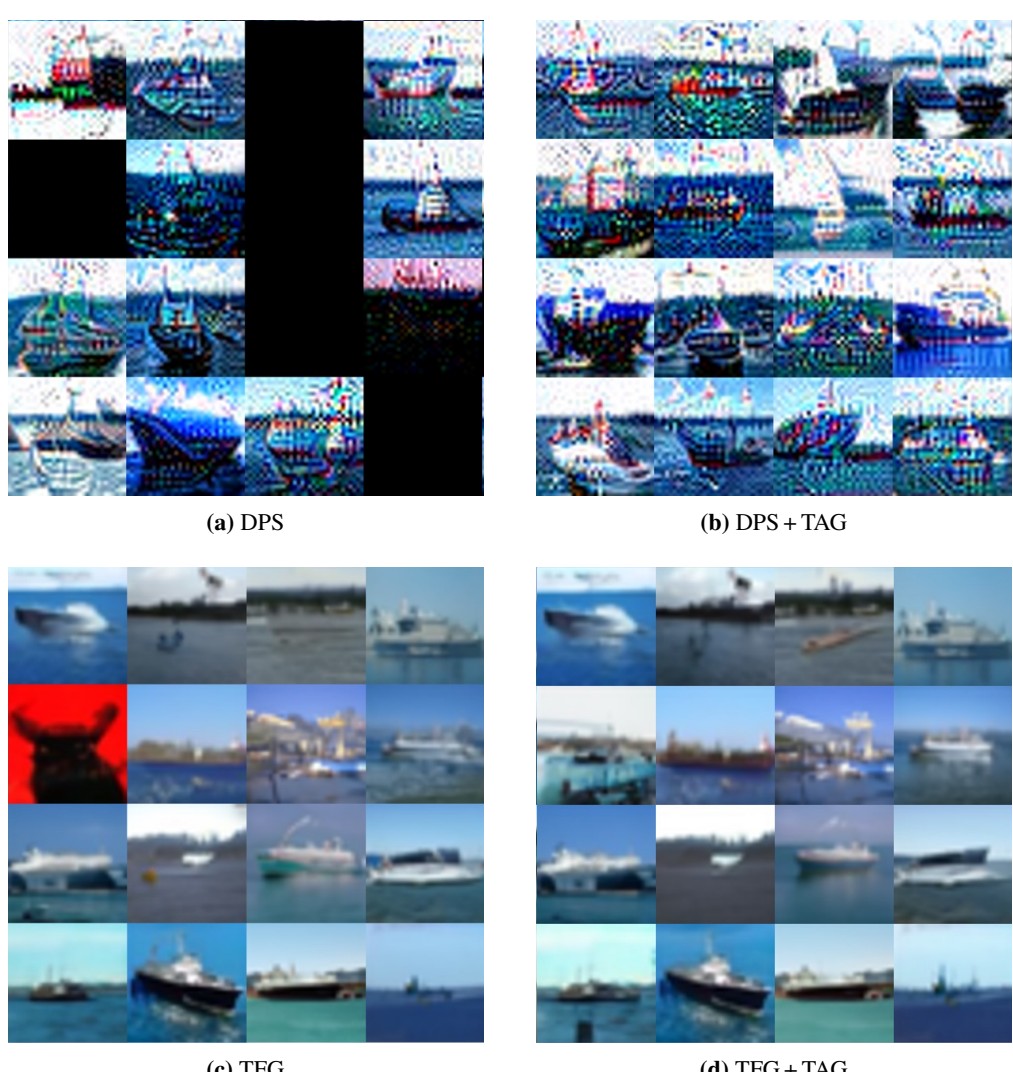

(a) DPS

(b) DPS + TAG

(c) TFG

(d) TFG + TAG

**Figure 14:** CIFAR10 with the target of 8 (Ship).

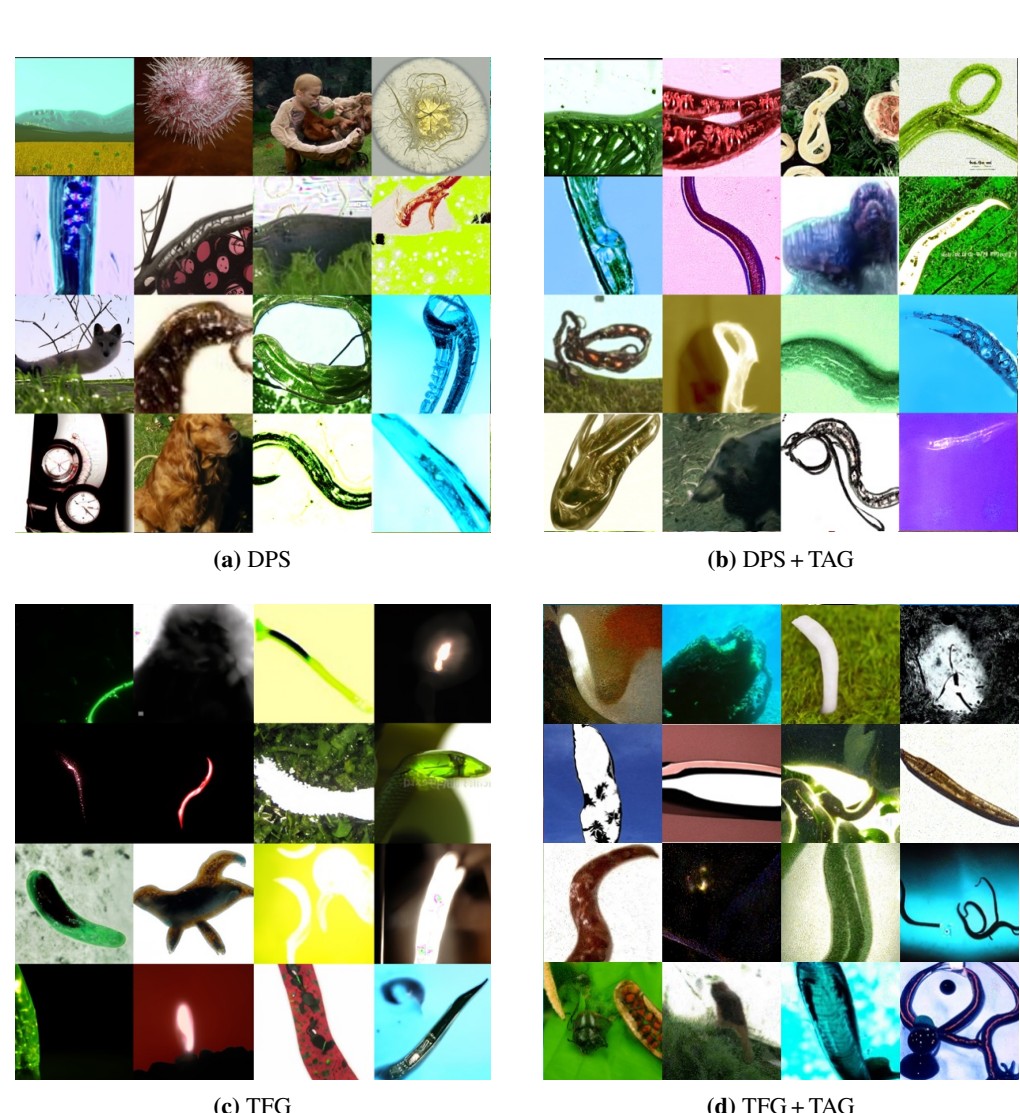

(a) DPS

(b) DPS + TAG

(c) TFG

(d) TFG + TAG

**Figure 15:** ImageNet with the target of 111 (Worm).

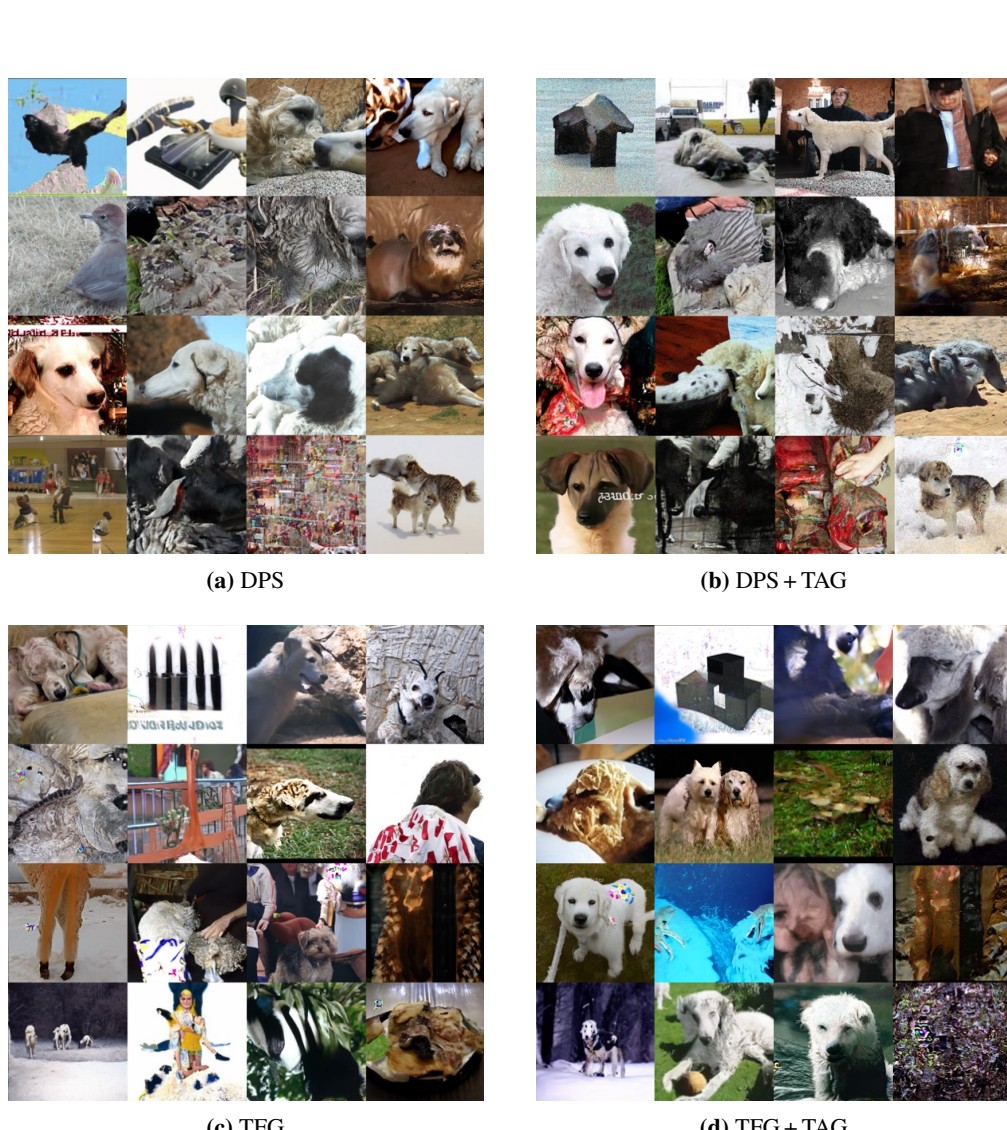

(a) DPS

(b) DPS + TAG

(c) TFG

(d) TFG + TAG

**Figure 16:** ImageNet with the target of 222 (Kuvasz).

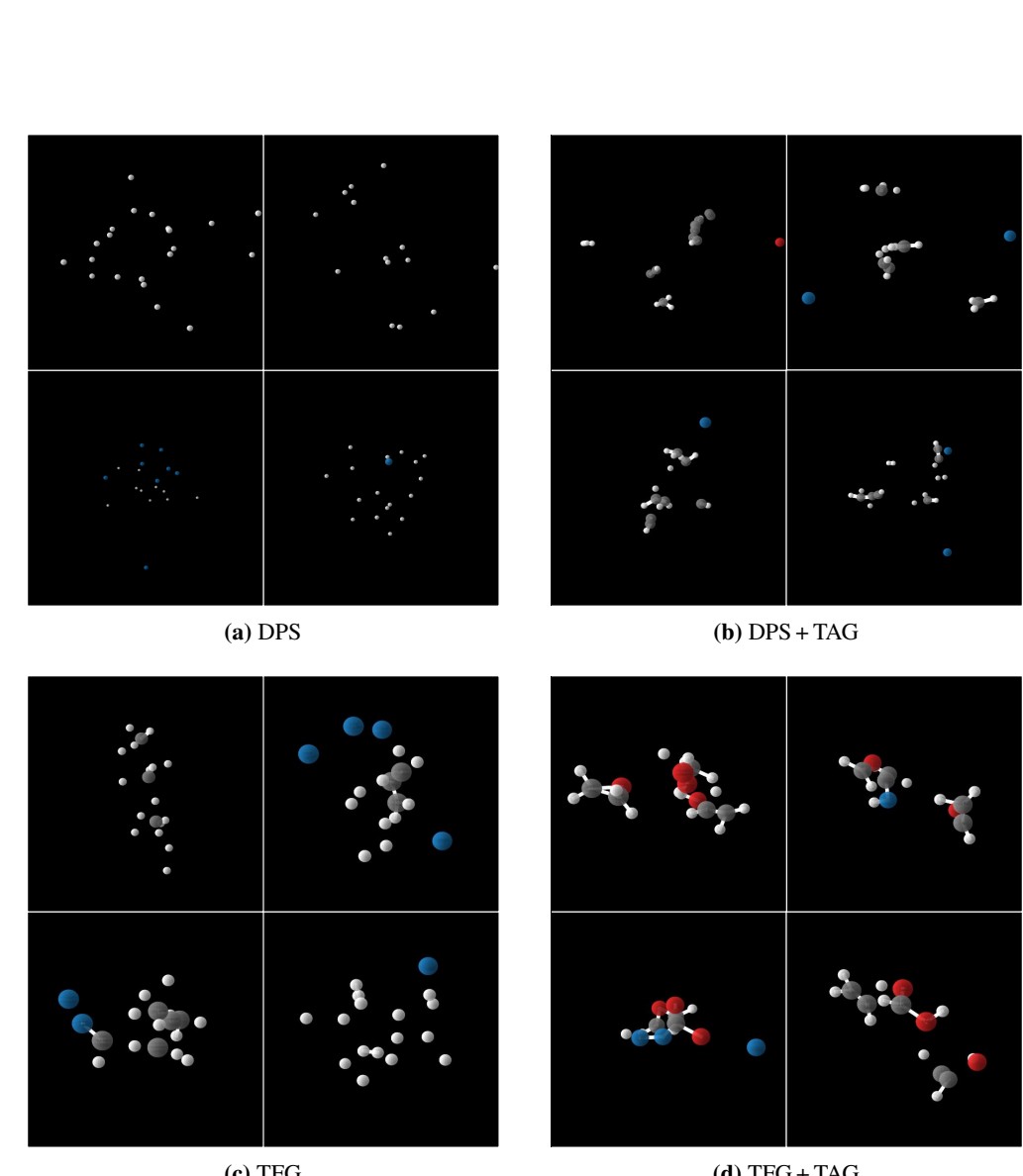

Figure 17: Molecule with condition of polarizability $\alpha$.

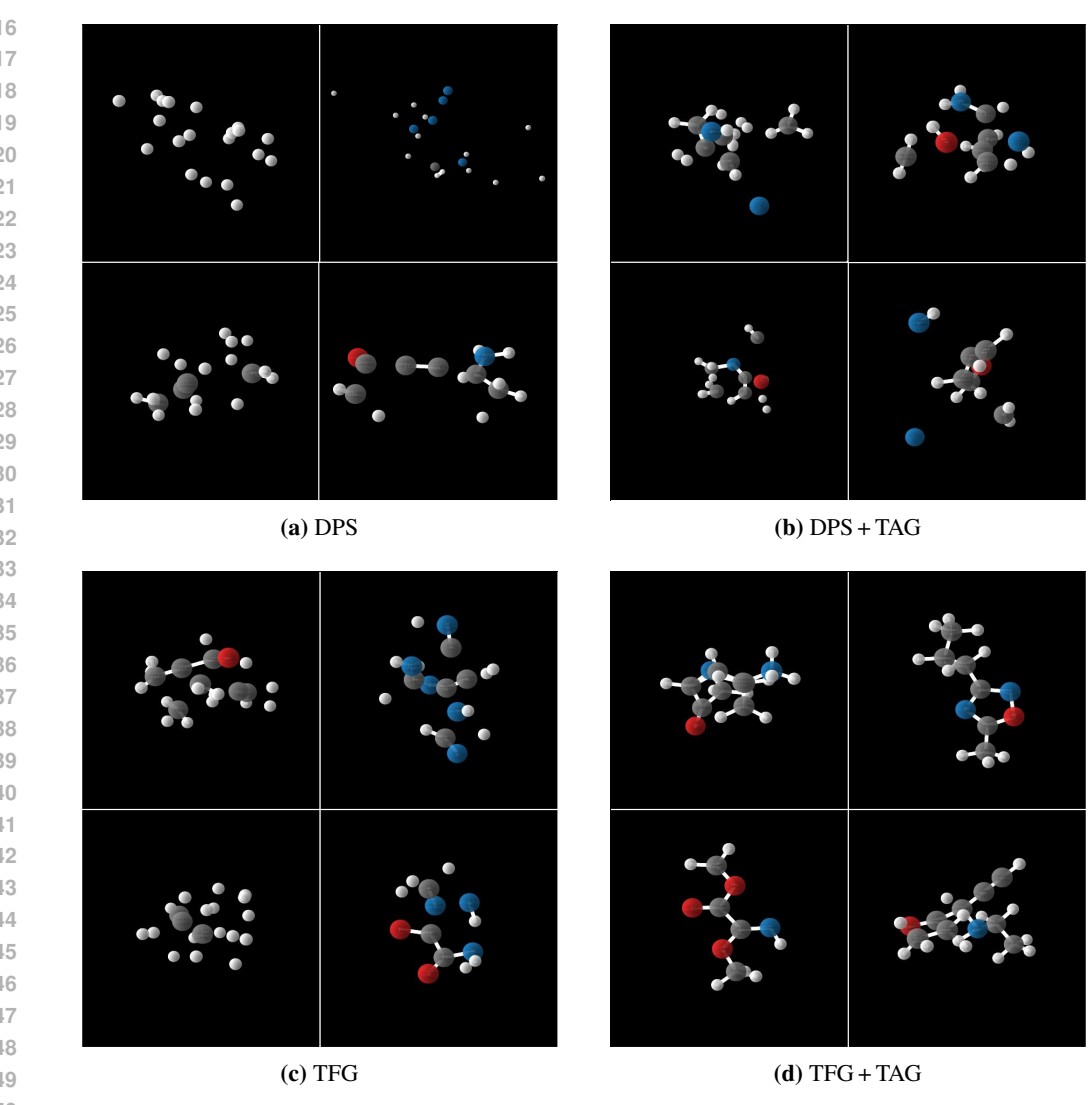

**(a)** DPS

**(b)** DPS + TAG

**(c)** TFG

**(d)** TFG + TAG

**Figure 18:** Molecule with condition of dipole $\mu$.

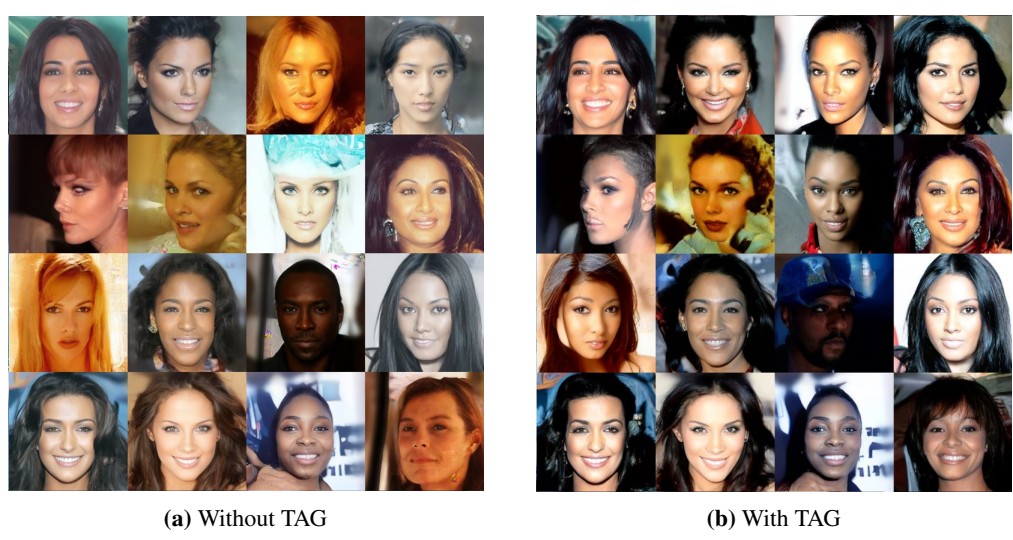

**(a)** Without TAG

**(b)** With TAG

**Figure 19:** CelebA with condition of Gender (female) + Hair (black hair).

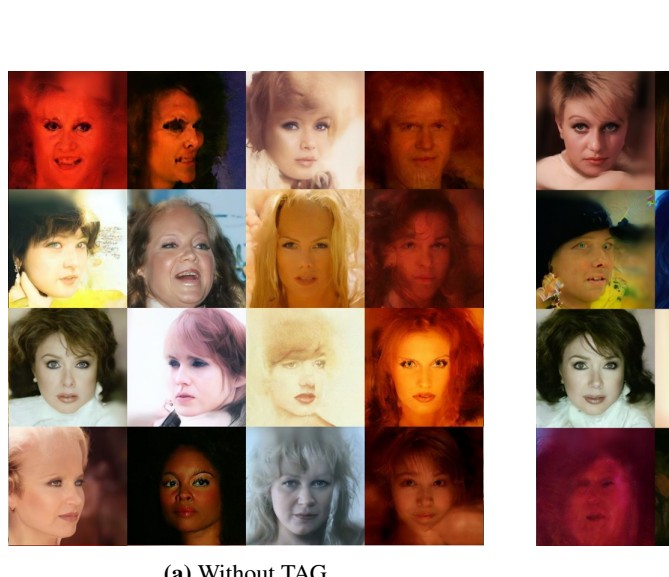 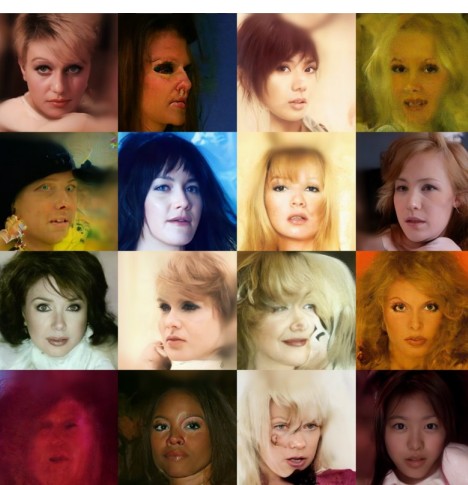

(a) Without TAG                 (b) With TAG

**Figure 20:** CelebA with condition of Gender (female) + Age (young).

