# OpenReview forum: "Temporal Alignment Guidance: On-manifold Sampling in Diffusion Models"
_ICLR.cc/2026/Conference — Submitted to ICLR 2026_

### Official Review · Reviewer_MJq2 · 2025-10-27

**Soundness:** 3
**Presentation:** 3
**Contribution:** 3
**Rating:** 6
**Confidence:** 3

**Summary:**

The paper proposes a novel method called Temporal Alignment Guidance (TAG), designed to address the off-manifold issue that arises when applying external guidance in diffusion models.
The approach introduces a Time Predictor that dynamically estimates the temporal deviation of samples from the true data manifold during generation, and employs a Time-Linked Score (TLS) to attract samples back toward high-density regions. TAG can be seamlessly integrated into existing diffusion models without retraining (training-free) and demonstrates significant improvements in both sample quality and stability across various domains, including image, molecular, and audio generation. Extensive experiments and detailed theoretical analyses show that TAG not only enhances generative fidelity, but also reduces temporal drift, suggesting its strong potential as a general-purpose guidance framework for diffusion models.

**Strengths:**

1. The paper introduces Temporal Alignment Guidance (TAG), which leverages a time predictor and time-linked score to effectively address the off-manifold issue in diffusion models under external guidance.
2. TAG operates in a training-free manner and can be seamlessly integrated as a plug-in into existing diffusion frameworks.
3. The authors provide theoretical analysis proving TAG’s ability to reduce generation error bounds, supported by extensive experiments across multiple domains that confirm its robustness and generality.

**Weaknesses:**

1. TAG employs a temporal gradient term to correct the sampling trajectory, but the paper does not provide an in-depth analysis of its error propagation or convergence boundaries within continuous-time dynamic systems.
2. The time guidance schedule $\omega$ in TAG relies on manually selected hyperparameters, without any adaptive update mechanism. As a result, the method still requires manual tuning across different noise levels, tasks, or model scales to achieve optimal performance.
3. In Section E.2, Table 10 shows that under a fixed noise schedule $\sigma=0.2$, as the guidance strength $\omega$ increases, both TG and FID values gradually decrease (ignoring the rebound of TG beyond $\omega$ = 40), while IS values steadily increase. (1) However, in Table 11 (lines 1997–1999), the parameter correspondence with Table 10 is unclear — in particular, for $\sigma=0.2$, the optimal parameter $\omega$ appears to be 45 instead of 4.5. The authors are advised to verify the correct parameter mapping. (2) In addition, since the results in Table 12 are almost identical to those in Table 11, consistency in numerical rounding and presentation should be ensured across both tables.
4. In Appendix G, the visual differences between methods are difficult to distinguish in several generated image comparisons (e.g., Figures 11–13 and 17). It is recommended that the authors include more representative samples or provide zoom-in crops to make the differences between TAG and other methods more visually discernible.
5. The paper does not include a formal analysis or quantitative discussion of the time complexity of TAG.
6. (1) In the Introduction, the sentence “This score approximation errors can accumulate over each timestep” contains a grammatical mismatch — the singular demonstrative “This” does not agree with the plural noun “errors.” (2) Lines 70–73 contain repetitive phrasing; removing redundancy would improve clarity and conciseness.

**Questions:**

Please see weakness.

---

> ### Author Response · Authors · 2025-11-23
> **Response to Reviewer MJq2 (1/3)**
>
> We are deeply grateful for your detailed review and for acknowledging the strengths of our work as follows:
>
> * Plug-and-play nature of TAG that can be seamlessly integrated into existing diffusion frameworks.
> * Provide theoretical analysis of generation error bound, aligned with experiments.
> * Robustness and generality of our approach with extensive experiments across multiple domains.
>
> Please note that all revisions made to address your concerns are marked in $\color{Crimson}{\text{red}}$ in the revised manuscript.
>
> We address your concerns below.
>
> ---
>
> ### [R4-1] TAG under continuous-time dynamics
> We thank the reviewer for the insightful comment regarding the theoretical analysis of TAG within continuous-time dynamic systems. We agree that establishing rigorous convergence boundaries is crucial.
>
> We clarify that a rigorous analysis of error propagation and convergence boundaries is already integral to our work (**Section 3.3, Appendix C**). **Theorem C.12** explicitly establishes the convergence boundary using the JKO scheme, proving that TAG tightens the upper bound of the Total Variation distance compared to the standard process $d\_{TV}(\tilde{p}\_{t\_0}, q\_0) \le d\_{TV}(p\_{t\_0}, q\_0) - \frac{G}{4\sqrt{F}}$ (F, G are appropriate constant detailed in Section C.7). Here, we leverage continuous time analysis including error dynamics including **Corollary C.6** which shows that the gradient flow of the KL divergence is accelerated by the term $A(t) = \sum_{i}\gamma_i \mathbb{E}[\tilde{r}_k \cdot s_i]$, effectively mitigating error accumulation. **Theorem 2 in Appendix C.5** further contains the natural generalization of TLS into continuous-time limit.
>
> We also empirically validate TAG in the continuous-time setting. To simulate this, we restrict the time predictor trained only in 25 diffusion steps and test whether it robustly generalize to more dense trajectory (100 steps). We implement the expected timestep $\mathbb{E}_{\phi}[t]$ to compute the loss for any continuous $t$. The result in **Table R4-1** shows a model trained on sparse discretization (25 steps) robustly generalizes to a dense trajectory (100 steps), outperforming the baseline and confirming TAG’s effectiveness in continuous dynamics.
>
> **Table R4-1.** Robustness of TAG with Continuous-Time Sampler (CIFAR-10)
> *Comparison of a model trained with sparse timesteps (25) evaluated on dense steps (100).*
>
> | Method | FID ↓ | Acc. ↑ | Time Gap |
> | :--- | :--- | :--- | :--- |
> | DPS | 255.5 | 46.4 | 70.7 |
> | DPS+TAG (full step) | 196.1 | 49.2 | 37.8 |
> | DPS+TAG (25 step) | 204.8 | 54.1 | 56.8 |

---

> ### Author Response · Authors · 2025-11-23
> **Response to Reviewer MJq2 (2/3)**
>
> ### [R4-2] Manual Hyperparameter Tuning
> We clarify that TAG naturally incorporates an implicit adaptive nature regarding noise levels through theoretical design, and demonstrate empirical robustness across diverse downstream tasks, effectively eliminating a manual tuning.
>
> **1. Adaptivity to noise levels** Our novel TLS term $\nabla_x \log p(t \mid x)$ is inherently designed to be adaptive to noise levels. By treating timestep $t$ as a random variable, TLS quantifies the deviation from the target noise manifold $\mathcal{M}_t$. As shown in **Theorem3.3**, this generates a corrective gradient that automatically strengthens when a sample drifts to competing manifolds (incorrect noise levels), ensuring structural adaptivity without the need for manual scheduling.
>
> **2. Empirically Tuning-Free** Regarding diverse downstream tasks and model scales, we found that a simple default of $\omega=1.0$ is sufficient by ensuring the correction term is magnitude-aligned with the generative term. To demonstrate this, we performed a wide sweep ($\omega \in [0.5, 8.0]$) comparing the default setting against the optimum found via our Efficient Search protocol (detailed in **[R2-3]**).
>
> **Table R4-2.**  Robustness & Optimization on CIFAR-10
> *Comparison of Efficient Search (32 samples, 25 steps) vs. Full Evaluation (10k samples, 100 steps)*
>
>
> | Metric | DPS | 0.5 | 1.0 | 2.0 | 4.0 | 5.5 | 6.0 | 6.5 | 8.0 |
> | :--- | :---: | :---: | :---: | :---: | :---: | :---: | :---: | :---: | :---: |
> | **Efficient Search**<br>(Validity $\uparrow$) | 12.5 | 25.0 | 40.6 | 40.6 | 56.3 | 65.6 | **68.8** | 65.6 | 43.4 |
> | **Full Eval**<br>(FID $\downarrow$) | 203.6 | 187.1 | 174.5 | 162.7 | **159.5** | 161.2 | 162.5 | 165.4 | 180.7 |
> | **Full Eval**<br>(Validity $\uparrow$) | 41.6 | 47.3 | 51.9 | 58.0 | 61.8 | 63.5 | **63.3** | 62.8 | 60.3 |
>
> The results demonstrate two key points. **1. Tuning-Free Robustness**: Even without any search, the default $\omega=1.0$ significantly outperforms the baseline (FID $203.6 \to 174.5$, Validity $41.6 \to 51.9$), confirming that TAG provides immediate value as a plug-and-play module. **2. Easy Optimization**: For users seeking peak performance, our Efficient Search using reduced samples and inference steps correctly identifies the optimal region validated by the full evaluation. This confirms that while TAG is robust at the default setting, finding the theoretical optimum is practically feasible and computationally cheap. Note that all experiments in our manuscript were conducted using the default $\omega=1.0$ without this search.
>
> ### [R4-3] Redundant table and typo
> Thank you for the detailed feedback.
>
> For Section E.2, we found that the mismatch in Table 10 and optimal guidance strength comes from the typo: the result in Table 10 comes from the experiment with the extra noise schedule of **$\sigma=0.3$** instead of $\sigma=0.2$. We correctly modify the value in **line 2023** and **caption in Table 10** for preventing any confusion.
>
> Regarding **Table 10** and **Table 11**, we noticed Table 11 can be redundant where all information is included in Table 10. Therefore, we decided to remove Table 11 and instead add additional explanation in  **line 2162 - line 2164** for the clarity.
>
> Moreover, following your advice, we also fixed a grammatical error, changing "this" to **"these"** (**line 43**) and removed repetitive phrasing in the introduction (**line 71**) for conciseness.
>
> We will carefully review and double check the grammatical error or numerical rounding in our final manuscript.
>
> ### [R4-4] Further visualization for clearness
> We appreciate the constructive suggestion regarding the visual clarity. To address this, we have revised **Figures 11–13** and **17** (Now $\color{Crimson}{\text{Figures 14–16 and 19}}$) by incorporating zoom-in crops that shows fine-grained details.

---

> ### Author Response · Authors · 2025-11-23
> **Response to Reviewer MJq2 (3/3)**
>
> ### [R4-5] Time complexity of TAG
> We measure the wall-clock latency per sampling step on a single A6000 GPU. As shown in **Table R4-3**, TAG introduces a negligible overhead across different model scales. Notably, this overhead becomes even more insignificant for Large Scale Latent Models (e.g., < 0.6% for Stable Diffusion). This ensures the method remains highly scalable, as the predictor operates efficiently in the reduced latent space; since the task is simply to estimate the noise variance, it does not require processing high-dimensional pixel data. This efficient performance stems from the fact that the additional gradient term is computed via backpropagation solely through the lightweight Time Predictor (SimpleCNN, ~1.5M parameters; see **Appendix E.4**), avoiding the heavy computation of the diffusion backbone.
>
> **Table R4-3.** Wall-clock Latency (per step)
> | Task | Total Sampling (ms) | TAG Sampling (ms) | Overhead (%) |
> | :--- | :---: | :---: | :---: |
> | CIFAR-10 | 115.5 | 4.4 | < 3.81% |
> | ImageNet | 384.9 | 8.23 | < 2.14% |
> | Stable Diffusion | 4.14 | 798.18 | < 0.52% |
>
> Additionally, regarding training efficiency, the Time Predictor requires minimal resources due to its small architecture (only $\sim 0.17\%$ of a standard Stable Diffusion UNet, as detailed in **Appendix E.4**). Furthermore, as analyzed in **Table 16 (Appendix E.4)**, training converges within 10k–30k iterations, requiring only a few hours on a single GPU. To further demonstrate this, we evaluate the performance trade-off across training steps in **Table R4-4** (please refer to **[R2-1]** for further details).
>
> **Table R4-4** Predictor Robustness on CIFAR-10 (10,000 samples)
>
> | Predictor Training Steps | **DPS**|**10k Steps (Early)** | **20k steps** | **30k Steps (Converged)**|
> | :--- | :---: | :---: |:---: |:---: |
> | **FID** ($\downarrow$) | 203.6 | 176.1 | 175.6|**174.5** |
> | **Accuracy** ($\uparrow$) | 41.6 | 48.5 | 51.2|**51.9** |
>
> As shown above, even with a small training cost (trained for just 10k steps, which takes 40 minutes on a single A6000), TAG significantly outperforms the baseline (FID 203.6 $\to$ 176.1). More details of the training cost of the predictor in **Table R3-2.** across diverse tasks. This represents a trivial one-time cost compared to the extensive resources required for training or fine-tuning the backbone model.
>
> ---
> Thank you again for your constructive feedback. We are happy to engage in further discussion if there are any remaining questions.

---

> > ### Comment · Reviewer_MJq2 · 2025-11-24
> >
> > Most of my concerns have been addressed well. I keep my original rating.

---

> ### Author Response · Authors · 2025-11-30
> **Thank you for your feedback and support**
>
> Dear Reviewer MJq2,
>
> We are thrilled to hear that our revisions have **successfully resolved all the concerns** you raised.
> Your constructive feedback was crucial in finalizing our paper, and we are grateful for your time and endorsement. Thank you for helping us strengthen our work.
>
> Best regards,
>
> The Authors

---

### Official Review · Reviewer_VLZo · 2025-10-27

**Soundness:** 3
**Presentation:** 2
**Contribution:** 3
**Rating:** 6
**Confidence:** 3

**Summary:**

The paper introduces Temporal Alignment Guidance (TAG), a plug-in guidance term for diffusion sampling that aims to keep trajectories on-manifold when external guidance is applied. The key idea is to learn a lightweight time predictor that estimates the timestep posterior. The authors then add the gradient of the Time-Linked Score to the model score at every step, which encourages the sample to remain consistent with the intended time manifold so that guidance does not push it into low-density regions. The paper provides a theoretical rationale for this correction and reports improvements across several scenarios.

**Strengths:**

1. The proposed method offers a clear and broadly applicable mechanism that enhances existing guided diffusion frameworks.
2. The manuscript is well-structured and readable, making its contributions accessible.
3. The theoretical component provides sound motivations that align well with the empirical findings.
4. A wide set of experiments across different domains supports the practical relevance and robustness of the approach.

**Weaknesses:**

1. Since the paper employs a large number of symbols, it would greatly improve readability to include a notation table summarizing the meaning of key variables and subscripts.
2. The paper does not state the relation between TAG and prior manifold-preserving guidance approaches [1–3].

[1] Yang, Lingxiao, et al. "Guidance with Spherical Gaussian Constraint for Conditional Diffusion." *International Conference on Machine Learning*. PMLR, 2024.

[2] He, Yutong, et al. "Manifold Preserving Guided Diffusion." *ICLR*. 2024.

[3] Chung, Hyungjin, et al. "CFG++: Manifold-constrained Classifier Free Guidance for Diffusion Models."  *ICLR*. 2025.

**Questions:**

1. How reliable is TAG when the time predictor $p_\phi(t\mid x)$ is imperfect? Does the method remain stable under strong external guidance or off-manifold drift?
2. Is the proposed Time-Gap metric quantitatively correlated with established quality measures such as FID, CLIP, or task-specific reward scores? Could the authors provide correlation analyses to demonstrate its validity as a general misalignment indicator?
3. What is the computational cost of training the time predictor and computing the additional gradient term $\nabla_x \log p_\phi(t\mid x)$ at each sampling step, especially in large latent models?

---

> ### Author Response · Authors · 2025-11-23
> **Response to Reviewer VLZo (1/3)**
>
> We sincerely appreciate your thorough and constructive feedback, which has been invaluable in improving our work. We are grateful for your acknowledgement that:
>
> * Well structured presentation and TAG's clearness and broad applicability.
> * Theoretical analysis with sound motivation, well aligned with experimental results.
> * Practicality and robustness of TAG across a wide range of different domains and tasks.
>
>
> Please note that all revisions made to address your concerns are marked in $\color{Crimson}{\text{red}}$ in the revised manuscript.
>
> We address your concerns below.
>
> ---
>
> ### [R3-1] Notation Table
> We thank the reviewer for the constructive suggestion. We have added a comprehensive Notation Table summarizing key variables and subscripts in $\color{Crimson}{\text{Table 7. (Appendix C)}}$ of the revised manuscript.
>
>
> ### [R3-2] Relation to Prior Manifold-Preserving Approaches
> We thank the reviewer for highlighting these relevant works. We clarify the distinct contributions of TAG compared to prior manifold-preserving methods [1-3]. While these approaches address manifold deviation through geometric constraints or vector corrections within the data space, TAG introduces a completely novel perspective: **identifying and correcting off-manifold errors via the temporal axis**. We are the first to frame this issue as a timestep deviation and correct it using the Time-Linked Score (TLS), providing a unique orthogonal contribution that can be applied on top of the prior methods [1-3]. We have revised our manuscript to incorporate this in $\color{crimson}{\text{Appendix D.1}}$.
>
> **1. DSG (Yang et al., 2024) [1]**
> DSG [1] constrains the guidance step within a spherical shell centered at the unconditional mean $\mu_\theta(x_t)$, based on the concentration of high-dimensional Gaussian distributions.
> While DSG relies on a projection onto a sphere to mitigate deviation, TAG employs a learned Time-Linked Score (TLS), $\nabla_x \log p(t|x)$. Unlike DSG's hard constraint which assumes isotropy, TAG provides a soft correction that respects the complex, non-isotropic structure of the density field, actively pulling samples back to the high-density region of the specific timestep manifold $\mathcal{M}_t$.
>
> **2. MPGD (He et al., 2024) [2]**
> MPGD [2] projects the guidance gradient onto the tangent space of the data manifold using a pre-trained autoencoder's Jacobian.
> MPGD relies on the locally linear approximation and requires an auxiliary autoencoder, which may not be available for all domains (e.g., molecular graphs). In contrast, TAG is model-agnostic and does not require manifold linearity or external autoencoders. It leverages the diffusion process's own temporal information, making it applicable to diverse domains (images, molecules, audio) where such geometric assumptions may fail.
>
> **3. CFG++ (Chung et al., 2025) [3]**
> CFG++ [3] modifies the vector field of classifier-free guidance to remain on the data manifold by correcting the unconditional score component.
> CFG++ is specific to the CFG framework and focuses on correcting the extrapolation error of the guidance vector itself. TAG is a general-purpose correction mechanism applicable to various guidance settings (classifier guidance, cfg, and training-free guidance). As shown in [R2-2], TAG can be integrated on top of CFG/CFG++ to further correct off-manifold drifts caused by aggressive scaling, offering an orthogonal layer of robustness.

---

> ### Author Response · Authors · 2025-11-23
> **Response to Reviewer VLZo (2/3)**
>
> ### [R3-3] Robustness of time predictor & Stability under Drift
> We confirm that TAG is highly robust to predictor imperfections and remains stable under strong drift.
>
> **1. Robustness to imperfect predictors**
> Our experiments demonstrate that TAG is highly robust to predictor imperfections because the task relies on estimating the global noise variance—a fundamental property of the diffusion process—rather than fitting complex, sample-specific data distributions. Consequently, the predictor captures the necessary corrective gradient direction $\nabla_x \log p(t|x)$ early in training. This is evidenced by **Table 16 (Appendix E.4)** and its larger samples version **Table R3-1**, where an early-stage predictor (10k steps) performs nearly as well as a converged one (30k steps) on a large scale (10,000 samples).
>
> To further stress-test this resilience against explicit prediction errors, we evaluated TAG while injecting Gaussian noise $\mathcal{N}(\mathbf{0}, \sigma^2\mathbf{I})$ into the predictor logits. As shown in **Table R3-2**, TAG maintains significant performance gains over the baseline even under substantial noise perturbation, degrading gracefully only at extreme levels. These results, combined with the effectiveness of lightweight architectures (SimpleCNN, **Table 9**), confirm that TAG relies on the corrective gradient direction rather than perfect probability calibration, ensuring stability even when the predictor is lightweight or noisy.
>
>
> **Table R3-1** Predictor Robustness on CIFAR-10 (10,000 samples)
> | Predictor Training Steps | **DPS**|**10k Steps (Early)** | **20k steps** | **30k Steps (Converged)**|
> | :--- | :---: | :---: |:---: |:---: |
> | **FID** ($\downarrow$) | 203.6 | 176.1 | 175.6|**174.5** |
> | **Accuracy** ($\uparrow$) | 41.6 | 48.5 | 51.2|**51.9** |
>
> **Table R3-2.** Robustness to logit noise Injection (1,000 samples)
> *(Stress test with additive Gaussian noise $\sigma$ on predictor logits)*
>
> | Noise Level ($\sigma$) | **DPS** | **0.0 (Standard)** | **0.1** | **0.5** | **1.0** | **2.0** | **5.0** | **10.0** |
> | :--- | :---: | :---: | :---: | :---: | :---: | :---: | :---: | :---: |
> | **FID** ($\downarrow$) | 218.9 | **195.7** | 200.9 | 199.3 | 196.1 | 214.7 | 283.2 | 348.7 |
> | **Accuracy** ($\uparrow$) | 41.4 | 50.3 | 50.4 | **53.2** | 52.7 | 48.9 | 36.7 | 26.6 |
>
>
> **2. Stability under Strong Guidance & Drift**
> TAG serves as a stabilizer under extreme conditions. In **Table 3 (Section 4)**, we tested increasing the guidance strength of DPS from 1.0 to 5.0. While the baseline collapsed (producing non-valid samples), TAG maintained high fidelity and validity, effectively counteracting the strong off-manifold forces. Additionally, in the corrupted reverse process experiment (**Table 1** & **Figure 3**), where we injected gaussian noise ($\sigma$ up to 0.3) to simulate severe drift, TAG significantly recovered generation quality (FID reduction from 410.1 to 223.2 at $\sigma=0.3$). These results confirm that TAG becomes more critical and effective as the external guidance or drift intensifies.
>
> ### [R3-4] Correlation of Time-Gap with quality measures
>
> Thank you for the constructive feedback. We conduct an additional set of experiments with DPS+TAG framework and measure the correlation between time-gap and the performance metric along with different TAG strengths. The result in **Table R3-3** shows that the decreasing timegap results in decreasing FID and increasing Validity, showing strong correlation between the performance metric and the time gap.
>
> **Table R3-3.** Time gap correlation (CIFAR10, 1,024 samples, 100 inference steps).
>
> | TAG strength ($w$) | 0.0 | 0.5 | 1.0 | 1.5 | 2.0 | 3.0 | 4.0 | 5.0 |
> | :--- | :---: | :---: | :---: | :---: | :---: | :---: | :---: | :---: |
> | **Time Gap** ($\downarrow$) | 69.4 | 58.3 | 52.9 | 48.9 | 45.9 | 42.4 | 41.5 | **41.4** |
> | **FID** ($\downarrow$) | 225.1 | 208.6 | 199.1 | 189.4 | 183.1 | **182.6** | 183.2 | 185.3 |
> | **Validity** ($\uparrow$) | 39.3 | 48.6 | 50.1 | 55.2 | 61.3 | 58.9 | **63.3** | 62.4 |

---

> ### Author Response · Authors · 2025-11-23
> **Response to Reviewer VLZo (3/3)**
>
> ### [R3-5] Computational cost of time predictor
> We confirm that the computational overhead of TAG is negligible compared to the backbone diffusion process, making it scalable even for large models.
>
> **1. Training Cost**
> The Time Predictor is designed as an extremely lightweight SimpleCNN with only ~1.5M parameters, as detailed in **Appendix E.4**. Notably, we utilized this identical lightweight architecture for all experiments, ranging from standard benchmarks to large-scale tasks without requiring a larger model. Consequently, training converges within 10k–30k iterations—as analyzed in Table 16—and can be completed in just a few hours on a single A6000 GPU as shown below (**Table R3-4**).
>
> **Table R3-4.** Time Predictor Training Cost (10k training steps).
> | Task | Wall-clock Time |
> | :--- | :---: |
> | CIFAR-10 | ~56 m |
> | Stable Diffusion | ~6 h 10 m |
> | Audio | ~5 h 22 m |
> | Molecule | ~2 h 18 m |
>
> **2. Inference Cost**
> For inference, the additional gradient term is computed solely through this lightweight predictor, avoiding the heavy computation of the diffusion backbone. To demonstrate this, we provide **Table R3-5** below, measuring the wall-clock latency across different model scales. The results indicate that TAG introduces a negligible overhead, which becomes even more insignificant for large-scale latent models (e.g., < 0.6% for Stable Diffusion). This ensures the method remains highly scalable, as the predictor operates efficiently in the reduced latent space. This is because the task is simply to estimate the noise variance, it does not require processing high-dimensional pixel data as mentioned in **[R1-3, 2-1, 3-3]**
>
> **Table R3-5**. Wall-clock Latency (per step).
> | Task | Total Sampling (ms) | TAG Sampling (ms) | Overhead (%) |
> | :--- | :---: | :---: | :---: |
> | CIFAR-10 | 115.5 | 4.4 | < 3.81% |
> | ImageNet | 384.9 | 8.23 | < 2.14 % |
> | Stable Diffusion | 4.14 | 798.18 | < 0.52% |
>
>
> ---
>
> Thank you again for your constructive feedback. We are happy to engage in further discussion if there are any remaining questions.
>
> ---
> ### References
> [1] Yang et al. Guidance with Spherical Gaussian Constraint for Conditional Diffusion. ICML 2024.
>
> [2] He et al. Manifold Preserving Guided Diffusion. ICLR 2024.
>
> [3] Chung et al. CFG++: Manifold-constrained Classifier Free Guidance for Diffusion Models. ICLR 2025.

---

> ### Comment · Reviewer_VLZo · 2025-11-26
>
> Thanks for your response, most of my concerns are addressed, and I'll keep my score.

---

> ### Author Response · Authors · 2025-11-30
> **Thank you for your feedback and support**
>
> Dear Reviewer VLZo,
>
> We are glad to hear that our response and the additional revisions have **successfully addressed all of your concerns**.
> Your constructive feedback has been instrumental in strengthening our paper. We appreciate your time and your positive assessment of our work.
>
> Best regards,
>
> The Authors

---

### Official Review · Reviewer_jUUY · 2025-10-28

**Soundness:** 3
**Presentation:** 2
**Contribution:** 3
**Rating:** 6
**Confidence:** 3

**Summary:**

This paper seeks to address the off-manifold phenomenon observed in diffusion models. Defined as the issue where even well-trained models accumulate errors during generation, particularly when arbitrary guidance is applied to steer samples toward desired properties. This phenomenon causes generated samples to deviate from the desired data manifold, ultimately compromising sample fidelity. To tackle this, the authors first employed a time predictor to estimate deviations from the desired data manifold at each timestep, identifying that a larger time gap correlates with diminished generation quality. Subsequently, they developed a novel guidance mechanism termed Temporal Alignment Guidance (TAG), which pulls samples back to the desired manifold at every timestep throughout the generation process.

**Strengths:**

+ The proposed method is intuitive and effective in addressing the off-manifold phenomenon of diffusion models.

+ Extensive experiments and theoretical analysis are conducted to demonstrate the improvements brought by the proposed method.

+ The proposed method can be applied to various downstream tasks to enhance their performance.

**Weaknesses:**

- The proposed method necessitates the integration of a time predictor, which must be tailored to different downstream models. Notably, the performance of this time predictor exerts a substantial influence on the overall efficacy of the method.

- The method presented in this paper is developed primarily by analyzing and addressing challenges arising from the use of classifier guidance in diffusion-based generation. However, it would be valuable to further investigate two key points: first, whether the off-manifold phenomenon also occurs in classifier-free guidance—a technique widely adopted in diffusion-based generation, and second, if it does, whether a analogous solution can be effectively applied.

- In Equation (7), the Temporal Alignment Guidance (TAG) introduces a parameter $\omega$ to control its strength. In practical applications, how should the value of this parameter be determined? And will the optimal value of $\omega$ vary significantly across different downstream tasks?

**Questions:**

Please see weaknesses above.

---

> ### Author Response · Authors · 2025-11-24
> **Response to Reviewer jUUY (1/4)**
>
> We thank you for your detailed review and for acknowledging the strengths of our work, including:
>
> * TAG's intuitiveness and effectiveness in solving the off-manifold phenomenon.
> * Extensive experimental validation with theoretical analysis.
> * The broad applicability of our proposed algorithm across various downstream tasks.
>
> Please note that all revisions made to address your concerns are marked in $\color{Crimson}{\text{red}}$ in the revised manuscript.
>
> We address your concerns below.
>
> ---
>
> ### [R2-1] Dependency on Time Predictor's performance
>
> We would like to clarify that the time-predictor is designed as a **lightweight and robust component requiring only a minimal integration cost**.
>
> The predictor learns to estimate the noise variance characteristic of each timestep which is a fundamental property of the diffusion process rather than fitting to the complex high-dimensional data distribution of individual samples. This is empirically supported by our diverse benchmarks (**Appendix E.3**), where the predictor consistently maintains low Time-Gaps and high generation quality across varied domains. Notably, unconditional time predictors match or exceed the performance of conditional ones (**Table 5**), proving that learning the global noise schedule is sufficient for effective guidance, thereby eliminating the need for tailoring the predictor to specific downstream models.
>
> Regarding integration cost, the predictor requires minimal resources. As detailed in **Table 9 (Appendix E.1)**, a SimpleCNN (only $\\sim 8.5\%$ of backbone parameters) achieves performance comparable to heavier models (e.g. Unet encoder [1]). Furthermore, our sensitivity analysis (**Table 16** and **Table R2-1** below) demonstrates that the performance gap between a predictor trained for only 10k steps versus 30k steps is minimal. To further stress-test this resilience against explicit prediction errors, we evaluated TAG while injecting Gaussian noise into the predictor logits. As shown in **Table R2-2**, TAG maintains significant performance gains over the baseline even under noise perturbation. These results collectively confirm that TAG relies on the **corrective gradient direction** rather than perfect probability calibration, making the method highly cost-effective and robust to predictor imperfections.
>
> **Table R2-1.** Predictor Robustness on CIFAR-10 (10,000 samples)
> | Predictor Training Steps | **DPS**|**10k Steps (Early)** | **20k steps** | **30k Steps (Converged)**|
> | :--- | :---: | :---: |:---: |:---: |
> | **FID** ($\downarrow$) | 203.6 | 176.1 | 175.6|**174.5** |
> | **Accuracy** ($\uparrow$) | 41.6 | 48.5 | 51.2|**51.9** |
>
> **Table R2-2.** Robustness to logit noise Injection (1,000 samples)
> *(Stress test with additive Gaussian noise $\sigma$ on predictor logits)*
>
> | Noise Level ($\sigma$) | **DPS** | **0.0 (Standard)** | **0.1** | **0.5** | **1.0** | **2.0** | **5.0** | **10.0** |
> | :--- | :---: | :---: | :---: | :---: | :---: | :---: | :---: | :---: |
> | **FID** ($\downarrow$) | 218.9 | **195.7** | 200.9 | 199.3 | 196.1 | 214.7 | 283.2 | 348.7 |
> | **Accuracy** ($\uparrow$) | 41.4 | 50.3 | 50.4 | **53.2** | 52.7 | 48.9 | 36.7 | 26.6 |

---

> ### Author Response · Authors · 2025-11-24
> **Response to Reviewer jUUY (2/4)**
>
> ### [R2-2] Off-Manifold Phenomenon in CFG and TAG
>
>
> To investigate whether off-manifold phenomenon occurs in CFG, we measure the Time-Gap across different CFG strengths $w_{cfg}$ in the large scale experiment (Stable Diffusion v1.5 ([2]) with HPSv2 prompt set, 100 samples per each) with total sampling step of 100.
>
> **Table R2-3.** Off-manifold phenomenon along CFG strength.
> | CFG strength ($w_{cfg}$) | 2 | 3 | 5 | 6 | 8 | 10 | 15 | 20
> | :--- | :---: | :---: |:---: |:---: | :---: | :---: | :---: | :---: |
> | Time Gap ($\downarrow$) | 31.44 | 28.90 | 26.17 | 25.54 | **25.15** | 25.77 | 28.94 | 31.34 |
> | Aesthetic Score ($\uparrow$) | 4.965 | 5.092 | 5.164 | 5.190 | 5.213 | 5.248 | **5.271** | 5.216 |
> | Clip Score ($\uparrow$) | **0.0582** | 0.0572 | 0.0557 | 0.0585 | 0.0565 | 0.0545 | 0.0523 | 0.0551 |
>
> The above result in **Table R2-3** indicates that indeed, **CFG can induce off-manifold samples** especially when guidance strength ($w_{cfg}$) becomes high. Interestingly, as seen in the **Table R2-3**, while the minimum of the Time-Gap metric occurs around $w_{cfg}=8$, which is known to be the optimal CFG value ([3]), other reward metrics show less perturbation. The overall effect is less pronounced in the presence of external guidance (DPS/TFG), which may be attributed to the model being exposed during training to the corresponding conditional and unconditional score functions.
>
> Now, to show the effectiveness of TAG under the off-manifold phenomenon in CFG, we apply TAG with high CFG strength ($w_{cfg}=20$).
>
> **Table R2-4.** TAG in high CFG strength scenario.
> | TAG strength ($w_{tag}$) | 0.0 | 0.25 | 0.5 | 0.75 | 1.00 | 1.25 | 1.50 | 1.75
> | :--- | :---: | :---: |:---: |:---: | :---: | :---: | :---: | :---: |
> | Time Gap ($\downarrow$) | 31.44 | 21.32 | 15.86 | 13.69 | 12.44 | 11.73 | **11.39** | 12.24 |
> | Aesthetic Score ($\uparrow$) | 5.216 | 5.232 | 5.237 | 5.217 | 5.251 | 5.245 | **5.256** | 5.243 |
>
>
> The results above (**Table R2-4**) show that TAG is still effective in reducing the off-manifold phenomenon (indicated by reduced time-gap) in the plain CFG scenario, **successfully mitigating the deviation**. Moreover, optimal TAG strength for time-gap metric coincides with the Aesthetic score and two metrics show clear correlation (lower time gap shows higher Aesthetic Score), though this effect is less pronounced than in the presence of external guidance term.
>
> We further interpret the above result by noting that, although well-optimized CFG methods may show fewer deviations, they fundamentally rely on a linear extrapolation between conditional and unconditional scores. Recent studies [4, 5] analyze that at high guidance scales, this extrapolation inevitably pushes the sampling trajectory away from the natural data manifold, resulting in intense color saturation or off-manifold artifacts. This aligns precisely with the off-manifold deviation defined in our work (Section 2), where the generated samples drift into low-density regions of the data distribution.
>
> TAG is uniquely positioned to address this issue because it introduces an orthogonal corrective force. While CFG modifies the score to satisfy the condition $c$ (often at the cost of realism), TAG’s Time-Linked Score (TLS) $\nabla_x \log p(t|x)$ provides a distinct gradient that pulls the sample back to the high-density manifold of the current noise level. As discussed in **[R1-1]** and visualized in **Figure 2**, this correction helps restore the valid noise level without conflicting with the semantic steering of the guidance. Therefore, integrating TAG into the CFG framework can serve as an effective manifold constraint, mitigating the off-manifold artifacts caused by excessive extrapolation.
>
> This perspective is consistent with recent manifold-preserving approaches like [4] and [6], which also identify the need for correction in CFG. However, unlike methods that require specific architectural changes or heavy optimization, TAG offers a lightweight, plug-and-play solution by leveraging the temporal consistency inherent in the diffusion process itself.
>
> We highly appreciate your insightful comment and have revised our manuscript to incorporate this in $\color{crimson}{\text{Appendix D.1}}$.

---

> ### Author Response · Authors · 2025-11-24
> **Response to Reviewer jUUY (3/4)**
>
> ### [R2-3] Practical Tuning of $\omega$
> Here, we clarify our practical hyperparameter selection process. We demonstrate that while the optimal strength may vary across tasks, a simple default ($\omega=1.0$) consistently offers robust improvements over baselines. Furthermore, our lightweight protocol allows users to easily identify the task-specific optimum with minimal computational cost.
>
> **1. Practical $\omega$ selection guideline**
> For $\omega$ selection, we follow a lightweight three-step search procedure using a small validation set.
> - **Step1. Initialization:** We initialize with $\omega=1$. In our implementation, we normalize the TLS term to match the magnitude of the score term. Under this balanced setup, $\omega=1.0$ acts as a reliable, balanced starting point.
> - **Step 2. Efficient Search:** We perform a coarse sweep (e.g., doubling values: 0.5, 1.0, 2.0, 4.0, 8.0) using a small number of samples (e.g., 32) with reduced inference steps (e.g., 25 steps). Based on the peak performance, we conduct a fine-grained refinement around that value (e.g., $\pm 0.5$). This approximation minimizes computational cost compared to full inference while preserving relative performance trends.
> - **Step3. Selection:** We select the strength that maximizes validity metric (e.g. acc.). We found that the trend in validity is robust even with small sample sizes, making it a reliable proxy unlike fidelity metric (e.g. FID) which is very sensitive to number of samples.
>
> This process requires only a small number of samples and negligible computation compared to full evaluation, resulting in minimal searching cost: all our searching experiments takes **less than 5 minutes** in a single A6000 gpu. We put the searching result across diverse domains in **Table R2-5,6,7**.
>
> **Table R2-5.** Efficient Search (CIFAR10, 32 samples, 25 steps)
> | Metric | **DPS** | 0.5 | **1.0** | 2.0 | 4.0 | 5.5 | 6.0 | 6.5 | 8.0 |
> | :--- | :---: | :---: | :---: | :---: | :---: | :---: | :---: | :---: | :---: |
> | **FID** ($\downarrow$) | 299.3| 304.1|326.6 | 315.8| 323.5|336.2 | 321.2| 324.5| 325.3|
> | **Validity** ($\uparrow$) |12.5 |25.0 | 40.6|40.6 |56.3 | 65.6|**68.8**|65.6 | 43.4|
>
>
> **Table R2-6.** Efficient Search (Molecule $\alpha$, 256 samples, 25 steps)
> | Metric | **DPS** | 0.5 | **1.0** | 2.0 | 4.0 | 5.5 | 6.0 | 6.5 | 8.0 |
> | :--- | :---: | :---: | :---: | :---: | :---: | :---: | :---: | :---: | :---: |
> | **MAE** ($\downarrow$) | 45.9| 34.3|39.9 | 48.8| 40.2|42.8 | 35.2| 30.4| 49.3|
> | **Stability** ($\uparrow$) |9.6 |12.9 | 17.1|13.2 |15.1 | 15.2|**17.3**|16.1 | 14.0|
>
>
> **Table R2-7.** Efficient Search (Audio declipping, 32 samples, 25 steps)
> | Metric | **TFG** | 0.2 | 0.3 | 0.4 | 0.5 | 0.6 | 0.8 | 1.0 |
> | :--- | :---: | :---:  | :---: | :---: | :---: | :---: | :---: | :---: |
> | **FAD** ($\downarrow$) | 13.7| 7.6| 7.0| 6.4|5.3 | 5.3| 5.3| 8.4|
> | **DTW** ($\downarrow$) |898 |351 | 323 |**303** | 357|335|421| 459|
>
>
> **2. Empirically Tuning-Free**
> To validate this protocol, we compared the performance of the fixed default $\omega=1.0$ against the optimal value found via the guideline above using a full evaluation using larger samples and more inference steps. We put the results in **Table R2-8,9,10**.
>
> **Table R2-8.** Full Evaluation (CIFAR10, 10k samples, 100 steps)
>
> | Metric | **DPS** | 0.5 | **1.0** | 2.0 | 4.0 | 5.5 | 6.0 | 6.5 | 8.0 |
> | :--- | :---: | :---: | :---: | :---: | :---: | :---: | :---: | :---: | :---: |
> | **FID** ($\downarrow$) |203.6 | 187.1| 174.5| 162.7|171.35| **161.2**|162.5 | 165.4|180.7 |
> | **Validity** ($\uparrow$) |41.6 | 47.3| 51.9|58.0 | 62.5| **63.5**|63.3 |62.8 |60.3 |
>
> **Table R2-9.** Full Evaluation (Molecule $\alpha$, 4096 samples, 100 steps)
> | Metric | **DPS** | 0.5 | **1.0** | 2.0 | 4.0 | 5.5 | 6.0 | 6.5 | 8.0 |10.0|
> | :--- | :---: | :---: | :---: | :---: | :---: | :---: | :---: | :---: | :---: | :---: |
> | **MAE** ($\downarrow$) |106.6 | 106.9|92.0 | 79.2|64.7| 58.0| 57.8| **56.7**|57.3|60.8|
> | **Stability** ($\downarrow$) |1.0 | 2.5| 7.3|10.6 | 16.7| 18.9|19.5 |**19.9** | 19.5|19.6|
>
>
>
> **Table R2-10.** Full Evaluation (Audio Declipping, 512 samples, 100 steps)
> | Metric | **TFG** | 0.2 | 0.3 | 0.4 | 0.5 | 0.6 | 0.8 | 1.0 |
> | :--- | :---: | :---:  | :---: | :---: | :---: | :---: | :---: | :---: |
> | **FAD** ($\downarrow$) | 2.7| 1.2| 1.0| 0.9|**0.8** | 1.3|1.27 |1.24 |
> | **DTW** ($\downarrow$) |277 |154 | **149** |165 | 159|220|248.2|233.6|
>
> Even without any tuning, the default $\omega=1.0$ outperforms the baseline DPS, proving that TAG is effectively tuning-free for obtaining immediate gains. The optimal region identified by the cheap proxy ($\omega \approx 6.0$) perfectly aligns with the best performance in the full evaluation. This confirms that users can safely rely on our low-cost protocol to unlock the full potential of TAG across diverse tasks.

---

> ### Author Response · Authors · 2025-11-24
> **Response to Reviewer jUUY (4/4)**
>
> Thank you again for your invaluable feedback. We are happy to engage in further discussion if there are any remaining questions.
>
> ---
> ### References
> [1] Dhariwal et al. Diffusion Models Beat GANs on Image Synthesis. NeurIPS 2021
>
> [2] Rombach et al. High-resolution image synthesis with latent diffusion models. CVPR 2024.
>
> [3] Wang et al. Analysis of Classifier-Free Guidance Weight Schedulers. TMLR 2024.
>
> [4] Chung et al. CFG++: Manifold-constrained classifier-free guidance for diffusion models. ICLR 2025
>
> [5] Sadat et al. No Training, No Problem: Rethinking Classifier-Free Guidance for Diffusion Models. ICLR 2025
>
> [6] He et al. Manifold preserving guided diffusion (MPGD). ICLR 2024.

---

### Official Review · Reviewer_fmcf · 2025-10-30

**Soundness:** 3
**Presentation:** 3
**Contribution:** 2
**Rating:** 4
**Confidence:** 3

**Summary:**

This paper addresses the problem in diffusion models where conditional gradients during sampling cause samples to gradually deviate from the true data manifold (referred to as the "off-manifold" phenomenon). It proposes an effective gradient correction strategy: Temporal Alignment Guidance (TAG). This method explicitly applies corrective gradients at each sampling timestep, pulling the samples back to high-probability regions of the original distribution. This suppresses deviations caused by unreasonable conditional guidance, ensuring the sampling trajectory remains aligned with the target manifold throughout the reverse diffusion process.

**Strengths:**

1. Controlling by external guidance, Multi-conditional guidance, Few-step generation, and Degradation of sample quality in low-density regions. The paper clearly explains the sources of the problem and validates them experimentally, which is highly convincing.

2. Introduces the Time-Linked Score (TLS) and provides corresponding probabilistic and energy-based theoretical explanations. Through theorems and propositions, it analyzes TAG’s convergence and anti-deviation properties, offering strong mathematical support for the method’s reliability.

3. Demonstrates improvements across tasks including image, audio, molecular, multi-conditional, and few-step generation, showing the method’s generality. Additionally, TAG is implemented as an external guidance module, making it lightweight and easy to deploy.

**Weaknesses:**

1. Lacks direct experimental verification of score approximation errors.

2. When combined with strong conditional guidance, TAG may conflict with the original guidance gradient, potentially causing samples to deviate from the target semantics or degrade generation quality. This scenario is not discussed in the paper.

3. The Time Predictor may introduce new error sources; if time classification is incorrect, its gradient might push samples in the wrong direction. The robustness of the model in this context is not discussed.

**Questions:**

1. In practical sampling, TAG applies gradients alongside the original conditional guidance. If their directions conflict, does TAG’s correction affect the effectiveness of conditional control in maintaining sample quality?

2. Could the authors provide a more intuitive quantification or visualization of score approximation error accumulation during sampling, and demonstrate TAG’s ability to reduce this error at each timestep, rather than relying solely on final sample metrics?

---

> ### Author Response · Authors · 2025-11-23
> **Response to Reviewer fmcf (1/3)**
>
> We sincerely appreciate your insightful and constructive feedback, which has been invaluable in improving our work. We are also grateful to the reviewer for acknowledging the key strengths of our approach, including:
>
> * TAG's strong mathematical support and theoretical analysis of convergence.
> * The clear identification of the "off-manifold" problem sources along with experimental validation.
> * The method's generality and effectiveness across diverse tasks including audio, molecule, image.
>
> Please note that all revisions made to address your concerns are marked in $\color{Crimson}{\text{red}}$ in the revised manuscript.
>
> The new visualizations made can be found in $\color{Crimson}{\text{Fig.4 (line 2067)}}$ and $\color{Crimson}{\text{Fig.6 (line 2118)}}$.
>
> We address your concerns below.
>
> ---
> ### [R1-1] Score approximation error accumulation
>
> To verify the score approximation error and TAG's ability to reduce it, we conduct an additional set of experiments. Specifically, we first examined the behavior of samples during the reverse process under the original DDPM sampler and TAG. For each sampler, we computed the Wasserstein-1 distance (W1) between the samples during the generation with external drift and without external drift. **Table R1-1** shows that applying TAG consistently reduces the W1 distance, pushing samples to the original sampling trajectory without any drift term. We also visualize in $\color{Crimson}{\text{Fig.4 (line 2067)}}$ in our revised manuscript for the comparison.
>
>
> **Table R1-1.** Trajectory Divergence Analysis ($W_1$ Distance $\downarrow$).
> | Method | $t=90$ | $t=80$ | $t=70$ | $t=60$ | $t=50$ | $t=40$ | $t=30$ | $t=20$ | $t=10$ | $t=0$ |
> | :--- | :---: | :---: | :---: | :---: | :---: | :---: | :---: | :---: | :---: | :---: |
> | Standard DDPM | **0.070** | 0.226 | 0.393  | 0.745 | 1.232 | 1.377 | 1.758 | 2.614  |  2.683| 3.286 |
> | TAG(Ours) | 0.077 | **0.194** | **0.326** | **0.526** | **0.681** | **0.876** | **1.026** | **1.238** | **1.287** | **1.736** |
>
> Next, we conduct additional analysis to directly measure the score approximation error by estimating Fisher Divergence ([1],[2]) ( $\mathbb{E}\_{\mathbf{x}\_t} \\left[ \\| s\_\theta(\mathbf{x}\_t, t) - \\nabla\_{\mathbf{x}\_t} \\log p\_t(\mathbf{x}\_t) \\|_2^2 \\right]$
> ) at each timestep $t$ where we approximate ground-truth score with test dataset. **Table R1-2** and **Table R1-3** report the instantaneous score-approximation error at each timestep and the cumulative score-approximation error during sampling (evaluated every 10 timesteps), providing a clear comparison of how the error evolves throughout the reverse process. Here, we also compare the original DDPM sampler and TAG in the presence of external drift. The result shows that TAG significantly reduces the score approximation error, by **more than 40 times compared to the original DDPM sampler**. We further visualize the result at $\color{Crimson}{\text{Fig.6 (line 2118)}}$ in our revised manuscript and put additional explanation in **Section E.1** in our manuscript.
>
> **Table R1-2.** Instantaneous score approximation error between DDPM vs TAG.
> | Method | $t=90$ | $t=80$ | $t=70$ | $t=60$ | $t=50$ | $t=40$ | $t=30$ | $t=20$ | $t=10$ | $t=1$ |
> | :--- | :---: | :---: | :---: | :---: | :---: | :---: | :---: | :---: | :---: | :---: |
> | Standard DDPM | 0.104 | 0.188 | 0.118  | 0.070 | 0.125 | 0.334 |  2.483 | 5.535e+1  | 2.767e+3 | 1.119e+7 |
> | TAG(Ours) | 0.106 | 0.216 | 0.123 | 0.074 |  0.067 |   0.093 | 0.303 | 2.431 | 7.534e+1 | 3.297e+5 |
> | Ratio(TAG/DDPM) | 1.022 | 1.150 | 1.036 | 1.056 | 0.540 |  0.278 | 0.122 | 0.044 | 0.027 | 0.027
>
> **Table R1-3.** Cumulative score approximation error between DDPM vs TAG.
> | Method | $t=90$ | $t=80$ | $t=70$ | $t=60$ | $t=50$ | $t=40$ | $t=30$ | $t=20$ | $t=10$ | $t=1$ |
> | :--- | :---: | :---: | :---: | :---: | :---: | :---: | :---: | :---: | :---: | :---: |
> | Standard DDPM | 0.379 | 2.142 | 3.749  | 4.676 | 5.587 | 7.683 | 1.867e+1 | 2.091e+2 | 8.010e+3 | 1.402e+7 |
> | TAG(Ours) | 0.405 | 2.340 | 3.985 | 4.886 | 5.483 | 6.244 | 8.068 | 1.856e+1 | 2.444e+2 | 3.860e+5 |
> | Ratio(TAG/DDPM) | 1.069 | 1.092 | 1.063 | 1.045 | 0.981 | 0.813 | 0.432 | 0.089 | 0.031 | 0.028 |
>
> The above results demonstrate the effectiveness of TAG in reducing score approximation error, both in each timestep and for overall sampling trajectory.

---

> ### Author Response · Authors · 2025-11-23
> **Response to Reviewer fmcf (2/3)**
>
> ### [R1-2] Interaction between TAG and guidance gradient
>
> We highly appreciate your insightful comment. We conduct an additional experiment to verify whether Time Linked Score(TLS) vector conflicts with the original score function or guidance term. Specifically, in the CIFAR-10 experiment, we measure how the average cosine similarity between the TLS and the diffusion model’s original score function changes, as well as the cosine similarity between the TLS and the DPS guidance, under different TAG strengths ($w$). The result in **Table R1-4** shows that despite improving performance with increasing $w$, cosine similarity of TLS and other vectors remains stable, close to 0. Interestingly, TLS and original diffusion score become more aligned with higher TAG strength, indicating that applying TAG can synergistically help the diffusion model’s score approximation.
>
> **Table R1-4.** Cosine similarity analysis in CIFAR-10 experiments (1,024 samples for each).
>
> | TAG strength ($w$) | 0.0 | 0.5 | 1.0 | 1.5 | 2.0 | 3.0 | 4.0 | 5.0 |
> | :---- | :---: | :---: |:---: | :---: | :---: | :---: | :---: | :---: |
> | **Cos_sim(TLS, Original score)** | 0.097 | 0.091 | 0.083| 0.090 |  0.089 | 0.098 | 0.101 | 0.111 |
> | **Cos_sim(TLS, DPS guidance)** | -0.024 | -0.025 | -0.026 | -0.027 | -0.026 | -0.023 | -0.0214 | -0.020 |
> | **FID** ($\downarrow$) | 225.1 | 208.6 | 199.1 | 189.4 | 183.1 | **182.6** | 183.2 | 185.3 |
> | **Validity** ($\uparrow$) | 39.26 | 48.63 | 50.10 | 55.18 | 61.33 | 58.98 | **63.28** | 62.40 |
>
> Moreover, we would like to clarify that the interaction between TAG and conditional score serves as an intended regularization rather than a destructive conflict. While an external guidance optimizes the conditional likelihood $p(c \mid x)$ often pushing samples off-manifold, TAG maximizes the time-dependent data likelihood $p(t \mid x)$ to restore manifold adherence. As visualized in **Figure 2**, the vector field with external drift alone (left) diverges from the data distribution, whereas applying TAG (right) realigns and strengthens the field towards the high density region. Importantly, since the **timestep axis is orthogonal to the spatial data manifold** where $x_t$ evolves, TAG's temporal correction restores the valid noise level without conflicting with the spatial steering of the external guidance as shown by the experiments.

---

> ### Author Response · Authors · 2025-11-23
> **Response to Reviewer fmcf (3/3)**
>
> ### [R1-3] Robustness against imperfect time predictor
> We clarify that the overall diffusion model maintains high stability even when the Time Predictor provides imperfect classification, as TAG is highly resilient to such variations. This is because the nature of the predictor is to estimate the global noise variance—a fundamental property of the diffusion process—rather than fitting to the complex high-dimensional data distribution of individual samples. Consequently, the predictor captures the necessary corrective gradient direction $\nabla_x \log p(t|x)$ early in training without requiring precise probability calibration or memorization of sample-specific details.
>
> Empirically, this is confirmed by our sensitivity analysis regarding training progress. As shown in **Table 16 (Appendix E.4)** and our large-scale evaluation in **Table R1-5** below, the generative performance gap between using an early-stage predictor (10k steps) and a fully converged one (30k steps) is minimal.
>
> To further simulate the robustness of TAG under an imperfect time predictor, we conduct additional experiment where we inject Gaussian noise to intentionally corrupt the output of time predictor $\mathcal{N}(\mathbf{0}, \sigma^2\mathbf{I})$ into the predictor logits $\mathbf{z}$ (i.e., $\tilde{\mathbf{z}} = \mathbf{z} + \sigma \cdot \mathbf{\epsilon}$) prior to gradient computation. As shown in **Table R1-6**, TAG maintains significant performance gains over the baseline even under noise perturbation, degrading gracefully only at extreme levels.
>
> These results, combined with the effectiveness of lightweight architectures shown in **Table 9 (Appendix E.1)**—where a SimpleCNN utilizing only $\sim 8.5\%$ of the backbone parameters achieves performance comparable to heavier models—collectively confirm that the guidance term acts as a soft, directionally robust correction, ensuring stability even when the predictor is lightweight, not fully converged, or noisy.
>
> **Table R1-5** Predictor robustness on CIFAR-10 (10,000 samples)
> | Predictor Training Steps | **DPS**|**10k Steps (Early)** | **20k steps** | **30k Steps (Converged)**|
> | :--- | :---: | :---: |:---: |:---: |
> | **FID** ($\downarrow$) | 203.6 | 176.1 | 175.6|**174.5** |
> | **Accuracy** ($\uparrow$) | 41.6 | 48.5 | 51.2|**51.9** |
>
> **Table R1-6.** Robustness to logit noise Injection (1,000 samples)
> *(Stress test with additive Gaussian noise $\sigma$ on predictor logits)*
>
> | Noise Level ($\sigma$) | **DPS** | **0.0 (Standard)** | **0.1** | **0.5** | **1.0** | **2.0** | **5.0** | **10.0** |
> | :--- | :---: | :---: | :---: | :---: | :---: | :---: | :---: | :---: |
> | **FID** ($\downarrow$) | 218.9 | **195.7** | 200.9 | 199.3 | 196.1 | 214.7 | 283.2 | 348.7 |
> | **Accuracy** ($\uparrow$) | 41.4 | 50.3 | 50.4 | **53.2** | 52.7 | 48.9 | 36.7 | 26.6 |
>
>
>
> ---
> Thank you again for your constructive feedback. We are happy to engage in further discussion if there are any remaining questions.
>
> ---
> ### References
> [1] Song et al. Score-Based Generative Modeling through Stochastic Differential Equations. ICLR 2021.
>
> [2] Li et al. Understanding Generalizability of Diffusion Models
> Requires Rethinking the Hidden Gaussian Structure. NeurIPS 2024.

---

### Author Response · Authors · 2025-11-27
**General Response**

Dear Reviewers and AC,

We sincerely appreciate your valuable time and effort in reviewing our manuscript. Your insightful feedback has been instrumental in improving and strengthening our work.

Our paper introduces **Temporal Alignment Guidance (TAG)**, a novel framework to address the off-manifold phenomenon in diffusion models. To our best knowledge, this is the first work to identify and correct off-manifold errors via the **temporal axis**, providing a unique orthogonal contribution to existing geometric constraints. As highlighted by multiple reviewers, our key contributions include:

* **Novelty & Principled Approach:** TAG offers a fresh perspective on manifold correction supported by rigorous theoretical analysis. (fmcf, VLZo)
* **Usability & Efficiency:** It serves as a lightweight, plug-and-play module that requires no retraining of the backbone model, making it broadly applicable. (jUUY, MJq2, VLZo)
* **Broad Effectiveness:** The method has been thoroughly validated through consistent improvements in generation quality across diverse domains, including images, audio, and molecules. (fmcf, jUUY, MJq2)

In response to reviewers’ constructive questions and suggestions, we have conducted extensive investigations that solidified the empirical standing of TAG:

**1. Practical Hyperparameter Selection & Robustness**
We addressed concerns regarding manual tuning by clarifying that TAG is empirically tuning-free with a robust default ($\omega=1.0$) that consistently outperforms baselines. Additionally, we provided a cost-effective determination protocol using a small validation set for users seeking optional refinement. (jUUY, MJq2)

**2. Mitigating Intrinsic Sampling Errors**
We moved beyond theoretical bounds to prove TAG's utility in standard pipelines. By identifying the off-manifold artifacts (e.g., over-saturation) inherent in widely-used classifier-free guidance (CFG) and quantifying score approximation errors, we demonstrated that TAG serves as a fundamental correction mechanism. (fmcf, jUUY)

**3. Efficiency, Scalability, and Reliability**
We confirmed TAG's scalability to large-scale models with negligible inference latency (< 5%) and minimal training requirements. Furthermore, we demonstrated that the method remains robust even with suboptimal predictors (e.g., early checkpoints, noise injection), proving its reliance on the robust global noise variance structure rather than fragile probability calibration. (fmcf, jUUY, VLZo, MJq2)

All revisions, including the Notation Table and enhanced visualizations, are reflected in the final manuscript in $\color{Crimson}{\text{red}}$. We have carefully incorporated your valuable feedback and kindly request a re-evaluation of our work based on these improvements. We are happy to engage in further discussion if there are any remaining questions.

Sincerely,

The Authors

---

### Meta-Review · Area_Chair_uS6U · 2025-12-18

**Summary:**

This paper introduces Temporal Alignment Guidance (TAG), a novel framework for diffusion sampling that treats time as a latent variable and uses a learned time predictor to correct temporal misalignment during sampling. The core idea is conceptually interesting and well motivated, and the paper presents a coherent theoretical perspective on off-manifold drift induced by guidance and discretization error. Theoretical arguments, while informal, are plausible and generally sound, and the method is applied across a wide range of domains.

Despite these strengths, I recommend rejection at this time. The primary reason is that the empirical validation in computer vision, the most mature and best-understood application domain for diffusion models , does not operate in a meaningful performance regime, making the reported comparisons difficult to interpret and insufficient to support acceptance at ICLR.

The paper includes extensive experiments on CIFAR-10 and ImageNet intended to demonstrate the benefits of TAG under accelerated or guided sampling. However, in these settings the absolute sample quality is extremely poor by current standards. For example, CIFAR-10 FID values on the order of ~60–70 are reported even at ~50–60 NFE. At present, such FID levels are far outside the range achieved by standard fast samplers (e.g., DDIM, DPM-Solver), which typically obtain an order of magnitude lower FID at comparable or lower computational budgets.

When operating so far from a competent baseline, relative improvements in FID are difficult to interpret meaningfully. In these regimes, samples are substantially off the data manifold, and changes in FID may reflect reductions in gross artifacts or noise rather than genuine improvements in generative modeling quality. As a result, it is unclear whether TAG provides benefit in realistic diffusion pipelines, or whether the observed gains are specific to severely underperforming baselines. Crucially, the paper does not evaluate TAG in any setting where the underlying sampler already performs well by contemporary computer vision standards. There are no results combining TAG with widely used fast solvers such as DDIM or DPM-Solver, nor any demonstration that TAG is neutral or beneficial in standard sampling regimes. Because computer vision diffusion is a well-understood reference domain with clear expectations for baseline competence, this absence significantly weakens the empirical case.

Taking into account the very weak and incomplete CV evaluations and the mixed and partially luckworm reviews, I have decided to suggest rejection. So said, I do appreciate the ideas in this paper and I think that it would become much stronger and compelling if it included experiements in an acceptible performance regime (sub 10 FID), even if the differences were to be small or to disappear altogether in this range.

**Reviewer Concerns:**

The reviewers generally view the paper as presenting an interesting and novel conceptual contribution to diffusion sampling. Across reviews, there is agreement that the idea of treating time as a latent variable and explicitly correcting temporal misalignment during sampling is original and well motivated. Reviewers find the framing of “off-manifold drift” due to guidance or discretization to be intuitive and useful, and several note that the proposed Time-Linked Score offers a fresh perspective that connects naturally to classifier guidance–style methods. The method is perceived as broadly applicable and conceptually clean, and reviewers appreciate that it can be added to existing diffusion models without retraining the main score network.

The theoretical component is generally regarded as reasonable and supportive of the main idea. While reviewers acknowledge that the theoretical guarantees are informal and not fully rigorous, they largely view this as acceptable for the type of contribution being made. The derivations and propositions are seen as providing intuition rather than strict guarantees, and reviewers do not identify major correctness issues. Requests for clarification tend to focus on assumptions, notation, or how certain quantities behave in practice, rather than on flaws in the theoretical logic itself.

The breadth of experiments is consistently cited as a strength. Reviewers appreciate that the method is evaluated across multiple domains, including computer vision, molecular generation, and audio tasks, and that it is tested under different guidance schemes and sampling conditions. This breadth contributes to the perception that TAG is a general-purpose technique rather than a narrowly tuned solution. In several reviews, the diversity of settings is taken as evidence that the idea has merit beyond a single benchmark or task.

At the same time, reviewers’ concerns are largely centered on clarity, robustness, and interpretation rather than on baseline competence. Questions are raised about how sensitive the method is to the accuracy of the time predictor, how hyperparameters such as guidance strength should be chosen, and whether the proposed Time-Gap metric reliably correlates with perceptual or task-level quality. Some reviewers ask for clearer explanations, better organization, or additional discussion to help readers understand when and why the method is most effective. These critiques are generally framed as requests for clarification or minor extensions, not as fundamental objections.

Notably, the reviewers tend to accept the experimental setups as given and focus on relative improvements within those setups. They interpret the reported results as showing that TAG can mitigate degradation under difficult or perturbed sampling conditions, and they do not deeply question whether the absolute performance levels in certain experiments are representative of standard practice in mature domains such as computer vision. As a result, concerns about the external validity of the vision experiments or about the choice of sampling baselines are largely absent from the reviews.

Following the author response, reviewers generally indicate that many of their questions have been addressed satisfactorily. The rebuttal is seen as clarifying methodological details and strengthening confidence in the proposed approach. While some residual concerns remain, the overall tone after rebuttal trends toward cautious optimism, with reviewers expressing that the idea is sound and that the paper could be suitable for acceptance, albeit not without limitations.

In summary, the reviews paint a picture of a paper that is conceptually strong, theoretically reasonable, and broad in scope, with reviewers largely aligned in seeing value in the core idea. The critiques focus on clarity, robustness, and interpretation rather than on the fundamental suitability of the experimental regimes, and the reviews do not substantially challenge whether the empirical evidence, particularly in computer vision, meets the standards of current best practice.

**Reviewer Scores:**

Based on the written reviews and the authors’ responses, I believe the reviewers’ scores would likely have remained largely unchanged had they been able to participate fully in the discussion. The reviews already reflect a fairly stable assessment: reviewers generally view the core idea as interesting and novel, the theoretical motivation as reasonable though informal, and the experimental results as broadly supportive within the evaluated settings. The rebuttal appears to have satisfactorily addressed most reviewer-raised concerns related to clarity, robustness, and interpretation, but it does not introduce fundamentally new evidence that would clearly warrant a significant upward revision of scores.

At the same time, there is no indication in the reviews that any reviewer held strong unresolved objections that would have led to a substantial downward revision. Most critiques were framed as requests for clarification or additional discussion rather than as blocking issues, and the overall tone after rebuttal suggests cautious confidence rather than enthusiasm or strong skepticism.

As a result, I would expect reviewer scores, had discussion occurred, to cluster around their original values, with at most minor adjustments (e.g., a half-point increase or decrease) rather than any major shifts.

---

### Decision · Program_Chairs · 2026-01-26

Reject